# Charting the conformal manifold of holographic CFT$_2$'s

## Camille Eloy[1] and Gabriel Larios[2]

**1** Theoretische Natuurkunde, Vrije Universiteit Brussel, and
the International Solvay Institutes, Pleinlaan 2, B-1050 Brussels, Belgium
**2** Mitchell Institute for Fundamental Physics and Astronomy,
Texas A&M University, College Station, TX, 77843, USA

## Abstract

We construct new continuous families of AdS$_3 \times S^3 \times$ T$^4$ and AdS$_3 \times S^3 \times S^3 \times S^1$ solutions in heterotic and type II supergravities. These families are found in three-dimensional consistent truncations and controlled by 17 parameters, which include TsT $\beta$ deformations and encompass several supersymmetric sub-families. The different uplifts are constructed in a unified fashion by means of Exceptional Field Theory (ExFT). This allows the computation of the Kaluza-Klein spectra around the deformations, to test the stability of the solutions, and to interpret them holographically and as worldsheet models. To achieve this, we describe how the half-maximal SO(8, 8) ExFT can be embedded into E$_{8(8)}$ ExFT.

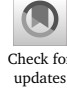
doi:10.21468/SciPostPhys.17.4.123

# 1 Introduction

In any Lorentz invariant quantum field theory in $d$ dimensions, operators can be classified according to their behaviour under the renormalisation group (RG), and for conformal field theories (CFTs) sitting at the fixed points of the RG flow this behaviour can be characterised by the operator's conformal dimension $\Delta$. For irrelevant operators, $\Delta$ exceeds the spacetime dimension and the RG flow takes the theory back to the original CFT. Conversely, relevant deformations are triggered by operators with $\Delta < d$ and RG flow drives the theory away from the starting point. A third class of deformations, called marginal, stay unaffected by changes in the energy scale. Instead, these marginal operators encode the space of theories into which the original theory can be deformed without breaking conformal invariance. This space is called the CFT's conformal manifold. Holographically, it corresponds to a continuous family of AdS solutions sharing the same cosmological constant, but having different internal spaces. Although there is no systematic way of constructing these gravity solutions from the CFT information, the AdS/CFT dictionary identifies marginal operators with modes in the bulk that are massless to all orders in the $1/N$ expansion.

Supersymmetry is expected to be required for holographic conformal manifolds to exist, as non-supersymmetric AdS solutions are believed to be unstable [1–3]. However, recent scrutiny [4] has revealed AdS$_4$ configurations that might evade this requirement, as all the standard decay channels, both perturbative and non-perturbative, are absent for this solution.

In AdS$_3$/CFT$_2$, the scenario could be richer and there is a long-standing counter-example [5,6] which is understood both in the field theory and gravity sides. It is based on current-current deformations of the two-dimensional CFT, which are known to be exactly marginal [7] despite possibly supersymmetry breaking. From the gravity perspective, these deformed solutions will remain in the small curvature regime for small values of the deformation parameters, and this assures that the deformed solution can also be studied in the supergravity approximation. More generally, in the supergravity approximation continuous deformations are identified with classically marginal operators, which will be exactly marginal only if they persist for finite values of $N$ and the CFT coupling. In the present work, we will mostly focus on the supergravity regime, and for this reason we will omit this distinction henceforth. However, we note that some quantum corrections can already be assessed in this limit from the $n$-point couplings among the supergravity modes [8].

The purpose of this note is to explore the landscape of continuously connected AdS$_3$ solutions in type IIB and heterotic supergravities, expanding the work of [9]. The class of theories we focus on is given by the near-horizon limit of NS5-F1 branes, and thus related by S-duality to the D1-D5 configuration [10–13] and have been recently studied in ref. [14–16]. These latter works, together with [17,18], conjecture that string theory on AdS$_3 \times S^3 \times S^3 \times S^1$ and AdS$_3 \times S^3 \times T^4$ are holographic duals to non-linear sigma models on symmetric SU(2)×U(1) and T$^4$ orbifolds, respectively. They take advantage of the absence of RR fluxes to encode the dynamics as a supersymmetric Wess-Zumino-Witten (WZW) model on the worldsheet, and for the most part focus on the tensionless string limit, where the supergravity description is not valid. The opposite limit, where all string excitations decouple, remains largely unexplored.

Previous works have studied deformations of the AdS$_3 \times S^3 \times S^3 \times S^1$ background in type IIB supergravity [19], and its AdS$_3 \times S^3 \times T^4$ counterpart in both the type IIB and heterotic theories [9]. This has shown very similar structures on both examples, and here we propose a generic framework to study their deformations for both half-maximal as well as maximal theories simultaneously. We enlarge our understanding of the landscape of deformations by exhibiting a 17-parameter family of solutions that includes cases which in 10$d$ correspond to Lunin-Maldacena TsT deformations [20] and Wilson loops analogous to the recently studied fibrations in [21–27]. The parameters generically break all supersymmetries, but in certain loci some supersymmetry is recovered. We further study the spectra of Kaluza-Klein excitations on these solutions, and discuss the perturbative stability of several supersymmetry breaking subfamilies. Additionally, given the fact that the deformations do not excite any RR fluxes, the solutions stay pure NSNS and can thus be described from a worldsheet point of view. This allows us to show that the marginal parameters induce $J\bar{J}$ operators on the worldsheet.

The techniques we employ to obtain these large conformal manifolds are based on a convenient feature of the AdS$_3 \times S^3 \times T^4$ and AdS$_3 \times S^3 \times S^3 \times S^1$ solutions: they admit consistent truncations to three-dimensional gauged supergravity. These are restrictions to a finite subset of modes in the Kaluza-Klein tower such that any solution of the three-dimensional gauged supergravity defines a solution of the full set of equations of motion in ten dimensions. Having a consistent truncation is a particularly valuable tool given that they give access to a subsector of the higher-dimensional theory using only the lower-dimensional dynamics. In the three-dimensional truncation, the theory for the modes retained has a scalar potential featuring stationary points. The solution at these points correspond to AdS$_3 \times \mathcal{K}$ solutions in ten dimensions, and in particular, marginal deformations correspond to flat directions in the 3$d$ potential, leading to continuous deformations of the internal manifold $\mathcal{K}$.

The existence of these consistent truncations can be exploited through the tools of Exceptional Field Theory (ExFT) [28,29]. ExFT is a reformulation of higher-dimensional supergravity making it formally covariant under the U duality group of the lower-dimensional theory obtained by toroidal reductions. The higher-dimensional fields are reorganised to mimic the ones

in lower dimensions, thus allowing the use of the U duality symmetry before the compactification already. This reformulation is extremely efficient to build and parameterise consistent truncations [30,31] and to compute Kaluza-Klein spectra [32–36] and higher-couplings [37] around any solution in the truncation. Most prominently, these techniques also apply to vacua preserving few or no (super)symmetries, which were beyond the reach of traditional methods. In this note, we focus on ExFTs based on the U duality groups of maximal and half-maximal supergravities in three dimensions, respectively given by $E_{8(8)}$ and $SO(8, n)$, which were constructed in ref. [38,39].

The rest of the paper is divided in two main parts. The first one exposes the technical tools necessary to the study, and can be skipped by readers only interested in the main results, which are presented in the second part. In sec. 2 we review the main features of maximal and half-maximal supergravities in three dimensions, and explain how the half-maximal theories can be embedded in their maximal counterparts. By these means, we will construct new $AdS_3 \times S^3 \times S^3 \times S^1$ and $AdS_3 \times S^3 \times T^4$ solutions in maximal supergravity through computations in the half-maximal theory. Sec. 3 introduces the $E_{8(8)}$ and $SO(8, n)$ exceptional field theories with an emphasis on their applications to the study of consistent truncations and Kaluza-Klein spectra. We show that $SO(8, 8)$ ExFT can be consistently embedded into its $E_{8(8)}$ analogue, and use this embedding to demonstrate that a consistent truncation of half-maximal supergravity automatically defines a consistent truncation in maximal supergravity. Finally, in sec. 4 we exemplify how this framework applies to the round $AdS_3 \times S^3 \times S^3 \times S^1$ and $AdS_3 \times S^3 \times T^4$ solutions in both type II and heterotic supergravity.

The second part is dedicated to the analysis of new families of marginal deformations of these solutions. This constitutes the main result of this note. In sec. 5, we present for each family the details of the ten-dimensional solution and explain how the deformation affects the spectrum of Kaluza-Klein modes. From the spectrum, we deduce possible supersymmetry enhancements and discuss the perturbative stability of the non-supersymmetric solutions by testing the masses of scalar fields against the Breitenlohner-Freedman bound [40]. All deformed solutions we present are purely NSNS and we use this fact in sec. 6 to study them from the point of view of the worldsheet action. This shows that these deformation parameters induce current-current operators of the original worldsheet model, and this can be used to predict the holographic CFT operators as combinations of $J\bar{J}$ deformations. We end in sec. 7 with some final comments and relegate further technical details to four appendices.

## 2 Gauged supergravities in $D = 3$

### 2.1 Half-maximal theories

The first instance of $AdS_3$ families leading to the $AdS_3 \times S^3 \times S^3 \times S^1$ and $AdS_3 \times S^3 \times T^4$ solutions mentioned above was found in [41] as a family of vacua in half-maximal $D = 3$ supergravity with four scalar multiplets. For theories containing $n$ scalar multiplets, the global symmetry of the ungauged theory is $SO(8, n)$ [42], and the pure supergravity multiplet containing the graviton and eight gravitini is supplemented by $8n$ scalars and spin-1/2 fermions. The former parameterise the manifold

$$\frac{SO(8, n)}{SO(8) \times SO(n)}, \tag{1}$$

and the gravitini and spin-1/2 fields transform respectively in the spinorial and co-spinorial of the denominator $SO(8)$. To describe the gauging, vectors can be included in this theory that are dual to the scalar and live in the adjoint representation of $SO(8, n)$ [43]. The gauging of these matter-coupled supergravities is specified by an embedding tensor $\Theta_{\bar{K}\bar{L}|\bar{M}\bar{N}}$, with indices in the

vector representation of SO(8, $n$). As customary, apart from introducing covariant derivatives[1]

$$D = d + \Theta_{\bar{K}\bar{L}|\bar{M}\bar{N}} A^{\bar{K}\bar{L}} T^{\bar{M}\bar{N}},\tag{2}$$

for $T^{\bar{M}\bar{N}}$ the generators of SO(8, $n$) in the relevant representation, such a gauging induces extra fermionic couplings and a potential for the scalars, respectively linear and quadratic in the embedding tensor. The Lagrangian of the gauged half-maximal theory reads [41, 43, 44]

$$e^{-1}\mathcal{L}_{\text{h.m.}} = R + \frac{1}{8} g^{\mu\nu} D_\mu M^{\bar{M}\bar{N}} D_\nu M_{\bar{M}\bar{N}} + e^{-1}\mathcal{L}_{\text{CS, h.m.}} - V_{\text{h.m.}} + \text{fermions},\tag{3}$$

with $M = \mathcal{V}\mathcal{V}^T$ for $\mathcal{V}_{\bar{M}}{}^{\bar{N}}$ the coset representative specifying the point in (1). The Chern-Simons kinetic term for the vectors is given by

$$\mathcal{L}_{\text{CS}} = -\varepsilon^{\mu\nu\rho} \Theta_{\bar{M}\bar{N}|\bar{P}\bar{Q}} A_\mu{}^{\bar{M}\bar{N}} \left( \partial_\nu A_\rho{}^{\bar{P}\bar{Q}} + \frac{1}{3} \Theta_{\bar{R}\bar{S}|\bar{U}\bar{V}} f^{\bar{P}\bar{Q},\bar{R}\bar{S}}{}_{\bar{X}\bar{Y}} A_\nu{}^{\bar{U}\bar{V}} A_\rho{}^{\bar{X}\bar{Y}} \right),\tag{4}$$

with $\varepsilon^{\mu\nu\rho}$ the constant Levi-Civita density and $f^{\bar{M}\bar{N},\bar{P}\bar{Q}}{}_{\bar{K}\bar{L}} = 4\delta_{[\bar{K}}{}^{[\bar{M}} \eta^{\bar{N}][\bar{P}} \delta_{\bar{L}]}{}^{\bar{Q}]}$ the structure constants of $\mathfrak{so}(8, n)$ for generators normalised as

$$T^{\bar{M}\bar{N}}{}_{\bar{P}}{}^{\bar{Q}} = 2\delta_{\bar{P}}{}^{[\bar{M}} \eta^{\bar{N}]\bar{Q}}.\tag{5}$$

Consistency of the gauging requires two constraints, one linear and the other quadratic in the embedding tensor. The linear constraint restricts the representations in which $\Theta_{\bar{K}\bar{L}|\bar{M}\bar{N}}$ can live. Given that it is antisymmetric in each pair of indices and symmetric under exchange of both pairs,[2] based only on its index structure it includes[3]

$$\left( \boxed{\phantom{x}} \otimes \boxed{\phantom{x}} \right)_{\text{sym}} \simeq \mathbf{1} \oplus \boxed{\phantom{xx}} \oplus \boxed{\phantom{xx}} \oplus \boxed{\phantom{x}}.\tag{6}$$

Supersymmetry of the gauged supergravity requires that not all representations in (6) appear in $\Theta$. In particular, one needs to implement the projection [44]

$$\mathbb{P}_{\boxplus} \Theta = 0,\tag{7}$$

which allows the embedding tensor to be parameterised as

$$\Theta_{\bar{K}\bar{L}|\bar{M}\bar{N}} = \theta_{\bar{K}\bar{L}\bar{M}\bar{N}} + \frac{1}{2}\left( \eta_{\bar{M}[\bar{K}} \theta_{\bar{L}]\bar{N}} - \eta_{\bar{N}[\bar{K}} \theta_{\bar{L}]\bar{M}} \right) + \theta\, \eta_{\bar{M}[\bar{K}} \eta_{\bar{L}]\bar{N}},\tag{8}$$

in terms of totally antisymmetric, symmetric traceless and singlet tensors. The second requirement, quadratic in the embedding tensor, is the invariance of $\Theta$ under gauge transformations generated by $\Theta$ itself. This amounts to the vanishing of

$$Q_{\bar{K}\bar{L}|\bar{M}\bar{N}|\bar{P}\bar{Q}} = -2\Theta_{\bar{K}\bar{L}|[\bar{M}}{}^{\bar{R}} \Theta_{\bar{N}]\bar{R}|\bar{P}\bar{Q}} - 2\Theta_{\bar{K}\bar{L}|[\bar{P}}{}^{\bar{R}} \Theta_{\bar{Q}]\bar{R}|\bar{M}\bar{N}},\tag{9}$$

with indices raised and lowered with the SO(8, $n$) invariant tensor $\eta_{\bar{M}\bar{N}}$. The space of non-trivial constraints can be computed to be

$$Q_{\bar{K}\bar{L}|\bar{M}\bar{N}|\bar{P}\bar{Q}} \subset \boxed{\phantom{xx}} \oplus \boxed{\phantom{x}} \oplus 2\times \boxed{\phantom{xx}} \oplus \boxed{\phantom{x}}.\tag{10}$$

---

[1] In 3$d$, all our indices have overbars so as to distinguish them from their ExFT counterparts, introduced in sec. 3.

[2] We do not consider gaugings of the trombone unless otherwise stated.

[3] We employ Young tableaux to refer to SO($N$) representations. For example, $\dim(\boxed{\phantom{x}}) = \frac{1}{2}(N-1)(N+2)$ and $\dim(\boxed{\phantom{xx}}) = \frac{1}{12}N(N-3)(N+1)(N+2)$.

The scalar potential and couplings describing the dynamics of the gauged $D = 3$ supergravity are determined entirely by the embedding tensor (8). The former, taking into consideration that the embedding tensors we are going to consider also satisfy the quadratic relation [39]

$$\theta_{[\bar{K}\bar{L}\bar{M}\bar{N}}\theta_{\bar{P}\bar{Q}\bar{R}\bar{S}]} = 0,\tag{11}$$

for them to be compatible with a generalised Scherk-Schwarz origin, is given by [41, 45]

$$\begin{aligned}
V_{\text{h.m.}} = \frac{1}{12}\,\theta_{\bar{K}\bar{L}\bar{M}\bar{N}}\theta_{\bar{P}\bar{Q}\bar{R}\bar{S}}\Big(&M^{\bar{K}\bar{P}}M^{\bar{L}\bar{Q}}M^{\bar{M}\bar{R}}M^{\bar{N}\bar{S}} - 6\,M^{\bar{K}\bar{P}}M^{\bar{L}\bar{Q}}\eta^{\bar{M}\bar{R}}\eta^{\bar{N}\bar{S}} \\
&+ 8\,M^{\bar{K}\bar{P}}\eta^{\bar{L}\bar{Q}}\eta^{\bar{M}\bar{R}}\eta^{\bar{N}\bar{S}} - 3\,\eta^{\bar{K}\bar{P}}\eta^{\bar{L}\bar{Q}}\eta^{\bar{M}\bar{R}}\eta^{\bar{N}\bar{S}}\Big) \\
+ \frac{1}{8}\,\theta_{\bar{K}\bar{L}}\theta_{\bar{P}\bar{Q}}\Big(&2\,M^{\bar{K}\bar{P}}M^{\bar{L}\bar{Q}} - 2\,\eta^{\bar{K}\bar{P}}\eta^{\bar{L}\bar{Q}} - M^{\bar{K}\bar{L}}M^{\bar{P}\bar{Q}}\Big) + 4\,\theta\,\theta_{\bar{K}\bar{L}}M^{\bar{K}\bar{L}} - 32\,\theta^2\,.
\end{aligned}\tag{12}$$

Critical points are those that annihilate

$$\begin{aligned}
\delta V_{\text{h.m.}} = \frac{1}{3}\,\theta_{\bar{K}\bar{L}\bar{M}\bar{N}}\theta_{\bar{P}\bar{Q}\bar{R}\bar{S}}\Big(&M^{\bar{K}\bar{P}}M^{\bar{L}\bar{Q}}M^{\bar{N}\bar{S}} - 3\,M^{\bar{K}\bar{P}}\eta^{\bar{L}\bar{Q}}\eta^{\bar{N}\bar{S}} + 2\,\eta^{\bar{K}\bar{P}}\eta^{\bar{L}\bar{Q}}\eta^{\bar{N}\bar{S}}\Big)j^{\bar{M}\bar{R}} \\
&+ \frac{1}{2}\Big(\theta_{\bar{M}\bar{P}}\theta_{\bar{N}\bar{Q}}M^{\bar{P}\bar{Q}} - \frac{1}{2}\theta_{\bar{M}\bar{N}}\theta_{\bar{P}\bar{Q}}M^{\bar{P}\bar{Q}}\Big)j^{\bar{M}\bar{N}} + 4\,\theta\,\theta_{\bar{M}\bar{N}}j^{\bar{M}\bar{N}}\,,
\end{aligned}\tag{13}$$

for arbitrary $j_{\bar{M}\bar{N}} \in \mathfrak{so}(8, n) \ominus (\mathfrak{so}(8) \oplus \mathfrak{so}(n))$. The rest of the couplings can be described through the dressed embedding tensor

$$T_{\bar{K}\bar{L}|\bar{M}\bar{N}} = (\mathcal{V}^{-1})_{\bar{K}}{}^{\bar{P}}(\mathcal{V}^{-1})_{\bar{L}}{}^{\bar{Q}}(\mathcal{V}^{-1})_{\bar{M}}{}^{\bar{R}}(\mathcal{V}^{-1})_{\bar{N}}{}^{\bar{S}}\Theta_{\bar{P}\bar{Q}|\bar{R}\bar{S}}\,,\tag{14}$$

which can be decomposed into $T_{\bar{K}\bar{L}\bar{M}\bar{N}}$, $T_{\bar{K}\bar{L}}$ and $T$ following (8). Given that fermions transform as representations of $SO(8) \times SO(n)$ in the denominator of (1), with the gravitini in the spinorial of $SO(8)$ and the spin-$1/2$ fields in the product of the co-spinorial of $SO(8)$ and the vector of $SO(n)$, it is useful to introduce indices $\bar{I}, \bar{A}, \dot{\bar{A}} \in [\![1, 8]\!]$ respectively in the vector, spinorial and co-spinorial of $SO(8)$, and hatted counterparts for $SO(n)$. This way, the fermion fields are denoted by $\psi^{\bar{A}}$ and $\chi^{\dot{\bar{A}}\hat{\bar{I}}}$.

In terms of the T-tensor (14), the bosonic masses are given by [41, 44, 46]

$$M_{(1)}{}^{\bar{M}\bar{N}}{}_{\bar{P}\bar{Q}} = \Big(\eta^{\bar{K}[\bar{M}}\eta^{\bar{N}]\bar{L}} - \delta^{\bar{K}[\bar{M}}\delta^{\bar{N}]\bar{L}}\Big)T_{\bar{K}\bar{L}|\bar{P}\bar{Q}}\,,\tag{15a}$$

$$M_{(0)\,\bar{M}\bar{N},\bar{P}\bar{Q}}^2\,j^{\bar{M}\bar{N}}j^{\bar{P}\bar{Q}} = m_{\bar{M}\bar{N},\bar{P}\bar{Q}}\,j^{\bar{M}\bar{N}}j^{\bar{P}\bar{Q}}\,,\tag{15b}$$

respectively for vectors and scalars. In the latter,

$$\begin{aligned}
m_{\bar{M}\bar{N},\bar{P}\bar{Q}} = {}&4\,T_{\bar{M}\bar{P}\bar{K}\bar{L}}\,T_{\bar{N}\bar{Q}\bar{R}\bar{S}}\,\delta^{\bar{K}\bar{R}}\delta^{\bar{L}\bar{S}} + \frac{4}{3}\,T_{\bar{M}\bar{U}\bar{K}\bar{L}}\,T_{\bar{P}\bar{V}\bar{R}\bar{S}}\,\delta_{\bar{N}\bar{Q}}\,\delta^{\bar{U}\bar{V}}\delta^{\bar{K}\bar{R}}\delta^{\bar{L}\bar{S}} \\
&- 4\,T_{\bar{M}\bar{P}\bar{K}\bar{L}}\,T_{\bar{N}\bar{Q}}{}^{\bar{K}\bar{L}} - 4\,T_{\bar{M}\bar{U}\bar{K}\bar{L}}\,T_{\bar{P}\bar{V}}{}^{\bar{K}\bar{L}}\,\delta_{\bar{N}\bar{Q}}\,\delta^{\bar{U}\bar{V}} + \frac{8}{3}\,T_{\bar{M}\bar{U}\bar{K}\bar{L}}\,T_{\bar{P}}{}^{\bar{U}\bar{K}\bar{L}}\,\delta_{\bar{N}\bar{Q}} \\
&+ 2\,T_{\bar{M}\bar{P}}\,T_{\bar{N}\bar{Q}} - T_{\bar{M}\bar{N}}\,T_{\bar{P}\bar{Q}} + 2\,T_{\bar{M}\bar{K}}\,T_{\bar{P}\bar{L}}\,\delta_{\bar{N}\bar{Q}}\,\delta^{\bar{K}\bar{L}} \\
&- T_{\bar{M}\bar{P}}\,T_{\bar{K}\bar{L}}\,\delta_{\bar{N}\bar{Q}}\,\delta^{\bar{K}\bar{L}} + 16\,T\,T_{\bar{M}\bar{P}}\,\delta_{\bar{N}\bar{Q}}\,,
\end{aligned}\tag{16}$$

and the $j^{\bar{M}\bar{N}}$ currents project adjoint indices onto the coset (1). The fermionic masses and couplings are specified by the $SO(8) \times SO(n)$-covariant fermion shifts, which read

$$\begin{cases}
A_1^{\bar{A}\bar{B}} = -\dfrac{1}{12}\gamma^{\bar{I}\bar{J}\bar{K}\bar{L}}{}_{\bar{A}\bar{B}}T_{\bar{I}\bar{J}\bar{K}\bar{L}} - \dfrac{1}{4}\delta^{\bar{A}\bar{B}}T_{\bar{I}\bar{I}} + 2\,\delta^{\bar{A}\bar{B}}T\,, \\[2mm]
A_2^{\bar{A}\dot{\bar{A}}\hat{\bar{I}}} = -\dfrac{1}{3}\gamma^{\bar{I}\bar{J}\bar{K}}{}_{\bar{A}\dot{\bar{A}}}T_{\bar{I}\bar{J}\bar{K}\hat{\bar{I}}} - \dfrac{1}{2}\gamma^{\bar{I}}{}_{\bar{A}\dot{\bar{A}}}T_{\bar{I}\hat{\bar{I}}}\,, \\[2mm]
A_3^{\dot{\bar{A}}\hat{\bar{I}}\dot{\bar{B}}\hat{\bar{J}}} = \dfrac{1}{12}\delta^{\hat{\bar{I}}\hat{\bar{J}}}\gamma^{\bar{I}\bar{J}\bar{K}\bar{L}}{}_{\dot{\bar{A}}\dot{\bar{B}}}T_{\bar{I}\bar{J}\bar{K}\bar{L}} + 2\gamma^{\bar{I}\bar{J}}{}_{\dot{\bar{A}}\dot{\bar{B}}}T_{\bar{I}\bar{J}\hat{\bar{I}}\hat{\bar{J}}} - 4\delta^{\dot{\bar{A}}\dot{\bar{B}}}\delta^{\hat{\bar{I}}\hat{\bar{J}}}T - 2\delta^{\dot{\bar{A}}\dot{\bar{B}}}T_{\hat{\bar{I}}\hat{\bar{J}}} + \dfrac{1}{4}\delta^{\dot{\bar{A}}\dot{\bar{B}}}\delta^{\hat{\bar{I}}\hat{\bar{J}}}T_{\bar{I}\bar{I}}\,,
\end{cases}\tag{17}$$

as

$$M^{\bar{A}\bar{B}}_{(3/2)} = -A_1^{\bar{A}\bar{B}}, \qquad M^{\bar{A}\bar{I}\bar{B}\bar{J}}_{(1/2)} = -A_3^{\bar{A}\bar{I}\bar{B}\bar{J}}. \qquad (18)$$

The SO(8) gamma matrices in (17) are constructed in appendix A.

Several choices for $n$ are relevant in string theory. The theory with four scalar multiplets was shown in [39] to arise from the truncation of $D = 6$ $\mathcal{N} = (1,1)$ and $\mathcal{N} = (2,0)$ supergravities, and the theory with $n = 8$ corresponds to the NSNS sector of the superstring [47,48]. In the following, we will review how half-maximal gauged supergravities based on SO(8,8) can be embedded into maximal supergravity in $D = 3$, which arises as a truncation of the type II superstrings. The addition of $n_v$ further scalar multiplets in $D = 3$ corresponds to the addition of $n_v$ vector multiplets in half-maximal $D = 10$ supergravity, which for $n_v = 16$ captures the Cartan subsector of the low-energy regime of the heterotic stings [49].

## 2.2 Maximal theories

To make contact with type IIB supergravity, we must embed the gauged SO(8,8) half-maximal theory into its maximal counterpart [50–52]. The matter content of this $\mathcal{N} = 16$ supergravity in three dimensions is comprised by the dreibein and 16 Majorana gravitino fields, which do not propagate degrees of freedom, together with 128 real scalar fields and 128 Majorana fermions. The scalars are coordinates of [53]

$$\frac{E_{8(8)}}{SO(16)} \supset \frac{SO(8,8)}{SO(8) \times SO(8)}, \qquad (19)$$

and together with the spin-1/2 fermions they represent the two inequivalent spinorial representations of the denominator SO(16). Despite redundant, to describe gaugings of this theory it is again useful to introduce the one-forms dual to the scalars, which furnish the adjoint representation of $E_{8(8)}$.

To describe the scalar dynamics, it is again convenient to represent the coset (19) in terms of a symmetric matrix

$$M_{\bar{\mathcal{M}}\bar{\mathcal{N}}} = \mathcal{V}_{\bar{\mathcal{M}}}{}^{\bar{\mathcal{P}}} \mathcal{V}_{\bar{\mathcal{N}}}{}^{\bar{\mathcal{Q}}} \Delta_{\bar{\mathcal{P}}\bar{\mathcal{Q}}}, \qquad (20)$$

with $\bar{\mathcal{M}} \in [\![1, 248]\!]$ labelling the adjoint representation of $E_{8(8)}$, and $\Delta_{\bar{\mathcal{P}}\bar{\mathcal{Q}}}$ a matrix such that $(t_{\bar{\mathcal{M}}})_{\bar{\mathcal{P}}}{}^{\bar{\mathcal{R}}} \Delta_{\bar{\mathcal{R}}\bar{\mathcal{Q}}}$ is symmetric if the generator $t_{\bar{\mathcal{M}}}$ is non-compact and anti-symmetric if compact. In terms of these fields, the Lagrangian reads

$$e^{-1}\mathcal{L}_{\text{max}} = R + \tfrac{1}{240} g^{\mu\nu} D_\mu M^{\bar{\mathcal{M}}\bar{\mathcal{N}}} D_\nu M_{\bar{\mathcal{M}}\bar{\mathcal{N}}} + e^{-1}\mathcal{L}_{\text{CS,max}} - V_{\text{max}} + \text{fermions}. \qquad (21)$$

The gauging is specified by a symmetric embedding tensor $X_{\bar{\mathcal{M}}\bar{\mathcal{N}}}$ such that covariant derivatives read

$$D = d + X_{\bar{\mathcal{M}}\bar{\mathcal{N}}} A^{\bar{\mathcal{M}}} t^{\bar{\mathcal{N}}}. \qquad (22)$$

An expression for the Chern-Simons contribution in (21) can be found in [50] and will not be needed in the sequel. For the gauging to preserve maximal supersymmetry, the embedding tensor must lie in the **1 ⊕ 3875** representation of $E_{8(8)}$ and obey the quadratic relation

$$X_{\bar{\mathcal{R}}\bar{\mathcal{P}}} X_{\bar{\mathcal{S}}(\bar{\mathcal{M}}} f_{\bar{\mathcal{N}})}{}^{\bar{\mathcal{R}}\bar{\mathcal{S}}} = 0 \iff [X_{\bar{\mathcal{M}}}, X_{\bar{\mathcal{N}}}]_{\bar{\mathcal{P}}}{}^{\bar{\mathcal{Q}}} = -X_{\bar{\mathcal{M}}\bar{\mathcal{N}}}{}^{\bar{\mathcal{R}}} X_{\bar{\mathcal{R}}\bar{\mathcal{P}}}{}^{\bar{\mathcal{Q}}}, \qquad (23)$$

with the gauge group generator in the adjoint representation defined in terms of the $E_{8(8)}$ structure constants as $X_{\bar{\mathcal{M}}\bar{\mathcal{N}}}{}^{\bar{\mathcal{P}}} = -X_{\bar{\mathcal{M}}\bar{\mathcal{Q}}} f^{\bar{\mathcal{Q}}}{}_{\bar{\mathcal{N}}}{}^{\bar{\mathcal{P}}}$ [54]. Throughout, $E_{8(8)}$ indices are raised and lowered with the Cartan-Killing form $\kappa_{\bar{\mathcal{M}}\bar{\mathcal{N}}}$ normalised as in equation (A.12).

For generic gaugings satisfying the above constraints, the potential and matter couplings are known in terms of SO(16)-covariant fermions shifts [50]. The former is also known to have a formally $E_{8(8)}$-covariant expression [55] given by

$$V_{\max} = X_{\bar{\mathcal{M}}\bar{\mathcal{N}}} X_{\bar{\mathcal{P}}\bar{\mathcal{Q}}} \Big( \frac{1}{28} M^{\bar{\mathcal{M}}\bar{\mathcal{P}}} M^{\bar{\mathcal{N}}\bar{\mathcal{Q}}} + \frac{1}{2} M^{\bar{\mathcal{M}}\bar{\mathcal{P}}} \kappa^{\bar{\mathcal{N}}\bar{\mathcal{Q}}} - \frac{3}{28} \kappa^{\bar{\mathcal{M}}\bar{\mathcal{P}}} \kappa^{\bar{\mathcal{N}}\bar{\mathcal{Q}}} - \frac{2}{6727} \kappa^{\bar{\mathcal{M}}\bar{\mathcal{N}}} \kappa^{\bar{\mathcal{P}}\bar{\mathcal{Q}}} \Big). \quad (24)$$

The embedding of half-maximal supergravity into the maximal theory then follows from[4]

$$
\begin{aligned}
E_{8(8)} &\supset & SO(8,8)\,, \\
\mathbf{248} &\to & \mathbf{120} + \mathbf{128}_s\,, \\
t^{\bar{\mathcal{M}}} &\to & \{t^{[\bar{M}\bar{N}]},\ t^{\bar{A}}\}\,,
\end{aligned}
\quad (26)
$$

with $\bar{M} \in [\![1,16]\!]$ labelling the vector representation of SO(8,8) as in sec. 2.1. The maximal embedding tensor thus decomposes under SO(8,8) into

$$\mathbf{1} \oplus \mathbf{3875} \to \mathbf{1} \oplus \mathbf{135} \oplus \mathbf{1820} \oplus \mathbf{1920}_c\,, \quad (27)$$

where one can recognise the three first representations as the ones appearing in (8). The spinorial representation $\mathbf{1920}_c$ cannot be excited in half-maximal supergravity, the SO(8,8) and $E_{8(8)}$ singlets can be identified, and the symmetric and four-fold antisymmetric tensors lie in the $\mathbf{3875}$ representation of $E_{8(8)}$. The explicit breaking of the embedding tensor components is [46]

$$X_{\bar{K}\bar{L}|\bar{M}\bar{N}} = 2\Theta_{\bar{K}\bar{L}|\bar{M}\bar{N}}\,, \qquad X_{\bar{A}\bar{B}} = -\theta\,\eta_{\bar{A}\bar{B}} + \frac{1}{48}\Gamma^{\bar{K}\bar{L}\bar{M}\bar{N}}_{\bar{A}\bar{B}}\theta_{\bar{K}\bar{L}\bar{M}\bar{N}}\,. \quad (28)$$

Details on the construction of $E_{8(8)}$ based on SO(8,8) can be found in appendix A. The chiral SO(8,8) gamma matrices are given by (A.7) if we work in the basis in which the SO(8,8) invariant metric assumes the diagonal form (A.2). In this basis, the charge conjugation matrix $\eta_{\bar{A}\bar{B}}$ is simply given by (A.3).

Breaking the $E_{8(8)}$ indices as in (26), the consistency condition (23) leads to three equations

$$X_{\bar{R}_1\bar{R}_2|\bar{P}_1\bar{P}_2} X_{\bar{S}_1\bar{S}_2|\bar{M}_1\bar{M}_2} f_{\bar{N}_1\bar{N}_2}{}^{\bar{R}_1\bar{R}_2,\bar{S}_1\bar{S}_2} = 0\,, \quad (29a)$$

$$X_{\bar{A}\bar{B}} X_{\bar{S}_1\bar{S}_2|\bar{M}_1\bar{M}_2} f_{\bar{C}}{}^{\bar{A}\bar{S}_1\bar{S}_2} = 0\,, \quad (29b)$$

$$X_{\bar{A}\bar{B}} X_{\bar{C}\bar{D}} f_{\bar{N}_1\bar{N}_2}{}^{\bar{A}\bar{C}} = 0\,. \quad (29c)$$

The first relation leads to (9), transforming as (10), upon the decomposition (28). The equations (29b) and (29c) imply extra compatibility conditions transforming in the $\mathbf{35} \oplus \mathbf{6435}_c$ of SO(8,8) [46] for the half-maximal gauging to admit an embedding into the maximal theory. Moreover, for the theory to be obtainable by Scherk-Schwarz reduction from type II/eleven-dimensional supergravity, the embedding tensor must also satisfy [19]

$$X_{\bar{\mathcal{M}}\bar{\mathcal{N}}} X^{\bar{\mathcal{M}}\bar{\mathcal{N}}} + \frac{1}{1922} \big( X_{\bar{\mathcal{M}}}{}^{\bar{\mathcal{M}}} \big)^2 = 0\,. \quad (30)$$

Supergravity vacua are solutions to the equation

$$\delta V_{\max} = \frac{1}{14} X_{\bar{\mathcal{M}}\bar{\mathcal{N}}} X_{\bar{\mathcal{P}}\bar{\mathcal{Q}}} \big( M^{\bar{\mathcal{N}}\bar{\mathcal{Q}}} + 7\kappa^{\bar{\mathcal{N}}\bar{\mathcal{Q}}} \big) j^{\bar{\mathcal{M}}\bar{\mathcal{P}}}\,, \quad (31)$$

---

[4]Given the breaking (26), the summing rule for $E_{8(8)}$ indices acquires extra combinatorial factors, e.g.

$$U^{\bar{\mathcal{M}}} V_{\bar{\mathcal{M}}} = \frac{1}{2} U^{\bar{M}\bar{N}} V_{\bar{M}\bar{N}} + U^{\bar{A}} V_{\bar{A}}\,. \quad (25)$$

with arbitrary $j^{\bar{\mathcal{M}}\bar{\mathcal{P}}} \in \mathfrak{e}_{8(8)} \ominus \mathfrak{so}(16)$. For a theory fulfilling (29), given a solution of (13) it automatically solves (31). The consistency of this truncation is guaranteed by the "fermion number" $\mathbb{Z}_2$ symmetry that acts on the SO(8,8) indices in (26) as

$$X^{[\bar{M}\bar{N}]} \mapsto X^{[\bar{M}\bar{N}]}, \qquad X^{\bar{A}} \mapsto -X^{\bar{A}}. \tag{32}$$

Instead of using the embedding (19) to construct the $E_{8(8)}/SO(16)$ coset representatives and define the dressed embedding tensor

$$\hat{T}_{\bar{\mathcal{M}}\bar{\mathcal{N}}} = (\mathcal{V}^{-1})_{\bar{\mathcal{M}}}{}^{\bar{\mathcal{P}}}(\mathcal{V}^{-1})_{\bar{\mathcal{N}}}{}^{\bar{\mathcal{Q}}}X_{\bar{\mathcal{P}}\bar{\mathcal{Q}}}, \tag{33}$$

one can equivalently build it as

$$\hat{T}_{\bar{K}\bar{L}|\bar{M}\bar{N}} = 2\,T_{\bar{K}\bar{L}|\bar{M}\bar{N}}, \qquad \hat{T}_{\bar{A}\bar{B}} = -T\,\eta_{\bar{A}\bar{B}} + \tfrac{1}{48}\,\Gamma^{\bar{K}\bar{L}\bar{M}\bar{N}}_{\bar{A}\bar{B}}\,T_{\bar{K}\bar{L}\bar{M}\bar{N}}, \tag{34}$$

in terms of the dressed embedding tensor in (14) for the half-maximal supergravity. At the different solutions of (31), the masses of the bosonic modes captured in gauged supergravity sit among the eigenvalues of

$$M_{(1)}{}^{\bar{\mathcal{M}}}{}_{\bar{\mathcal{N}}} = -\left(\Delta^{\bar{\mathcal{M}}\bar{\mathcal{P}}} + \kappa^{\bar{\mathcal{M}}\bar{\mathcal{P}}}\right)\hat{T}_{\bar{\mathcal{P}}\bar{\mathcal{N}}}, \tag{35a}$$

$$M^2_{(0).\bar{\mathcal{A}}\bar{\mathcal{B}}} = \mathcal{P}_{\bar{\mathcal{A}}}{}^{\bar{\mathcal{M}}\bar{\mathcal{N}}}\mathcal{P}_{\bar{\mathcal{B}}}{}^{\bar{\mathcal{P}}\bar{\mathcal{Q}}}\left[\tfrac{1}{7}\hat{T}_{\bar{\mathcal{M}}\bar{\mathcal{P}}}\hat{T}_{\bar{\mathcal{N}}\bar{\mathcal{Q}}} + \left(\tfrac{1}{7}\Delta^{\bar{\mathcal{K}}\bar{\mathcal{L}}} + \kappa^{\bar{\mathcal{K}}\bar{\mathcal{L}}}\right)\hat{T}_{\bar{\mathcal{M}}\bar{\mathcal{K}}}\hat{T}_{\bar{\mathcal{L}}\bar{\mathcal{P}}}\Delta_{\bar{\mathcal{N}}\bar{\mathcal{Q}}}\right], \tag{35b}$$

for vectors and scalars, respectively. In the mass matrix for the scalars, $\mathcal{P}_{\bar{\mathcal{A}}}{}^{\bar{\mathcal{M}}\bar{\mathcal{N}}} = (t_{\bar{\mathcal{A}}})_{\bar{\mathcal{P}}}{}^{\bar{\mathcal{M}}}\Delta^{\bar{\mathcal{P}}\bar{\mathcal{N}}}$ are the projectors onto the non-compact generators of (19), with $\bar{\mathcal{A}} \in [\![1,128]\!]$ labelling the spinorial of SO(16).

As for the scalars, fermion mass matrices for the maximal theory are written naturally in SO(16)-covariant form [50,51],[5]

$$\hat{A}_{1\bar{M}\bar{N}} = \frac{16}{7}\theta\,\eta_{\bar{M}\bar{N}} + \frac{2}{7}\hat{T}_{\bar{M}\bar{P},\bar{N}\bar{Q}}\,\eta^{\bar{P}\bar{Q}},$$

$$\hat{A}_{2\bar{M}.\bar{\mathcal{A}}} = -\frac{2}{7}\Gamma^{\bar{N}}_{\bar{\mathcal{A}}\bar{\mathcal{A}}}\,\eta^{\bar{\mathcal{A}}\bar{\mathcal{B}}}\,\hat{T}_{\bar{M}\bar{N}\,\bar{\mathcal{B}}}, \tag{36}$$

$$\hat{A}_{3.\bar{\mathcal{A}}\bar{\mathcal{B}}} = 4\theta\,\eta_{\bar{\mathcal{A}}\bar{\mathcal{B}}} + \frac{1}{24}\Gamma^{\bar{M}\bar{N}\bar{P}\bar{Q}}_{\bar{\mathcal{A}}\bar{\mathcal{B}}}\,\hat{T}_{\bar{M}\bar{N}\bar{P}\bar{Q}},$$

with indices $\bar{M} \in [\![1,16]\!]$, and $\bar{\mathcal{A}} \in [\![1,128]\!]$ respectively in the vector and co-spinorial of SO(16). As discussed in appendix A, the invariant tensors $\eta_{\bar{M}\bar{N}}$, $\eta_{\bar{\mathcal{A}}\bar{\mathcal{B}}}$ and $\eta_{\bar{\mathcal{A}}\bar{\mathcal{B}}}$ for this signature are simply given by identity matrices and the SO(16) components of the dressed embedding tensor follow from (33) under the decomposition in (A.14).

As in the half-maximal case, the eigenvalues of these matrices also encompass non-physical modes such as the vectors which are not gauged by the embedding tensor (and therefore sit outside the gauge group), and massless scalars that serve as Goldstone modes for the massive vectors.

## 2.3 Gaugings for $S^3 \times M^4$

In the following, unless otherwise stated, we will restrict ourselves to the case $n = 8$. As will become apparent in sec. 3, a convenient basis to describe the SO(8,8)-supergravities is such

---

[5]The coefficients for the terms in $\theta$ are not tested by our solutions.

that the invariant metric $\eta_{\bar{M}\bar{N}}$ is given by

$$\eta_{\bar{M}\bar{N}} = \begin{pmatrix} 0 & 1 & 0 & 0 & 0 & 0 & 0 & 0 \\ 1 & 0 & 0 & 0 & 0 & 0 & 0 & 0 \\ 0 & 0 & 0 & \mathbb{1}_3 & 0 & 0 & 0 & 0 \\ 0 & 0 & \mathbb{1}_3 & 0 & 0 & 0 & 0 & 0 \\ 0 & 0 & 0 & 0 & 0 & \mathbb{1}_3 & 0 & 0 \\ 0 & 0 & 0 & 0 & \mathbb{1}_3 & 0 & 0 & 0 \\ 0 & 0 & 0 & 0 & 0 & 0 & 0 & 1 \\ 0 & 0 & 0 & 0 & 0 & 0 & 1 & 0 \end{pmatrix}, \tag{37}$$

according to the breaking

$$\begin{aligned} \mathrm{SO}(8,8) &\supset \mathrm{SO}(1,1) \times \mathrm{GL}(3) \times \mathrm{GL}(3) \times \mathrm{SO}(1,1), \\ X^{\bar{M}} &\longrightarrow \{X^{\bar{0}}, X_{\bar{0}}, X^{\bar{m}}, X_{\bar{m}}, X^{\bar{i}}, X_{\bar{i}}, X^{\bar{7}}, X_{\bar{7}}\}. \end{aligned} \tag{38}$$

This choice aligns the $D = 3$ supergravity with the coordinates that solve the section constraint in Exceptional Field Theory. For this reason, we take the ranges of the indices above as $\bar{m} \in [\![1,3]\!]$ and $\bar{i} \in [\![4,6]\!]$. In this basis, the class of embedding tensors determining the half-maximal supergravities of interest are specified by the choice

$$\theta_{\bar{0}\bar{0}} = -4\sqrt{1+\alpha^2}, \qquad \theta_{\bar{M}\bar{N}\bar{P}\bar{0}} = -\frac{1}{2}X_{\bar{M}\bar{N}\bar{P}}, \tag{39}$$

with

$$\begin{aligned} X_{\bar{m}\bar{n}\bar{p}} &= \varepsilon_{\bar{m}\bar{n}\bar{p}}, & X_{\bar{m}}{}^{\bar{n}\bar{p}} &= \varepsilon_{\bar{m}\bar{n}\bar{p}}, & X^{\bar{m}}{}_{\bar{n}}{}^{\bar{p}} &= \varepsilon_{\bar{m}\bar{n}\bar{p}}, & X^{\bar{m}\bar{n}}{}_{\bar{p}} &= \varepsilon_{\bar{m}\bar{n}\bar{p}}, \\ X_{\bar{i}\bar{j}\bar{k}} &= \alpha\,\varepsilon_{\bar{i}\bar{j}\bar{k}}, & X_{\bar{i}}{}^{\bar{j}\bar{k}} &= \alpha\,\varepsilon_{\bar{i}\bar{j}\bar{k}}, & X^{\bar{i}}{}_{\bar{j}}{}^{\bar{k}} &= \alpha\,\varepsilon_{\bar{i}\bar{j}\bar{k}}, & X^{\bar{i}\bar{j}}{}_{\bar{k}} &= \alpha\,\varepsilon_{\bar{i}\bar{j}\bar{k}}, \end{aligned} \tag{40}$$

and $\alpha \in \mathbb{R}^+$ a free parameter.[6] For $\alpha \neq 0$, this theory admits an uplift into ten-dimensional supergravities on $S^3 \times S^3 \times S^1$. As we will see in eq. (124), the ratio of the $S^3$ radii is then given by $\alpha$. For $\alpha = 0$, the uplift is on $S^3 \times \mathrm{T}^4$ and the sphere has unit radius, *c.f.* eq. (129).

The embedding tensor described in [41] can be obtained by taking the $\alpha \to 0$ limit of (39) and truncating SO(8, 8) down to SO(8, 4). For generic $\alpha$, the half-maximal supergravity in $3d$ resulting from (39) has gauge group

$$G^{\alpha \neq 0} = \left[\mathrm{SO}(3) \ltimes T^3\right]^4 \times T^2, \tag{41}$$

which, for $\alpha = 0$, reduces to

$$G^{\alpha = 0} = \left[\mathrm{SO}(3) \ltimes T^3\right]^2 \times T^8, \tag{42}$$

with the remaining SO(4) becoming a global symmetry.

One can verify that the SO(8, 8) embedding tensor in (39) does verify the quadratic constraint (9) and also the compatibility conditions with maximal supergravity. In fact, it satisfies the stronger relations

$$\theta_{\bar{M}}{}^{\bar{P}\bar{Q}\bar{R}}\theta_{\bar{N}\bar{P}\bar{Q}\bar{R}} = 0, \qquad \theta_{[\bar{K}\bar{L}\bar{M}\bar{N}}\theta_{\bar{P}\bar{Q}\bar{R}\bar{S}]} = 0, \qquad \theta_{\bar{M}\bar{N}}\theta^{\bar{M}\bar{N}} = 0, \tag{43}$$

after which the others automatically follow. Upon embedding the half-maximal embedding tensor (39) into its $\mathrm{E}_{8(8)}$ counterpart via (28), these identities also guarantee that (30) holds, and therefore the resulting embedding tensor can be obtained via generalized Scherk-Schwarz reduction of $\mathrm{E}_{8(8)}$-ExFT.

---

[6]The sign of $\alpha$ only affects the chirality of the fermionic modes, and can be taken to be positive without loss of generality. Following the ten-dimensional uplifts, its range can in fact be restricted to $[0, 1]$ (see eq. (202)).

In the maximal theory, the gauge groups (41) and (42) are promoted into

$$G^{\alpha \neq 0} = \left[ SO(3)^4 \ltimes \Sigma \right] \times \left( T^1 \right)^2, \tag{44}$$

for non-vanishing $\alpha$, which reduces to

$$G^{\alpha = 0} = \left[ SO(3)^2 \ltimes \Sigma' \right] \times \left( T^1 \right)^8, \tag{45}$$

in the $\alpha = 0$ case. Here $\Sigma$ is a nilpotent subalgebra decomposing as

$$\Sigma \simeq \mathcal{T}_{12} \oplus \hat{\mathcal{T}}_{32}, \tag{46}$$

where $\mathcal{T}_{12}$ is an abelian subalgebra transforming in the adjoint of $SO(3)^4$, and $\hat{\mathcal{T}}_{32}$ represents two copies of the $(\frac{1}{2}, \frac{1}{2}, \frac{1}{2}, \frac{1}{2})$ of $SO(3)^4$ which close into $\mathcal{T}_{12}$. In (45), the nilpotent subalgebra is $\Sigma' \sim \mathcal{T}_6 \oplus \hat{\mathcal{T}}_{32}$ now representing the adjoint of $SO(3)^2$ and eight copies of its bi-spinor representation, which close into $\mathcal{T}_6$. The groups in (44) and (45) have the expected structure of gauged groups of three-dimensional Chern-Simons gauged supergravity [52] (see also sec. 3.2 of ref. [56]).

## 2.4 Solutions

In the half-maximal theory, a family of solutions annihilating (13) with embedding tensor (39) is given by the natural inclusion of the two-parameter locus found in the $SO(8, 4)$ theory [41] into $SO(8, 8)$. In the basis (38) and with the generators of $\mathfrak{so}(8, 8)$ normalised as is (5), it can be characterised by the representative

$$\mathcal{V} = \exp\left[ -\omega\, T^{\bar{3}}{}_{\bar{3}} - \frac{\omega \zeta}{1 - e^{-\omega}} \left( T^{\bar{3}\bar{7}} - T^{\bar{3}}{}_{\bar{7}} \right) \right] = \begin{pmatrix} \mathbb{1}_4 & 0 & 0 & 0 & 0 & 0 & 0 \\ 0 & e^{-\omega} & 0 & e^{\omega} \zeta^2 & 0 & \zeta & -\zeta \\ 0 & 0 & \mathbb{1}_2 & 0 & 0 & 0 & 0 \\ 0 & 0 & 0 & e^{\omega} & 0 & 0 & 0 \\ 0 & 0 & 0 & 0 & \mathbb{1}_6 & 0 & 0 \\ 0 & 0 & 0 & e^{\omega} \zeta & 0 & 1 & 0 \\ 0 & 0 & 0 & -e^{\omega} \zeta & 0 & 0 & 1 \end{pmatrix}. \tag{47}$$

All points in this family of solutions share the AdS radius

$$\ell^2_{\text{AdS}} = -\frac{2}{V_0} = \frac{1}{1 + \alpha^2}. \tag{48}$$

For $\alpha \neq 0$, the preserved gauge group out of (41) is $SO(4) \times SO(2) \times SO(2)$ at generic values of the parameters, whilst it reduces to $SO(2) \times SO(2)$ in the $\alpha = 0$ case. There are special loci where symmetry enhances. On the line

$$\zeta^2 = 1 - e^{-2\omega}, \tag{49}$$

two more vectors become massless,[7] and the gauge symmetry becomes $SO(4) \times SO(3) \times SO(2)$ for $\alpha \neq 0$. At the scalar origin, it further enhances to $SO(4) \times SO(4)$. For $\alpha = 0$, one of these $SO(4)$ factors is always absent from the gauge group, and instead there is a global factor. Whenever $\zeta = 0$, this global symmetry is $SO(4)$, which is broken down to $SO(3)_{\text{diag}}$ otherwise.

---

[7]We consider massless vectors and gravitini in spite of the fact that, together with the massless graviton, in $D = 3$ they are non-propagating. We find this useful as they correspond to the unbroken (super-)symmetry of the solution.

At generic points of the two-parameter family no supersymmetry is preserved. As discussed in [41], the symmetry enhancement at (49) corresponds to the locus where four gravitini become massless, resulting in $\mathcal{N} = (0,4)$ preserved supersymmetry. Away from the supersymmetric locus, stability is not guaranteed, as can already be observed at the gauged supergravity level. The modes that trigger instabilities can already be found in the $SO(8,4)$ truncation with $\alpha = 0$ of [41].

The family of solutions (47) can be embedded into the sigma model (19) pertaining to the maximal theory. Following (26), the representative reads

$$\mathcal{V}_{\bar{\mathcal{M}}}{}^{\bar{\mathcal{N}}} = \exp\Big[-\omega f^{\bar{3}}{}_{\bar{3}} - \frac{\omega\zeta}{1-e^{-\omega}}\big(f^{\bar{3}\bar{7}} - f^{\bar{3}}{}_{\bar{7}}\big)\Big], \tag{50}$$

with indices in the (38) basis. As shown in [9], for $\alpha = 0$ this family can be uplifted into type IIB supergravity on an $AdS_3 \times S^3 \times T^4$ background, with the parameters $\omega$ and $\zeta$ controlling the squashing of the $S^3$ and its fibration over one of the torus directions. With this intuition, a detailed analysis of (13) allows us to promote the solution (50) into a 15-parameter family for generic $\alpha$, with two extra moduli in the case $\alpha = 0$. The coset representative depending on these 17 parameters can be given as

$$
\begin{aligned}
\mathcal{V}_{\bar{\mathcal{M}}}{}^{\bar{\mathcal{N}}} = \exp\Big[ &-\omega f^{\bar{3}}{}_{\bar{3}} - \frac{\omega}{1-e^{-\omega}}\big(\chi_1 f^{\bar{3}\bar{7}} + \chi_2 f_{\bar{3}}{}^{\bar{7}} + \beta_1 f^{\bar{3}}{}_{\bar{7}} + \beta_2 f_{\bar{3}\bar{7}}\big) \\
&-\tilde{\omega} f^{\bar{6}}{}_{\bar{6}} - \frac{\tilde{\omega}}{1-e^{-\tilde{\omega}}}\big(\tilde{\chi}_1 f^{\bar{6}\bar{7}} + \tilde{\chi}_2 f_{\bar{6}}{}^{\bar{7}} + \tilde{\beta}_1 f^{\bar{6}}{}_{\bar{7}} + \tilde{\beta}_2 f_{\bar{6}\bar{7}}\big) \\
&-\Xi_1 f^{\bar{3}\bar{6}} - \Xi_2 f^{\bar{3}}{}_{\bar{6}} - \Xi_3 f_{\bar{3}}{}^{\bar{6}} - \Xi_4 f_{\bar{3}\bar{6}} - \sigma_4 f^{\bar{4}}{}_{\bar{4}} - \sigma_5 f^{\bar{5}}{}_{\bar{5}} - \sigma_7 f^{\bar{7}}{}_{\bar{7}}\Big],
\end{aligned}
\tag{51}
$$

with $\sigma_4$ and $\sigma_5$ stabilised to zero in the $\alpha \neq 0$ case. This class of solutions generalises the ones found in ref. [19] and [9], as will be described in sec. 5. This conformal manifold is entirely contained inside $SO(7,7) \subset SO(8,8) \subset E_{8(8)}$. Despite intensive search, no solution has been found in the half-maximal theory where excited scalars lie outside this $SO(7,7)$. Generically, the gauge group breaks to $SO(2)^4$ for $\alpha \neq 0$ and $SO(2)^2$ for $\alpha = 0$, and all supersymmetries are broken. At certain loci, partial (super-)symmetry enhancements take place, as will be discussed for the computation of the Kaluza-Klein spectra in sec. 5. As will also be described in that section, the solutions with non-vanishing values for the $\beta$'s and $\omega$ correspond to TsT transformations of the undeformed background. In fact, some subfamilies uplift to the standard Lunin-Maldecena deformations [20]. It is a remarkable characteristic of $D = 3$ supergravity that such $\beta$ deformations can be captured in a consistent truncation, unlike in the otherwise similar $AdS_4 \times S^7$ and $AdS_5 \times S^5$ solutions [20, 57, 58].

From a $3d$ perspective, the parameters in (51) span $\mathbb{R}^{17}$. Nevertheless, as will be seen in sec. 5, geometric identifications in $10d$ render the $\chi_i$ and $\tilde{\chi}_i$ moduli periodic. Similarly, string theory dualities [20] also compactify the $\beta$ directions.

## 3 Exceptional field theories in 3$d$

Exceptional field theories are duality covariant formulations of supergravity theories in higher dimensions, *i.e.* before any dimensional reduction. In the following we are interested in the reformulations that make explicit the duality symmetries that appear in reductions from heterotic and type II supergravities down to three dimensions, namely the $SO(8,n)$ and $E_{8(8)}$ exceptional field theories of ref. [39] and [38]. We review here these constructions and explain how to use them to build and study new solutions in ten dimensions for both heterotic and type II thanks to consistent truncations down to half-maximal supergravity in three dimensions.

## 3.1 Review of $SO(8, n)$ exceptional field theory

We are interested in the $SO(8, n)$-covariant reformulation of half-maximal $10d$ supergravity, first constructed in ref. [39], to make contact with the $3d$ gauged supergravity in sec. 2.1. The bosonic fields of such an extended field theory are

$$\{g_{\mu\nu},\ \mathcal{M}_{MN},\ \mathcal{A}_\mu{}^{MN},\ \mathcal{B}_{\mu MN}\},\tag{52}$$

with $\mu \in [\![0, 2]\!]$ and $M, N \in [\![1, 8 + n]\!]$ in the fundamental of $SO(8, n)$. The metric $g_{\mu\nu}$ is a $3d$ metric, $\mathcal{M}_{MN}$ is the generalised metric parameterising the coset $SO(8, n)/(SO(8) \times SO(n))$ and $\mathcal{A}_\mu{}^{MN}$ parameterise the gauge fields of the gauged supergravity. The gauge fields $\mathcal{B}_{\mu MN}$ are covariantly constrained and necessary for the gauge algebra to close, as we will review below. The internal indices of both $\mathcal{A}_\mu{}^{MN}$ and $\mathcal{B}_{\mu MN}$ belong to the adjoint representation of $SO(8, n)$. All these fields depend on 3 external coordinates $x^\mu$ and internal ones $Y^{MN}$ in the adjoint representation of $SO(8, n)$. Their dependence on $Y^{MN}$ is subject to the section constraints

$$\partial_{[MN} \otimes \partial_{PQ]} = 0,\tag{53a}$$

$$\eta^{PQ}\partial_{MP} \otimes \partial_{NQ} = 0,\tag{53b}$$

such that there are only 7 physical coordinates $y^i$ among $Y^{MN}$. The $\otimes$ product in eq. (53) means that the derivatives act on any combination of fields or gauge parameters.

The Lagrangian of the $SO(8, n)$ exceptional field theory is

$$\mathscr{L}^{SO(8,n)} = \sqrt{|g|}\left(\widehat{R}^{SO(8,n)} + \frac{1}{8}D_\mu\mathcal{M}_{MN}D^\mu\mathcal{M}^{MN} + \mathscr{L}^{SO(8,n)}_{\text{int}}\right) + \mathscr{L}^{SO(8,n)}_{\text{CS}}.\tag{54}$$

The first term is an $SO(8, n)$ covariantisation of the scalar curvature (see ref. [39] for more details). The second term is the kinetic term for the generalised metric, and the Chern-Simons term, which ensures the on-shell duality between scalars and vectors, is given by[8]

$$\mathscr{L}^{SO(8,n)}_{\text{CS}} = 2\,\varepsilon^{\mu\nu\rho}\Big(F_{\mu\nu}{}^{MN}\mathcal{B}_{\rho MN} + \partial_\mu\mathcal{A}_{\nu N}{}^K\partial_{KM}\mathcal{A}_\rho{}^{MN} - \frac{2}{3}\partial_{MN}\partial_{KL}\mathcal{A}_\mu{}^{KP}\mathcal{A}_\nu{}^{MN}\mathcal{A}_{\rho P}{}^L$$

$$+ \frac{2}{3}\mathcal{A}_\mu{}^{LN}\partial_{MN}\mathcal{A}_\nu{}^M{}_P\partial_{KL}\mathcal{A}_\rho{}^{PK} - \frac{4}{3}\mathcal{A}_\mu{}^{LN}\partial_{MP}\mathcal{A}_\nu{}^M{}_N\partial_{KL}\mathcal{A}_\rho{}^{PK}\Big),\tag{55}$$

where $F_{\mu\nu}{}^{MN}$ are the Yang-Mills field strength associated to $\mathcal{A}_\mu{}^{MN}$ (see eq. (2.55) of ref. [39] for an explicit expression). Finally, the potential is [36]

$$\mathscr{L}^{SO(8,n)}_{\text{int}} = \frac{1}{8}\partial_{KL}\mathcal{M}_{MN}\partial_{PQ}\mathcal{M}^{MN}\mathcal{M}^{KP}\mathcal{M}^{LQ} + \partial_{MK}\mathcal{M}^{NP}\partial_{NL}\mathcal{M}^{MQ}\mathcal{M}_{PQ}\mathcal{M}^{KL}$$

$$- \frac{1}{4}\partial_{MN}\mathcal{M}^{PK}\partial_{KL}\mathcal{M}^{MQ}\mathcal{M}_P{}^L\mathcal{M}_Q{}^N - \partial_{MK}\mathcal{M}^{NK}\partial_{NL}\mathcal{M}^{ML}$$

$$+ g^{-1}\partial_{MN}g\,\partial_{KL}\mathcal{M}^{MK}\mathcal{M}^{NL} + \frac{1}{4}\mathcal{M}^{MK}\mathcal{M}^{NL}g^{-2}\partial_{MN}g\,\partial_{KL}g$$

$$+ \frac{1}{4}\mathcal{M}^{MK}\mathcal{M}^{NL}\partial_{MN}g_{\mu\nu}\partial_{KL}g^{\mu\nu}.\tag{56}$$

Such defined, the Lagrangian (54) is invariant under local generalised internal diffeomorphisms, defined by their action on a vector $V^M$ of weight $\lambda$ as follows

$$\mathcal{L}^{SO(8,n)}_{(\Lambda,\Sigma)}V^M = \Lambda^{KL}\partial_{KL}V^M + 2\left(\partial^{KM}\Lambda_{KN} - \partial_{KN}\Lambda^{KM} + 2\Sigma^M{}_N\right)V^N + \lambda\,\partial_{KL}\Lambda^{KL}V^M.\tag{57}$$

---

[8]The global factor has been corrected compared to [39] by recovering the $SO(8, 8)$ theory as a truncation of the $E_{8(8)}$ ExFT reviewed in the following.

To make sure that these generalised diffeomorphisms close into an algebra, the gauge parameters $\Sigma_{MN}$ are subject to constraints similar to eq. (53),

$$\begin{cases} \Sigma_{[MN} \Sigma_{PQ]} = 0, \\ \eta^{NP} \Sigma_{MN} \Sigma_{PQ} = 0, \end{cases} \qquad \begin{cases} \Sigma_{[MN} \partial_{PQ]} = 0, \\ \eta^{NP} \Sigma_{MN} \partial_{PQ} = 0. \end{cases} \tag{58}$$

The associated covariant external derivatives used in eq. (54) are defined as

$$D_\mu = \partial_\mu - \mathcal{L}^{\mathrm{SO}(8,n)}_{(\mathcal{A}_\mu, \mathcal{B}_\mu)}, \tag{59}$$

with the weights of the fields in (52) and the gauge parameters $\Lambda^{MN}$ and $\Sigma_{MN}$ specified as

| | $g_{\mu\nu}$ | $\mathcal{M}_{MN}$ | $\mathcal{A}_\mu^{MN}$ | $\mathcal{B}_{\mu MN}$ | $\Lambda^M$ | $\Sigma_{MN}$ |
|---|---|---|---|---|---|---|
| $\lambda$ | 2 | 0 | 1 | 0 | 1 | 0 |

$$\tag{60}$$

To ensure the invariance of the action, the gauge fields $\mathcal{B}_\mu$ must also enter constraints analogous to (58).

The section constraints (53) for the SO$(8,n)$ theory admit two inequivalent solutions [39]. One corresponds to the $\mathcal{N} = (2,0)$ theory in six dimensions coupled to 5 self-dual and $n-3$ anti self-dual tensor fields and $5(n-3)$ scalars. Such a theory cannot be oxidised to more than six dimensions. For the alternate solution of (53), the theory (54) describes the NSNS sector of ten-dimensional supergravity coupled to $n-8$ ten-dimensional vectors. Setting $n = 8$ and denoting the physical internal coordinates as $y^i$ with $i \in [\![1,7]\!]$, the constraints (53) are solved by breaking

$$\begin{array}{ccc} \mathrm{SO}(8,8) & \supset & \mathrm{SO}(1,1) \times \mathrm{GL}(7), \\ X^M & \longrightarrow & \{X^0, X_0, X^i, X_i\}, \end{array} \tag{61}$$

and restricting coordinate dependence to $y^i = Y^{i0}$. The ExFT indices are aligned with the ones of the three-dimensional half-maximal theory by embedding $\mathrm{GL}(3) \times \mathrm{GL}(3) \times \mathrm{SO}(1,1) \subset \mathrm{GL}(7)$ as in (38). The explicit dictionary between the SO$(8,8)$-ExFT generalised metric and the internal components of the NSNS fields is given by [9]

$$\begin{aligned} \mathcal{M}^{00} &= \hat{g}^{-1} e^{\hat{\Phi}/2}, \\ \mathcal{M}^{0i} &= \tfrac{1}{6!} \mathcal{M}^{00} \varepsilon^{i j_1 \dots j_6} \tilde{b}_{j_1 \dots j_6}, \\ \mathcal{M}^{00} \mathcal{M}^{ij} - \mathcal{M}^{0i} \mathcal{M}^{0j} &= \hat{g}^{-1} \hat{g}^{ij}, \\ \mathcal{M}^{00} \mathcal{M}^i{}_j - \mathcal{M}^{0i} \mathcal{M}^0{}_j &= \hat{g}^{-1} \hat{g}^{ik} b_{kj}, \end{aligned} \tag{62}$$

where $\hat{g}_{ij}$ is the purely internal block of the ten-dimensional metric in Einstein frame, and $\hat{g}$ its determinant. The ExFT fields $b$ and $\tilde{b}$ do not directly embed into the ten-dimensional two-form, but determine its field strength $\hat{H} = d\hat{B}$ through

$$\hat{H} = db + e^{\hat{\Phi}/8} \star_{10} d\tilde{b}, \tag{63}$$

with the ten-dimensional Hodge star taken with respect to the Einstein-frame metric. To describe our configuration in the string frame, the only change needed is the usual rescaling of the metric $\hat{g}_{s\hat{\mu}\hat{\nu}} = e^{\hat{\Phi}/2} \hat{g}_{\hat{\mu}\hat{\nu}}$.

The consistent truncation of the NSNS sector of type II supergravity on $S^3 \times T^4$ and $S^3 \times \widetilde{S}^3 \times S^1$ down to a half-maximal supergravity can be described in terms of generalised Scherk-Schwarz Ansätze, where the dependence on external and internal coordinates factorises. The dependence

on the former is carried by the $D = 3$ fields and the latter by a group-valued twist matrix $U(Y)$ and a scale factor $\rho(Y)$ of weight $-1$. The precise factorisation reads [39]

$$
\begin{aligned}
g_{\mu\nu}(x, Y) &= \rho(Y)^{-2} g_{\mu\nu}(x), \\
\mathcal{M}_{MN}(x, Y) &= U_M{}^{\bar{M}}(Y) U_N{}^{\bar{N}}(Y) M_{\bar{M}\bar{N}}(x), \\
\mathcal{A}_\mu{}^{MN}(x, Y) &= \rho(Y)^{-1} (U^{-1})_{\bar{M}}{}^M(Y)(U^{-1})_{\bar{N}}{}^N(Y) A_\mu{}^{\bar{M}\bar{N}}(x), \\
\mathcal{B}_{\mu KL}(x, Y) &= -\frac{1}{4}\rho(Y)^{-1} U_{M\bar{N}}(Y)\partial_{KL}(U^{-1})_{\bar{M}}{}^M(Y) A_\mu{}^{\bar{M}\bar{N}}(x).
\end{aligned}
\tag{64}
$$

On the right-hand sides, $g_{\mu\nu}, M_{\bar{M}\bar{N}}$ and $A_\mu{}^{\bar{M}\bar{N}}$ are the fields of the half-maximal three-dimensional supergravity described in sec. 2.1. The truncation to these fields is consistent if

$$
\mathcal{L}^{\mathrm{SO}(8,n)}_{(\mathcal{U}_{\bar{K}\bar{L}}, \Sigma_{\bar{K}\bar{L}})}(U^{-1})_{\bar{M}}{}^M = 2\Theta_{\bar{K}\bar{L}|\bar{M}}{}^{\bar{N}}(U^{-1})_{\bar{N}}{}^M,
\tag{65}
$$

with

$$
\mathcal{U}_{\bar{K}\bar{L}}{}^{KL} = \rho^{-1}(U^{-1})_{[\bar{K}}{}^K (U^{-1})_{\bar{L}]}{}^L, \quad \text{and} \quad \Sigma_{\bar{K}\bar{L}, KL} = -\frac{1}{4}\rho^{-1}\partial_{KL}(U^{-1})_{[\bar{K}}{}^P U_{P\bar{L}]},
\tag{66}
$$

and a constant embedding tensor $\Theta_{\bar{K}\bar{L}|\bar{M}\bar{N}}$. This tensor specifies the explicit gauging and its components (8) can be expressed using the twist matrix and the scaling function as

$$
\begin{aligned}
\theta_{\bar{K}\bar{L}\bar{M}\bar{N}} &= -3\rho^{-1} J_{[\bar{K}\bar{L}, \bar{M}\bar{N}]}, \\
\theta_{\bar{M}\bar{N}} &= 2\rho^{-1} J_{\bar{K}(\bar{M}, \bar{N})}{}^{\bar{K}} - \eta_{\bar{M}\bar{N}}\theta + \xi_{\bar{M}\bar{N}}, \\
\theta &= -\frac{2}{8+n}\rho^{-1} J_{\bar{K}\bar{L}}{}^{\bar{K}\bar{L}},
\end{aligned}
\tag{67}
$$

with the SO(8,8) currents $J_{\bar{M}\bar{N}, \bar{K}}{}^{\bar{L}} = (U^{-1})_{\bar{M}}{}^M (U^{-1})_{\bar{N}}{}^N (U^{-1})_{\bar{K}}{}^K \partial_{MN} U_K{}^{\bar{L}}$ and the trombone gauging

$$
\xi_{\bar{M}\bar{N}} = 2\rho^{-2}(U^{-1})_{\bar{M}}{}^K (U^{-1})_{\bar{N}}{}^L \partial_{KL}\rho - 2\rho^{-1} J_{\bar{K}[\bar{M}, \bar{N}]}{}^{\bar{K}}.
\tag{68}
$$

In the following, all twist matrices will be such that $\xi_{\bar{M}\bar{N}} = 0$, allowing for a Lagrangian formulation of the three-dimensional supergravity. For the SO$(8,n)$ case with $n > 8$ relevant to heterotic supergravity, equations (64)–(68) generalise straightforwardly.

## 3.2 Review of $E_{8(8)}$ exceptional field theory

We can similarly employ an exceptional field theory suited to studying compactifications of maximal $10d$ supergravity (and $11d$ supergravity) down to $3d$. As detailed in sec. 2.2, the duality group is then $E_{8(8)}$. The $E_{8(8)}$-covariant reformulation of type IIB and $11d$ supergravities is $E_{8(8)}$ exceptional field theory [38]. Its structure is very similar to what we described in the previous section. The fields are

$$
\{g_{\mu\nu}, \mathcal{M}_{\mathcal{M}\mathcal{N}}, \mathcal{A}_\mu{}^{\mathcal{M}}, \mathcal{B}_{\mu\mathcal{M}}\},
\tag{69}
$$

alongside their fermionic superpartners. As before, they depend on both the external coordinates $x^\mu$ and on a set of 248 extended coordinates $Y^{\mathcal{M}}$. Here and in (69), the index $\mathcal{M} \in [\![1, 248]\!]$ is the adjoint index of $E_{8(8)}$. The dependence on the $Y^{\mathcal{M}}$ coordinates must be restricted by the section constraints

$$
\kappa^{\mathcal{M}\mathcal{N}}\partial_{\mathcal{M}} \otimes \partial_{\mathcal{N}} = 0,
\tag{70a}
$$

$$
f^{\mathcal{M}\mathcal{N}}{}_{\mathcal{P}}\partial_{\mathcal{M}} \otimes \partial_{\mathcal{N}} = 0,
\tag{70b}
$$

$$
(\mathbb{P}_{3875})_{\mathcal{M}\mathcal{N}}{}^{\mathcal{K}\mathcal{L}}\partial_{\mathcal{K}} \otimes \partial_{\mathcal{L}} = 0,
\tag{70c}
$$

which have two inequivalent solutions. One preserves seven physical coordinates and corresponds to type IIB supergravity, and the other has eight coordinates and is associated to M-theory. The $E_{8(8)}$ structure constants $f_{\mathcal{MN}}{}^{\mathcal{P}}$ and Cartan-Killing metric $\kappa^{\mathcal{MN}}$ can be respectively found in eq. (A.11) and (A.12) in appendix A. The components of the projector $\mathbf{248 \otimes 248 \mapsto 3875}$ can also be found in (A.21).

The theory is invariant under gauge symmetries generated by the $E_{8(8)}$ generalised Lie derivative. On a vector $V^{\mathcal{M}}$ of weight $\lambda$, it acts as

$$\mathcal{L}^{\mathrm{E}_{8(8)}}_{(\Lambda,\Sigma)} V^{\mathcal{M}} = \Lambda^{\mathcal{N}} \partial_{\mathcal{N}} V^{\mathcal{M}} - \Big(60\,(\mathbb{P}_{248})^{\mathcal{M}}{}_{\mathcal{N}}{}^{\mathcal{K}}{}_{\mathcal{L}} \partial_{\mathcal{K}} \Lambda^{\mathcal{L}} - f^{\mathcal{MK}}{}_{\mathcal{N}} \Sigma_{\mathcal{K}}\Big) V^{\mathcal{N}} + \lambda\, V^{\mathcal{M}} \partial_{\mathcal{N}} \Lambda^{\mathcal{N}}, \quad (71)$$

with $(\mathbb{P}_{248})^{\mathcal{M}}{}_{\mathcal{N}}{}^{\mathcal{K}}{}_{\mathcal{L}}$ in (A.21). As previously, the closure of the algebra of $\mathcal{L}^{\mathrm{E}_{8(8)}}_{(\Lambda,\Sigma)}$ imposes constraints on the gauge parameters $\Sigma_{\mathcal{M}}$ and $\mathcal{B}_{\mu\mathcal{M}}$ fields similar to eq. (70), and the fields in (69) need to be assigned weights analogously to the (60) assignment.

The bosonic Lagrangian, invariant under eq. (71), is given by [38]

$$\mathscr{L}^{\mathrm{E}_{8(8)}} = \sqrt{|g|}\Big(\widehat{R}^{\mathrm{E}_{8(8)}} + \frac{1}{240} D_{\mu} \mathcal{M}_{\mathcal{MN}} D^{\mu} \mathcal{M}^{\mathcal{MN}} + \mathscr{L}^{\mathrm{E}_{8(8)}}_{\mathrm{int}}\Big) + \mathscr{L}^{\mathrm{E}_{8(8)}}_{\mathrm{CS}}. \quad (72)$$

We denote $\widehat{R}^{\mathrm{E}_{8(8)}}$ the $E_{8(8)}$-covariantised Ricci scalar and define the $E_{8(8)}$-covariant derivative as[9]

$$D_{\mu} = \partial_{\mu} - \mathcal{L}^{\mathrm{E}_{8(8)}}_{(\mathcal{A}_{\mu},\mathcal{B}_{\mu})}. \quad (73)$$

The potential term $\mathscr{L}^{\mathrm{E}_{8(8)}}_{\mathrm{int}}$ reads

$$\begin{aligned}
\mathscr{L}^{\mathrm{E}_{8(8)}}_{\mathrm{int}} = {}& \frac{1}{240} \mathcal{M}^{\mathcal{MN}} \partial_{\mathcal{M}} \mathcal{M}^{\mathcal{KL}} \partial_{\mathcal{N}} \mathcal{M}_{\mathcal{KL}} - \frac{1}{2} \mathcal{M}^{\mathcal{MN}} \partial_{\mathcal{M}} \mathcal{M}^{\mathcal{KL}} \partial_{\mathcal{L}} \mathcal{M}_{\mathcal{NK}} \\
& - \frac{1}{7200} f^{\mathcal{NQ}}{}_{\mathcal{P}} f^{\mathcal{MS}}{}_{\mathcal{R}} \mathcal{M}^{\mathcal{PK}} \partial_{\mathcal{M}} \mathcal{M}_{\mathcal{QK}} \mathcal{M}^{\mathcal{RL}} \partial_{\mathcal{N}} \mathcal{M}_{\mathcal{SL}} \\
& + \frac{1}{2} g^{-1} \partial_{\mathcal{M}} g\, \partial_{\mathcal{N}} \mathcal{M}^{\mathcal{MN}} + \frac{1}{4} \mathcal{M}^{\mathcal{MN}} g^{-2} \partial_{\mathcal{M}} g\, \partial_{\mathcal{N}} g + \frac{1}{4} \mathcal{M}^{\mathcal{MN}} \partial_{\mathcal{M}} g_{\mu\nu} \partial_{\mathcal{N}} g^{\mu\nu}, \quad (74)
\end{aligned}$$

and the Chern-Simons term $\mathscr{L}^{\mathrm{E}_{8(8)}}_{\mathrm{CS}}$ has the following expression:

$$\begin{aligned}
\mathscr{L}^{\mathrm{E}_{8(8)}}_{\mathrm{CS}} = {}& \frac{1}{2} \varepsilon^{\mu\nu\rho} \Big( F_{\mu\nu}{}^{\mathcal{M}} \mathcal{B}_{\rho\,\mathcal{M}} - f_{\mathcal{KL}}{}^{\mathcal{N}} \partial_{\mu} \mathcal{A}_{\nu}{}^{\mathcal{K}} \partial_{\mathcal{N}} \mathcal{A}_{\rho}{}^{\mathcal{L}} - \frac{2}{3} f^{\mathcal{N}}{}_{\mathcal{KL}} \partial_{\mathcal{M}} \partial_{\mathcal{N}} \mathcal{A}_{\mu}{}^{\mathcal{K}} \mathcal{A}_{\nu}{}^{\mathcal{M}} \mathcal{A}_{\rho}{}^{\mathcal{L}} \\
& - \frac{1}{3} f_{\mathcal{MKL}} f^{\mathcal{KP}}{}_{\mathcal{Q}} f^{\mathcal{LR}}{}_{\mathcal{S}} \mathcal{A}_{\mu}{}^{\mathcal{M}} \partial_{\mathcal{P}} \mathcal{A}_{\nu}{}^{\mathcal{Q}} \partial_{\mathcal{R}} \mathcal{A}_{\rho}{}^{\mathcal{S}} \Big). \quad (75)
\end{aligned}$$

We refer to the eq. (2.26) of ref. [38] for the expression of the covariant field strength $F_{\mu\nu}{}^{\mathcal{M}}$ of $\mathcal{A}_{\mu}{}^{\mathcal{M}}$, which will not be needed in the following.

Within $E_{8(8)}$ exceptional field theory, the Scherk-Schwarz Ansatz describes consistent truncations of type II supergravity down to maximal $D = 3$ gauged supergravities. It is expressed in term of a twist matrix $U_{\mathcal{M}}{}^{\bar{\mathcal{M}}} \in E_{8(8)}$ and a scaling function $\rho$, and parallels eq. (64) [54,59]:

$$\begin{aligned}
g_{\mu\nu}(x,Y) &= \rho(Y)^{-2} g_{\mu\nu}(x), \\
\mathcal{M}_{\mathcal{MN}}(x,Y) &= U_{\mathcal{M}}{}^{\bar{\mathcal{M}}}(Y) U_{\mathcal{N}}{}^{\bar{\mathcal{N}}}(Y) M_{\bar{\mathcal{M}}\bar{\mathcal{N}}}(x), \\
\mathcal{A}_{\mu}{}^{\mathcal{M}}(x,Y) &= \rho(Y)^{-1} (U^{-1})_{\bar{\mathcal{M}}}{}^{\mathcal{M}}(Y) A_{\mu}{}^{\bar{\mathcal{M}}}(x), \\
\mathcal{B}_{\mu\mathcal{M}}(x,Y) &= \frac{\rho(Y)^{-1}}{60} f_{\bar{\mathcal{M}}}{}^{\bar{\mathcal{P}}\bar{\mathcal{Q}}} (U^{-1})_{\bar{\mathcal{P}}\mathcal{P}}(Y) \partial_{\mathcal{M}} (U^{-1})_{\bar{\mathcal{Q}}}{}^{\mathcal{P}}(Y) A_{\mu}{}^{\bar{\mathcal{M}}}(x).
\end{aligned} \quad (76)$$

---

[9]For the sake of readability, we use the same notation for the $SO(8,n)$ and $E_{8(8)}$ covariant derivatives in eq. (59) and (73).

The fields $g_{\mu\nu}$, $M_{\bar{\mathcal{M}}\bar{\mathcal{N}}}$ and $A_\mu{}^{\bar{\mathcal{M}}}$ now belong to the maximal three-dimensional supergravity described in sec. 2.2. The truncation to these fields is consistent if the following condition for generalised parallelisability is satisfied:

$$\mathcal{L}^{\mathrm{E}_{8(8)}}_{(\mathcal{U}_{\bar{\mathcal{M}}},\Sigma_{\bar{\mathcal{M}}})}\mathcal{U}_{\bar{\mathcal{N}}}{}^{\mathcal{M}} = X_{\bar{\mathcal{M}}\bar{\mathcal{N}}}{}^{\bar{\mathcal{P}}}\mathcal{U}_{\bar{\mathcal{P}}}{}^{\mathcal{M}}\,, \tag{77}$$

where

$$\mathcal{U}_{\bar{\mathcal{M}}}{}^{\mathcal{M}} = \rho^{-1}(U^{-1})_{\bar{\mathcal{M}}}{}^{\mathcal{M}}\,, \qquad \Sigma_{\bar{\mathcal{M}}\mathcal{M}} = \frac{1}{60}\rho^{-1}f_{\bar{\mathcal{M}}}{}^{\bar{\mathcal{P}}\bar{\mathcal{Q}}}(U^{-1})_{\bar{\mathcal{P}}\mathcal{P}}\partial_{\mathcal{M}}(U^{-1})_{\bar{\mathcal{Q}}}{}^{\mathcal{P}}\,, \tag{78}$$

and with constant torsion [19]

$$\begin{aligned} X_{\bar{\mathcal{M}}\bar{\mathcal{N}}}{}^{\bar{\mathcal{P}}} &= -\rho^{-1}\mathcal{J}_{\bar{\mathcal{M}}\bar{\mathcal{N}}}{}^{\bar{\mathcal{P}}} + \rho^{-1}f^{\bar{\mathcal{P}}}{}_{\bar{\mathcal{N}}\bar{\mathcal{Q}}}f^{\bar{\mathcal{Q}}\bar{\mathcal{K}}}{}_{\bar{\mathcal{L}}}\mathcal{J}_{\bar{\mathcal{K}}\bar{\mathcal{M}}}{}^{\bar{\mathcal{L}}} - \frac{1}{60}\rho^{-1}f^{\bar{\mathcal{P}}\bar{\mathcal{K}}}{}_{\bar{\mathcal{N}}}f_{\bar{\mathcal{M}}\bar{\mathcal{L}}}{}^{\bar{\mathcal{Q}}}\mathcal{J}_{\bar{\mathcal{K}}\bar{\mathcal{Q}}}{}^{\bar{\mathcal{L}}} \\ &\quad -\frac{1}{2}\rho^{-1}f^{\bar{\mathcal{P}}}{}_{\bar{\mathcal{N}}\bar{\mathcal{Q}}}f^{\bar{\mathcal{Q}}\bar{\mathcal{K}}}{}_{\bar{\mathcal{M}}}\mathcal{J}_{\bar{\mathcal{R}}\bar{\mathcal{K}}}{}^{\bar{\mathcal{R}}} + \left(\delta_{\bar{\mathcal{M}}}{}^{\bar{\mathcal{K}}}\delta_{\bar{\mathcal{N}}}{}^{\bar{\mathcal{P}}} - \frac{1}{2}f_{\bar{\mathcal{M}}}{}^{\bar{\mathcal{L}}\bar{\mathcal{K}}}f_{\bar{\mathcal{N}}\bar{\mathcal{L}}}{}^{\bar{\mathcal{P}}}\right)\xi_{\bar{\mathcal{K}}}\,, \end{aligned} \tag{79}$$

which can be identified with the embedding tensor of the three-dimensional gauged supergravity. Here we have introduced the $\mathrm{E}_{8(8)}$ current $\mathcal{J}_{\bar{\mathcal{M}}\bar{\mathcal{N}}}{}^{\bar{\mathcal{P}}} = (U^{-1})_{\bar{\mathcal{M}}}{}^{\mathcal{K}}(U^{-1})_{\bar{\mathcal{N}}}{}^{\mathcal{L}}\partial_{\mathcal{K}}U_{\mathcal{L}}{}^{\bar{\mathcal{P}}}$ and the trombone gauging

$$\xi_{\bar{\mathcal{M}}} = 2(U^{-1})_{\bar{\mathcal{M}}}{}^{\mathcal{N}}\partial_{\mathcal{N}}\rho^{-1} + \rho^{-1}\partial_{\mathcal{N}}(U^{-1})_{\bar{\mathcal{M}}}{}^{\mathcal{N}}\,. \tag{80}$$

As before, we will always consider $\xi_{\bar{\mathcal{M}}} = 0$. This consistency condition is most nicely expressed once projected on the adjoint representation

$$X_{\bar{\mathcal{M}}\bar{\mathcal{N}}} = -2\rho^{-1}\mathcal{J}_{(\bar{\mathcal{M}}\bar{\mathcal{N}})} - \rho^{-1}\mathcal{J}_{\bar{\mathcal{K}}(\bar{\mathcal{M}}}{}^{\bar{\mathcal{L}}}f_{\bar{\mathcal{N}})\bar{\mathcal{L}}}{}^{\bar{\mathcal{K}}}\,, \tag{81}$$

with

$$X_{\bar{\mathcal{M}}\bar{\mathcal{N}}} = \tfrac{1}{60}X_{\bar{\mathcal{M}}\bar{\mathcal{P}}\bar{\mathcal{Q}}}f_{\bar{\mathcal{N}}}{}^{\bar{\mathcal{P}}\bar{\mathcal{Q}}}\,, \qquad \text{and} \qquad \mathcal{J}_{\bar{\mathcal{M}}\bar{\mathcal{N}}} = \tfrac{1}{60}\mathcal{J}_{\bar{\mathcal{M}}\bar{\mathcal{P}}\bar{\mathcal{Q}}}f_{\bar{\mathcal{N}}}{}^{\bar{\mathcal{P}}\bar{\mathcal{Q}}}\,. \tag{82}$$

### 3.3 ExFT matryoshka

We embed the $SO(8,8)$ exceptional field theory into its $\mathrm{E}_{8(8)}$ counterpart by breaking the latter group as in eq. (26):

$$\begin{array}{ccc} \mathrm{E}_{8(8)} & \longrightarrow & SO(8,8)\,, \\ X^{\mathcal{M}} & \longrightarrow & \left\{X^{[MN]}, X^{\mathcal{A}}\right\}. \end{array} \tag{83}$$

The $SO(8,8)$-ExFT coordinates $Y^{MN}$ of sec. 3.1 are identified with the components in the **120** of the $\mathrm{E}_{8(8)}$ coordinates $Y^{\mathcal{M}}$, and all fields and parameters are independent of $Y^{\mathcal{A}}$,

$$Y^{MN} \subset Y^{\mathcal{M}}\,, \quad \text{and} \quad \partial_{\mathcal{A}} = 0\,. \tag{84}$$

The fields of the two theories can also be related through (83). The relevant sigma models are identified through the inclusion (19), and the vectors in the adjoint of $SO(8,8)$ are identified in the two theories:

$$\mathcal{A}_\mu^{\mathrm{E}_{8(8)}MN} = 2\mathcal{A}_\mu^{SO(8,8)MN}\,, \quad \text{and} \quad \mathcal{B}_{\mu MN}^{\mathrm{E}_{8(8)}} = 4\mathcal{B}_{\mu MN}^{SO(8,8)}\,. \tag{85}$$

The remaining components are identified with the Ramond-Ramond fields of maximal supergravity. From an $SO(8,8)$ perspective, the consistency of the truncation to the NSNS sector follows from the projection in (32).

In the following, we describe how the $\mathrm{E}_{8(8)}$ section constraints and generalised Lie derivatives are related to their $SO(8,8)$ counterparts. For configurations that admit a generalised Leibniz parallelisation in the $SO(8,8)$ theory, we detail how to build a twist matrix $U_{\mathcal{M}}{}^{\bar{\mathcal{M}}}$ from $U_M{}^{\bar{M}}$ in such a way that the embedding tensors in the corresponding consistent truncations are related as in (28).

**Section constraints** For the adjoint coordinate dependence (84), the $E_{8(8)}$ section conditions (70) follow from the SO(8,8) ones (53). This can be seen explicitly using the SO(8,8) decomposition of the $E_{8(8)}$ structure constants given in (A.11). For the conditions (70a) and (70b), the non-trivial components are

$$\kappa^{\mathcal{MN}}\partial_{\mathcal{M}}\otimes\partial_{\mathcal{N}}=-\frac{1}{8}\,\partial_{MN}\otimes\partial^{MN}\,,$$
$$f^{MN,PQ}{}_{RS}\,\partial_{MN}\otimes\partial_{PQ}=-4\,\partial_{[R}{}^{T}\otimes\partial_{S]T}\,,\tag{86}$$

which vanish as a consequence of eq. (53b). Concerning the last condition (70c), let us first note that it is equivalent to eq. (70a) and (70b) together with

$$f^{\mathcal{M}}{}_{\mathcal{KR}}f^{\mathcal{N}}{}_{\mathcal{L}}{}^{\mathcal{R}}\,\partial_{\mathcal{M}}\otimes\partial_{\mathcal{N}}-2\,\partial_{(\mathcal{K}}\otimes\partial_{\mathcal{L})}=0\,.\tag{87}$$

The only non trivial components of this equation are

$$f^{\mathcal{M}}{}_{MN,\mathcal{R}}f^{\mathcal{N}}{}_{PQ}{}^{\mathcal{R}}\,\partial_{\mathcal{M}}\otimes\partial_{\mathcal{N}}-\partial_{MN}\otimes\partial_{PQ}-\partial_{PQ}\otimes\partial_{MN}=-6\,\partial_{[MN}\otimes\partial_{PQ]}\,,$$
$$f^{\mathcal{M}}{}_{\mathcal{AR}}f^{\mathcal{N}}{}_{\mathcal{B}}{}^{\mathcal{R}}\,\partial_{\mathcal{M}}\otimes\partial_{\mathcal{N}}=-\frac{1}{16}\left(\Gamma^{IJ}\Gamma^{KL}\right)_{\mathcal{AB}}\partial_{IJ}\otimes\partial_{KL}\,.\tag{88}$$

They both vanish thanks to the SO(8,8) section condition (53) and

$$\left(\Gamma^{MN}\Gamma^{PQ}\right)_{\mathcal{AB}}=\Gamma^{MNPQ}_{\mathcal{AB}}+2\,\eta^{M[P}\Gamma^{Q]N}_{\mathcal{AB}}-2\,\eta^{N[P}\Gamma^{Q]M}_{\mathcal{AB}}-2\,\eta^{M[P}\eta^{Q]N}\,\eta_{\mathcal{AB}}\,.\tag{89}$$

For the solution of the section constraint in (61), the dictionary between the $E_{8(8)}$-ExFT generalised metric and the internal components of the NSNS fields is obtained by further splitting SO(8,8) under SO(1,1) × GL(7) and using eq. (62). The internal components of the RR fluxes could be computed similarly through the components of the $E_{8(8)}$-ExFT generalised metric in the **128** of SO(8,8). However, as the deformations of the $AdS_3 \times S^3 \times T^4$ and $AdS_3 \times S^3 \times \widetilde{S}^3 \times S^1$ solutions we consider do not excite those fluxes, this part of the dictionary will not be needed in here.

**Generalised Lie derivative** With the coordinates (84), the $E_{8(8)}$ generalised Lie derivative (71) decomposes as

$$\mathcal{L}^{E_{8(8)}}_{(\Lambda,\Sigma)}V^{MN}=\mathcal{L}^{SO(8,8)}_{(\hat{\Lambda},\hat{\Sigma})}V^{MN}+\frac{1}{8}\left(\Gamma^{MN}\Gamma^{KL}\right)_{\mathcal{AB}}V^{\mathcal{A}}\partial_{KL}\Lambda^{\mathcal{B}}-\frac{1}{2}\left(\Gamma^{MN}\right)^{\mathcal{A}}{}_{\mathcal{B}}\Sigma_{\mathcal{A}}V^{\mathcal{B}}\,,$$
$$\mathcal{L}^{E_{8(8)}}_{(\Lambda,\Sigma)}V^{\mathcal{A}}=\mathcal{L}^{SO(8,8)}_{(\hat{\Lambda},\hat{\Sigma})}V^{\mathcal{A}}-\frac{1}{2}\left(\Gamma^{PQ}\right)^{\mathcal{A}}{}_{\mathcal{B}}V^{\mathcal{B}}\partial_{KQ}\Lambda^{K}{}_{P}+\frac{1}{16}\left(\Gamma^{PQ}\Gamma^{KL}\right)^{\mathcal{A}}{}_{\mathcal{B}}V_{PQ}\partial_{KL}\Lambda^{\mathcal{B}}\tag{90}$$
$$+\frac{1}{4}\left(\Gamma^{KL}\right)^{\mathcal{AB}}\left(\Sigma_{\mathcal{B}}V_{KL}+\Sigma_{KL}V_{\mathcal{B}}\right)\,,$$

where $(\hat{\Lambda}^{MN},\hat{\Sigma}_{MN})=(\frac{1}{2}\Lambda^{MN},\frac{1}{4}\Sigma_{MN})$, in accordance with eq. (85), and $V^{\mathcal{A}}$ is considered a set of SO(8,8) scalars. Restricting all the objects to have vanishing components in the spinorial representation of the orthogonal group, the generalised Lie derivative of the $E_{8(8)}$ theory can be observed to reduce to the one for the SO(8,8) ExFT.

**Uplift** An $E_{8(8)}$ twist matrix satisfying the consistency condition (77) can be constructed from an SO(8,8) twist matrix satisfying the condition (65). We identify the scale factors $\rho$ and define[10]

$$U_{\mathcal{M}}{}^{\bar{\mathcal{M}}}=\begin{pmatrix} 2\,U_{[M}{}^{\bar{M}}U_{N]}{}^{\bar{N}} & 0 \\ 0 & U_{\mathcal{A}}{}^{\bar{\mathcal{A}}} \end{pmatrix}\,,\tag{91}$$

---

[10]The coefficients in eq. (91) are different from those in ref. [9] to match the summing convention (25).

where $U_{\mathcal{A}}{}^{\bar{\mathcal{A}}}$ is a $\mathbf{128}_s$ representation of $U_M{}^{\bar{M}}$,

$$U_{\mathcal{A}}{}^{\bar{\mathcal{A}}} = \exp\left(\frac{1}{2} u_{MN} \Gamma^{MN}\right)_{\mathcal{A}}{}^{\bar{\mathcal{A}}}, \tag{92}$$

where the matrix $u$ is such that $U_M{}^{\bar{M}} = \exp\left(u_{PQ} T^{PQ}\right)_M{}^{\bar{M}}$, with $T^{\bar{M}\bar{N}}$ the generators of $\mathfrak{so}(8,8)$ normalised as in eq. (5). Then, using the decomposition (90) of the generalised Lie derivative, the $E_{8(8)}$ generalised parallelisability condition (77) has the following non-vanishing components:

$$
\begin{aligned}
\mathcal{L}^{E_{8(8)}}_{(\mathcal{U}_{\bar{M}\bar{N}}, \Sigma_{\bar{M}\bar{N}})} \mathcal{U}_{\bar{K}\bar{L}}{}^{MN} &= 4\,\Theta_{\bar{M}\bar{N}|[\bar{K}}{}^{[\bar{P}} \delta_{\bar{L}]}{}^{\bar{Q}]} \mathcal{U}_{\bar{P}\bar{Q}}{}^{MN}, \\
\mathcal{L}^{E_{8(8)}}_{(\mathcal{U}_{\bar{M}\bar{N}}, \Sigma_{\bar{M}\bar{N}})} \mathcal{U}_{\bar{\mathcal{A}}}{}^{\mathcal{A}} &= \frac{1}{2}\,\Theta_{\bar{M}\bar{N}|\bar{K}\bar{L}} \left(\Gamma^{\bar{K}\bar{L}}\right)_{\bar{\mathcal{A}}}{}^{\bar{\mathcal{B}}} \mathcal{U}_{\bar{\mathcal{B}}}{}^{\mathcal{A}}, \\
\mathcal{L}^{E_{8(8)}}_{(\mathcal{U}_{\bar{\mathcal{A}}}, \Sigma_{\bar{\mathcal{A}}})} \mathcal{U}_{\bar{\mathcal{B}}}{}^{MN} &= \frac{1}{4}\left(-\theta\,\eta_{\bar{\mathcal{A}}\bar{\mathcal{C}}} + \frac{1}{48} \Gamma^{\bar{P}\bar{Q}\bar{R}\bar{S}}{}_{\bar{\mathcal{A}}\bar{\mathcal{C}}}\, \theta_{\bar{P}\bar{Q}\bar{R}\bar{S}}\right) \left(\Gamma^{\bar{M}\bar{N}}\right)^{\bar{\mathcal{C}}}{}_{\bar{\mathcal{B}}} \mathcal{U}_{\bar{M}\bar{N}}{}^{MN}, \\
\mathcal{L}^{E_{8(8)}}_{(\mathcal{U}_{\bar{\mathcal{A}}}, \Sigma_{\bar{\mathcal{A}}})} \mathcal{U}_{\bar{M}\bar{N}}{}^{\mathcal{A}} &= \frac{1}{2}\left(-\theta\,\eta_{\bar{\mathcal{A}}\bar{\mathcal{B}}} + \frac{1}{48} \Gamma^{\bar{P}\bar{Q}\bar{R}\bar{S}}{}_{\bar{\mathcal{A}}\bar{\mathcal{B}}}\, \theta_{\bar{P}\bar{Q}\bar{R}\bar{S}}\right) \left(\Gamma_{\bar{M}\bar{N}}\right)^{\bar{\mathcal{B}}\bar{\mathcal{C}}} \mathcal{U}_{\bar{\mathcal{C}}}{}^{\mathcal{A}},
\end{aligned}
\tag{93}
$$

where we used SO(8,8) consistency equation (67). Hence, the consistency of the $E_{8(8)}$ Ansatz (76) is ensured by the one of the SO(8,8) Ansatz (64). The components of the resulting $E_{8(8)}$ embedding tensor read

$$X_{\bar{M}\bar{N},\bar{P}\bar{Q}} = 2\,\Theta_{\bar{M}\bar{N},\bar{P}\bar{Q}}, \qquad X_{\bar{\mathcal{A}}\bar{\mathcal{B}}} = -\theta\,\eta_{\bar{\mathcal{A}}\bar{\mathcal{B}}} + \frac{1}{48} \Gamma^{\bar{M}\bar{N}\bar{P}\bar{Q}}{}_{\bar{\mathcal{A}}\bar{\mathcal{B}}}\, \theta_{\bar{M}\bar{N}\bar{P}\bar{Q}}, \qquad X_{\bar{M}\bar{N},\bar{\mathcal{A}}} = 0. \tag{94}$$

The relation between the embedding tensors reproduces the three-dimensional embedding tensor (28). Thus, a twist matrix $U_M{}^{\bar{M}} \in SO(8,8)$ and a scale factor $\rho$ satisfying the consistency condition (65) will both give a consistent truncation of half-maximal ten-dimensional supergravity down to $\mathcal{N} = 8$ three-dimensional supergravity through eq. (64) and a consistent truncation of IIB supergravity down to $\mathcal{N} = 16$ supergravity in $3d$ through eq. (76) and (91). In sec. 4, we describe the pairs $(\rho, U_M{}^{\bar{M}})$ suited to the reductions on $S^3 \times \tilde{S}^3 \times S^1$ and $S^3 \times T^4$.

## 3.4 Kaluza-Klein spectroscopy

On Leibniz parallelisable solutions of exceptional field theory, the Kaluza-Klein spectrum can be obtained by extending the Scherk-Schwarz factorisations in (64) and (76) to include the linearised perturbations. These linear perturbations have a natural tower structure when expanded in terms of the harmonics of the most symmetric configuration homeomorphic to the relevant background [32, 33]. In fact, only the scalar harmonics are needed and the levels are not mixed by the mass operators, a feature that turns the computation of the Kaluza-Klein masses into a diagonalisation problem for a set of mass matrices. In the following we will discuss how to compute the Kaluza-Klein spectrum on any solution that uplifts from $3d$ supergravity using these ExFT techniques.

### 3.4.1 SO(8,n) mass matrices

For the modes arising from the $10d$ metric, dilaton, Kalb-Ramond field, and possibly extra ten-dimensional vector multiplets, it suffices to extend the Scherk-Schwarz Ansatz (64) in analogy with [36]. Starting from a background specified by three-dimensional SO(8,n)-supergravity fields

$$\{g_{\mu\nu},\, M_{\bar{M}\bar{N}},\, A_\mu{}^{\bar{M}\bar{N}}\} = \{\bar{g}_{\mu\nu},\, \Delta_{\bar{M}\bar{N}},\, 0\}, \tag{95}$$

we consider the expansion

$$
g_{\mu\nu}(x,Y) = \rho(Y)^{-2}\big(\bar{g}_{\mu\nu}(x) + h_{\mu\nu}{}^{\mathbf{\Lambda}}(x)\mathcal{Y}^{\mathbf{\Lambda}}(Y)\big),
$$
$$
\mathcal{M}_{MN}(x,Y) = U_M{}^{\bar{M}}(Y)U_N{}^{\bar{N}}(Y)\big(\Delta_{\bar{M}\bar{N}} + j_{\bar{M}\bar{N}}{}^{\mathbf{\Lambda}}(x)\mathcal{Y}^{\mathbf{\Lambda}}(Y)\big),
$$
$$
\mathcal{A}_\mu{}^{MN}(x,Y) = \rho(Y)^{-1}(U^{-1})_{\bar{M}}{}^M(Y)(U^{-1})_{\bar{N}}{}^N(Y)A_\mu{}^{\bar{M}\bar{N}\,\mathbf{\Lambda}}(x)\mathcal{Y}^{\mathbf{\Lambda}}(Y),
$$
$$
\mathcal{B}_{\mu KL}(x,Y) = -\frac{1}{4}\rho(Y)^{-1}U_{M\bar{N}}(Y)\partial_{KL}(U^{-1})_{\bar{M}}{}^M(Y)A_\mu{}^{\bar{M}\bar{N}\,\mathbf{\Lambda}}(x)\mathcal{Y}^{\mathbf{\Lambda}}(Y),
$$

(96)

extending (64). Here, $\mathbf{\Lambda}$ denotes a possibly composite Kaluza-Klein index which will depend on the topology of the background solution. These harmonics lead to the definition of $\mathring{\mathcal{T}}_{\bar{M}\bar{N}}{}^{\mathbf{\Lambda}\mathbf{\Sigma}}$ as the constant representation matrix encoded in the $SO(8,n)$ twist matrix as

$$
\rho^{-1}(U^{-1})_{\bar{M}}{}^M(U^{-1})_{\bar{N}}{}^N\partial_{MN}\mathcal{Y}^{\mathbf{\Lambda}} = -2\,\mathring{\mathcal{T}}_{\bar{M}\bar{N}}{}^{\mathbf{\Lambda}\mathbf{\Sigma}}\mathcal{Y}^{\mathbf{\Sigma}}.
$$

(97)

The properties of the twist matrix (67) guarantee that the $\mathring{\mathcal{T}}_{\bar{M}\bar{N}}{}^{\mathbf{\Lambda}\mathbf{\Sigma}}$ represent the gauge algebra, with the commutator normalised as [36]

$$
\big[\mathring{\mathcal{T}}_{\bar{M}\bar{N}},\mathring{\mathcal{T}}_{\bar{P}\bar{Q}}\big] = -\Theta_{\bar{M}\bar{N}|[\bar{P}}{}^{\bar{K}}\,\mathring{\mathcal{T}}_{\bar{Q}]\bar{K}} + \Theta_{\bar{P}\bar{Q}|[\bar{M}}{}^{\bar{K}}\,\mathring{\mathcal{T}}_{\bar{N}]\bar{K}}.
$$

(98)

To describe backgrounds corresponding to other points of the scalar manifold, it is convenient to dress this tensor analogously to eq. (14),

$$
\mathcal{T}_{\bar{M}\bar{N}} = (\mathcal{V}^{-1})_{\bar{M}}{}^{\bar{K}}(\mathcal{V}^{-1})_{\bar{N}}{}^{\bar{L}}\,\mathring{\mathcal{T}}_{\bar{K}\bar{L}}.
$$

(99)

Then, the Kaluza-Klein mass matrices are those presented in [36,41], which we reproduce here in the present notation. The mass matrices corresponding to the bosonic Kaluza-Klein modes read

$$
M_{(2)}^2{}^{\mathbf{\Sigma}\mathbf{\Omega}} = -2\,\delta^{\bar{M}\bar{P}}\delta^{\bar{N}\bar{Q}}\mathcal{T}_{\bar{M}\bar{N}}{}^{\mathbf{\Sigma}\mathbf{\Gamma}}\mathcal{T}_{\bar{P}\bar{Q}}{}^{\mathbf{\Gamma}\mathbf{\Omega}},
$$

(100a)

$$
M_{(1)}{}^{\bar{M}\bar{N}}{}_{\bar{P}\bar{Q}}{}^{\mathbf{\Sigma}\mathbf{\Omega}} = \big(\eta^{\bar{K}[\bar{M}}\eta^{\bar{N}]\bar{L}} - \delta^{\bar{K}[\bar{M}}\delta^{\bar{N}]\bar{L}}\big)\big(T_{\bar{K}\bar{L}|\bar{P}\bar{Q}}\delta^{\mathbf{\Sigma}\mathbf{\Omega}} + 4\,\mathcal{T}_{\bar{K}[\bar{P}}{}^{\mathbf{\Sigma}\mathbf{\Omega}}\eta_{\bar{Q}]\bar{L}}\big),
$$

(100b)

$$
M_{(0)\,\bar{M}\bar{N},\bar{P}\bar{Q}}^2{}^{\mathbf{\Sigma}\mathbf{\Omega}}j^{\bar{M}\bar{N},\mathbf{\Sigma}}j^{\bar{P}\bar{Q},\mathbf{\Omega}} = \big(m_{\bar{M}\bar{N},\bar{P}\bar{Q}}\delta^{\mathbf{\Sigma}\mathbf{\Omega}} + m'_{\bar{M}\bar{N},\bar{P}\bar{Q}}{}^{\mathbf{\Sigma}\mathbf{\Omega}}\big)j^{\bar{M}\bar{N},\mathbf{\Sigma}}j^{\bar{P}\bar{Q},\mathbf{\Omega}},
$$

(100c)

where $m_{\bar{M}\bar{N},\bar{P}\bar{Q}}$ is given in eq. (16) and

$$
\begin{aligned}
m'_{\bar{M}\bar{N},\bar{P}\bar{Q}}{}^{\mathbf{\Sigma}\mathbf{\Omega}} =\ & 8\,T_{\bar{M}\bar{P}\bar{R}\bar{K}}\,\delta_{\bar{N}}{}^{\bar{R}}\delta^{\bar{K}\bar{L}}\,\mathcal{T}_{\bar{Q}\bar{L}}{}^{\mathbf{\Sigma}\mathbf{\Omega}} + 8\,T_{\bar{M}\bar{P}\bar{R}\bar{K}}\,\delta_{\bar{Q}}{}^{\bar{R}}\delta^{\bar{K}\bar{L}}\,\mathcal{T}_{\bar{N}\bar{L}}{}^{\mathbf{\Sigma}\mathbf{\Omega}} \\
& - 8\,\eta_{\bar{M}\bar{P}}\,T_{\bar{N}\bar{Q}\bar{K}\bar{L}}\,\delta^{\bar{K}\bar{R}}\delta^{\bar{L}\bar{S}}\,\mathcal{T}_{\bar{R}\bar{S}}{}^{\mathbf{\Sigma}\mathbf{\Omega}} + 8\,\eta_{\bar{M}\bar{P}}\,T_{\bar{N}\bar{Q}\bar{K}\bar{L}}\,\mathcal{T}^{\bar{K}\bar{L}\,\mathbf{\Sigma}\mathbf{\Omega}} \\
& + 8\,\big(T_{\bar{M}\bar{P}} + T\,\eta_{\bar{M}\bar{P}}\big)\,\mathcal{T}_{\bar{N}\bar{Q}}{}^{\mathbf{\Sigma}\mathbf{\Omega}} + 2\,\eta_{\bar{M}\bar{P}}\,\eta_{\bar{N}\bar{Q}}\,\delta^{\bar{K}\bar{R}}\delta^{\bar{L}\bar{S}}\,\mathcal{T}_{\bar{K}\bar{L}}{}^{\mathbf{\Sigma}\mathbf{\Lambda}}\mathcal{T}_{\bar{R}\bar{S}}{}^{\mathbf{\Lambda}\mathbf{\Omega}} \\
& + 16\,\delta_{\bar{M}\bar{P}}\,\delta^{\bar{K}\bar{L}}\,\mathcal{T}_{\bar{Q}\bar{L}}{}^{\mathbf{\Sigma}\mathbf{\Lambda}}\mathcal{T}_{\bar{N}\bar{K}}{}^{\mathbf{\Lambda}\mathbf{\Omega}} - 4\,\delta_{\bar{M}}{}^{\bar{K}}\delta_{\bar{P}}{}^{\bar{L}}\,\mathcal{T}_{\bar{Q}\bar{L}}{}^{\mathbf{\Sigma}\mathbf{\Lambda}}\mathcal{T}_{\bar{N}\bar{K}}{}^{\mathbf{\Lambda}\mathbf{\Omega}} \\
& + 16\,\mathcal{T}_{\bar{M}\bar{P}}{}^{\mathbf{\Sigma}\mathbf{\Lambda}}\mathcal{T}_{\bar{N}\bar{Q}}{}^{\mathbf{\Lambda}\mathbf{\Omega}}.
\end{aligned}
$$

(101)

In turn, the mass matrices for the fermionic fields are[11]

$$
M_{(3/2)}{}^{\bar{A}\bar{B},\mathbf{\Lambda}\mathbf{\Sigma}} = -A_1^{\bar{A}\bar{B}}\,\delta^{\mathbf{\Lambda}\mathbf{\Sigma}} + 2\,\gamma^{\bar{I}\bar{J}}{}_{\bar{A}\bar{B}}\,\mathcal{T}_{\bar{I}\bar{J}}{}^{\mathbf{\Lambda}\mathbf{\Sigma}},
$$

(102a)

$$
M_{(1/2)}{}^{\bar{A}\hat{\bar{I}}\bar{B}\hat{\bar{J}},\mathbf{\Lambda}\mathbf{\Sigma}} = -A_3^{\bar{A}\hat{\bar{I}}\bar{B}\hat{\bar{J}}}\,\delta^{\mathbf{\Lambda}\mathbf{\Sigma}} - 2\,\gamma^{\bar{I}\bar{J}}{}_{\bar{A}\bar{B}}\,\delta_{\hat{\bar{I}}\hat{\bar{J}}}\,\mathcal{T}_{\bar{I}\bar{J}}{}^{\mathbf{\Lambda}\mathbf{\Sigma}} + 8\,\delta_{\bar{A}\bar{B}}\,\mathcal{T}_{\hat{\bar{I}}\hat{\bar{J}}}{}^{\mathbf{\Lambda}\mathbf{\Sigma}},
$$

(102b)

in terms of the shift tensors in (17) and the $SO(8)\times SO(8)$ components of $\mathcal{T}_{\bar{M}\bar{N}}{}^{\mathbf{\Lambda}\mathbf{\Sigma}}$.

---

[11]This corrects a sign in eq. (4.13) of [41].

As in previously studied 3$d$ Kaluza-Klein spectra [36,41], all the eigenvalues of the graviton and gravitino mass matrices correspond to physical modes in the spectrum (on the proviso remarked in footnote 7), and one must take into account that each of the eigenvalues of (100a) in fact corresponds to two states of opposite spin. The eigenvalues of the remaining matrices include the Goldstone modes which are absorbed by massive gravitons, gravitini and vectors in the super-BEH mechanism upon taking into account the off-diagonal couplings between modes of different spin. Ignoring these couplings, the eigenvalues to be discarded can be identified given the masses of the gravitons and gravitini [41,60]. The relevant relations in $D = 3$ are [41]

$$\left(m_{(1)}\ell_{\text{AdS}}\right)_{\text{Goldstone}} = \pm 2\sqrt{1+\left(m_{(2)}\ell_{\text{AdS}}\right)^2}, \qquad \left(m_{(1/2)}\ell_{\text{AdS}}\right)_{\text{goldstino}} = 3\,m_{(3/2)}\ell_{\text{AdS}}, \tag{103}$$

for goldstinos and Goldstone vectors. Out of the eigenvalues of the scalar mass matrix (100c), one must also remove the usual massless fields corresponding to longitudinal polarisations of massive vectors, as well as two values for every massive graviton. One of them is always zero and the other is given by

$$\left(m_{(0)}\ell_{\text{AdS}}\right)^2_{\text{Goldstone}} = -3\left(m_{(2)}\ell_{\text{AdS}}\right)^2. \tag{104}$$

### 3.4.2 $\mathrm{E}_{8(8)}$ mass matrices

For the spectrum of the full type II supergravity, we need to consider a deformation of (76) for $\mathrm{E}_{8(8)}$ ExFT. Around the background specified by 3$d$ fields

$$\{g_{\mu\nu},\, M_{\bar{\mathcal{M}}\bar{\mathcal{N}}},\, A_\mu^{\bar{\mathcal{M}}}\} = \{\bar{g}_{\mu\nu},\, \Delta_{\bar{\mathcal{M}}\bar{\mathcal{N}}},\, 0\}, \tag{105}$$

the fluctuation Ansatz is [9]

$$g_{\mu\nu}(x,Y) = \rho(Y)^{-2}\left(\bar{g}_{\mu\nu}(x) + h_{\mu\nu}{}^{\boldsymbol{\Sigma}}(x)\,\mathcal{Y}^{\boldsymbol{\Sigma}}(Y)\right),$$
$$\mathcal{M}_{\mathcal{M}\mathcal{N}}(x,Y) = U_{\mathcal{M}}{}^{\bar{\mathcal{M}}}(Y)U_{\mathcal{N}}{}^{\bar{\mathcal{N}}}(Y)\left(\Delta_{\bar{\mathcal{M}}\bar{\mathcal{N}}} + j_{\bar{\mathcal{M}}\bar{\mathcal{N}}}{}^{\boldsymbol{\Sigma}}(x)\,\mathcal{Y}^{\boldsymbol{\Sigma}}(Y)\right),$$
$$A_\mu{}^{\mathcal{M}}(x,Y) = \rho(Y)^{-1}(U^{-1})_{\bar{\mathcal{M}}}{}^{\mathcal{M}}(Y)A_\mu{}^{\bar{\mathcal{M}},\boldsymbol{\Sigma}}(x)\,\mathcal{Y}^{\boldsymbol{\Sigma}}(Y),$$
$$\mathcal{B}_{\mu\mathcal{M}}(x,Y) = \frac{\rho(Y)^{-1}}{60}f_{\bar{\mathcal{M}}}{}^{\bar{\mathcal{P}}\bar{\mathcal{Q}}}(U^{-1})_{\bar{\mathcal{P}}\mathcal{P}}(Y)\partial_{\mathcal{M}}(U^{-1})_{\bar{\mathcal{Q}}}{}^{\mathcal{P}}(Y)A_\mu{}^{\bar{\mathcal{M}},\boldsymbol{\Sigma}}(x)\,\mathcal{Y}^{\boldsymbol{\Sigma}}(Y). \tag{106}$$

The scalar fluctuations are parametrised as $j_{\bar{\mathcal{M}}\bar{\mathcal{N}}}{}^{\boldsymbol{\Sigma}} = 2\mathcal{P}_{\bar{\mathcal{A}},\bar{\mathcal{M}}\bar{\mathcal{N}}}\Phi^{\bar{\mathcal{A}},\boldsymbol{\Sigma}}$, where $\mathcal{P}_{\bar{\mathcal{A}},\bar{\mathcal{M}}\bar{\mathcal{N}}}$ is the projector onto the coset (19). The scalar harmonics satisfy

$$\rho^{-1}(U^{-1})_{\bar{\mathcal{M}}}{}^{\mathcal{M}}\partial_{\mathcal{M}}\mathcal{Y}^{\boldsymbol{\Sigma}} = -\hat{\mathcal{T}}_{\bar{\mathcal{M}}}{}^{\boldsymbol{\Sigma}\boldsymbol{\Omega}}\,\mathcal{Y}^{\boldsymbol{\Omega}}, \tag{107}$$

such that the constant matrices $\hat{\mathcal{T}}_{\bar{\mathcal{M}}}{}^{\boldsymbol{\Sigma}\boldsymbol{\Omega}}$ define the algebra

$$\left[\hat{\mathcal{T}}_{\bar{\mathcal{M}}},\hat{\mathcal{T}}_{\bar{\mathcal{N}}}\right] = X_{[\bar{\mathcal{M}}\bar{\mathcal{N}}]}{}^{\bar{\mathcal{K}}}\hat{\mathcal{T}}_{\bar{\mathcal{K}}}. \tag{108}$$

As in eq. (33) and (99), these matrices can be dressed to describe backgrounds corresponding to other points of the scalar manifold.

With the twist matrix (91) and the physical coordinates embedded in $Y^{MN}$ as in eq. (84), the matrices $\hat{\mathcal{T}}_{\bar{\mathcal{M}}}{}^{\boldsymbol{\Sigma}\boldsymbol{\Omega}}$ have as only non-vanishing components

$$\hat{\mathcal{T}}_{\bar{M}\bar{N}} = 2\mathring{\mathcal{T}}_{\bar{M}\bar{N}}, \tag{109}$$

where $\mathring{\mathcal{T}}_{MN}$ is the SO(8,8) tensor in (97).

Inserting the Ansatz (106) into the ExFT action (72), one can read off the bosonic mass matrices

$$M_{(2)}^2{}^{\Sigma\Omega} = -\Delta^{\bar{\mathcal{M}}\bar{\mathcal{N}}}\hat{\mathcal{T}}_{\bar{\mathcal{M}}}{}^{\Sigma\Gamma}\hat{\mathcal{T}}_{\bar{\mathcal{N}}}{}^{\Gamma\Omega}, \tag{110a}$$

$$M_{(1)}^{\bar{\mathcal{M}}\Sigma}{}_{\bar{\mathcal{N}}}{}^{\Omega} = -\left(\Delta^{\bar{\mathcal{M}}\bar{\mathcal{P}}} + \kappa^{\bar{\mathcal{M}}\bar{\mathcal{P}}}\right)\left(X_{\bar{\mathcal{P}}\bar{\mathcal{N}}}\delta^{\Sigma\Omega} + f_{\bar{\mathcal{P}}\bar{\mathcal{N}}}{}^{\bar{\mathcal{Q}}}\hat{\mathcal{T}}_{\bar{\mathcal{Q}}}{}^{\Sigma\Omega}\right), \tag{110b}$$

$$\begin{aligned}
M_{(0)\mathscr{A}}^2{}^{\Sigma}{}_{\bar{\mathscr{B}}}{}^{\Omega} &= \delta^{\Sigma\Omega}\mathcal{P}_{\mathscr{A}}{}^{\bar{\mathcal{M}}\bar{\mathcal{N}}}\mathcal{P}_{\bar{\mathscr{B}}}{}^{\bar{\mathcal{P}}\bar{\mathcal{Q}}}\left(\frac{1}{7}X_{\bar{\mathcal{M}}\bar{\mathcal{P}}}X_{\bar{\mathcal{N}}\bar{\mathcal{Q}}} + X_{\bar{\mathcal{M}}\bar{\mathcal{K}}}\left(\frac{1}{7}\Delta^{\bar{\mathcal{K}}\bar{\mathcal{L}}} + \kappa^{\bar{\mathcal{K}}\bar{\mathcal{L}}}\right)X_{\bar{\mathcal{L}}\bar{\mathcal{P}}}\Delta_{\bar{\mathcal{N}}\bar{\mathcal{Q}}}\right) \\
&\quad + 2\left(X_{\bar{\mathcal{M}}\mathscr{A}\bar{\mathscr{B}}} - 2X_{[\mathscr{A}\bar{\mathscr{B}}]\bar{\mathcal{M}}}\right)\Delta^{\bar{\mathcal{M}}\bar{\mathcal{N}}}\hat{\mathcal{T}}_{\bar{\mathcal{N}}}{}^{\Sigma\Omega} + 2X_{\bar{\mathcal{M}}\mathscr{A}\bar{\mathscr{B}}}\kappa^{\bar{\mathcal{M}}\bar{\mathcal{N}}}\hat{\mathcal{T}}_{\bar{\mathcal{N}}}{}^{\Sigma\Omega} \\
&\quad - \left(\hat{\mathcal{T}}_{\bar{\mathcal{M}}}\hat{\mathcal{T}}_{\bar{\mathcal{N}}}\right)^{\Sigma\Omega}\Delta^{\bar{\mathcal{M}}\bar{\mathcal{N}}}\kappa_{\mathscr{A}\bar{\mathscr{B}}} + 2\left(\hat{\mathcal{T}}_{\bar{\mathcal{M}}}\hat{\mathcal{T}}_{\bar{\mathcal{N}}}\right)^{\Sigma\Omega}\Delta^{\bar{\mathcal{P}}\bar{\mathcal{Q}}}f_{\mathscr{A}\bar{\mathcal{P}}}{}^{\bar{\mathcal{N}}}f_{\bar{\mathscr{B}}\bar{\mathcal{Q}}}{}^{\bar{\mathcal{M}}} - 2\left(\hat{\mathcal{T}}_{\bar{\mathscr{B}}}\hat{\mathcal{T}}_{\mathscr{A}}\right)^{\Sigma\Omega}.
\end{aligned} \tag{110c}$$

Upon considering the supersymmetric completion of (72) in [54] and the expansions [34]

$$\psi_\mu{}^M(x,Y) = \rho^{-1/2}(Y)\delta^{M\bar{M}}\psi_\mu{}^{\bar{M}\Lambda}(x)\mathcal{Y}^\Lambda(Y), \quad \chi^{\mathscr{A}}(x,Y) = \rho^{1/2}(Y)\delta^{\mathscr{A}\bar{\mathscr{A}}}\chi^{\bar{\mathscr{A}}\Lambda}(x)\mathcal{Y}^\Lambda(Y), \tag{111}$$

for the ExFT gravitini and spin-1/2 fields, their mass matrices can also be found to be

$$\begin{aligned}
M_{(3/2)}{}^{\bar{M}\Sigma,\bar{N}\Omega} &= -\hat{A}_1{}^{\bar{M}\bar{N}}\delta^{\Sigma\Omega} - 4(\mathcal{V}^{-1})_{\bar{M}\bar{N}}{}^{\bar{\mathcal{M}}}\hat{\mathcal{T}}_{\bar{\mathcal{M}}}{}^{\Sigma\Omega}, \\
M_{(1/2)}{}^{\bar{\mathscr{A}}\Sigma,\bar{\bar{\mathscr{B}}}\Omega} &= -\hat{A}_3{}^{\bar{\mathscr{A}}\bar{\bar{\mathscr{B}}}}\delta^{\Sigma\Omega} - \Gamma^{\bar{M}\bar{N}}{}_{\bar{\mathscr{A}}\bar{\bar{\mathscr{B}}}}(\mathcal{V}^{-1})_{\bar{M}\bar{N}}{}^{\bar{\mathcal{M}}}\hat{\mathcal{T}}_{\bar{\mathcal{M}}}{}^{\Sigma\Omega},
\end{aligned} \tag{112}$$

in terms of the shift matrices in eq. (36). The mass matrices (110) and (112) also contain unphysical Goldstone modes that need to be removed using (103) and (104) and decoupled vectors.

# 4 The round $S^3 \times S^3 \times S^1$ and $S^3 \times T^4$ solutions

In this section we show how the techniques discussed in sec. 3 apply to the consistent truncations on the round $AdS_3 \times S^3 \times S^3 \times S^1$ and $AdS_3 \times S^3 \times T^4$ solutions, and how can be used to compute their associated Kaluza-Klein spectra.

## 4.1 Scherk-Schwarz factorisation

**Twist matrix for $S^3 \times S^3 \times S^1$**  The relevant pair $(\rho, U_M{}^{\bar{M}})$ which makes contact with the embedding tensor (39) can be constructed out of two copies of the SO(4,4)-ExFT parallelisation discussed in [41] as

$$(U^{-1})_{\bar{M}}{}^M = \begin{pmatrix}
\rho & 0 & \sqrt{1+\alpha^2}\,\mathring{\xi}^m & 0 & \sqrt{1+\alpha^2}\,\mathring{\tilde{\xi}}^i & 0 & 0 \\
0 & \rho^{-1} & 0 & 0 & 0 & 0 & 0 \\
0 & -\rho^{-1}\sqrt{1+\alpha^2}\,\mathcal{Z}_{\bar{m}m}\mathring{\xi}^m & \mathcal{K}_{\bar{m}}{}^m & \mathcal{Z}_{\bar{m}m} & 0 & 0 & 0 \\
0 & -\rho^{-1}\sqrt{1+\alpha^2}\,\mathcal{Z}^{\bar{m}}{}_m\mathring{\xi}^m & \mathcal{K}^{\bar{m}m} & \mathcal{Z}^{\bar{m}}{}_m & 0 & 0 & 0 \\
0 & -\rho^{-1}\sqrt{1+\alpha^{-2}}\,\widetilde{\mathcal{Z}}_{\bar{i}i}\mathring{\tilde{\xi}}^i & 0 & 0 & \alpha\,\widetilde{\mathcal{K}}_{\bar{i}}{}^i & \alpha^{-1}\widetilde{\mathcal{Z}}_{\bar{i}i} & 0 \\
0 & -\rho^{-1}\sqrt{1+\alpha^{-2}}\,\widetilde{\mathcal{Z}}^{\bar{i}}{}_i\mathring{\tilde{\xi}}^i & 0 & 0 & \alpha\,\widetilde{\mathcal{K}}^{\bar{i}i} & \alpha^{-1}\widetilde{\mathcal{Z}}^{\bar{i}}{}_i & 0 \\
0 & 0 & 0 & 0 & 0 & 0 & \mathbb{1}_2
\end{pmatrix}, \tag{113}$$

in terms of the SO(8, 8) ⊃ SO(1, 1) × GL(3) × GL(3) × SO(1, 1) breaking of both flat and curved indices such that

$$X^M = \{X^0, X_0, X^m, X_m, X^i, X_i, X^7, X_7\}, \tag{114}$$

following eq. (38). The parameter $\alpha$ is the same as in sec. 2, and the objects appearing in eq. (113) are constructed from the Killing vectors on the round $S^3$'s,

$$K_{\alpha\beta\,m} = \mathcal{Y}_{[\alpha|}\partial_m\mathcal{Y}_{|\beta]}, \qquad \widetilde{K}_{\tilde{\alpha}\tilde{\beta}\,i} = \mathcal{Y}_{[\tilde{\alpha}|}\partial_i\mathcal{Y}_{|\tilde{\beta}]}, \tag{115}$$

with $\mathcal{Y}^{\alpha}$ and $\mathcal{Y}^{\tilde{\alpha}}$ the harmonics that embed the spheres in $\mathbb{R}^4$ as $\delta_{\alpha\beta}\mathcal{Y}^{\alpha}\mathcal{Y}^{\beta}=1$ and likewise for tilded indices. The fiducial unit-radius metrics on the round $S^3$'s can be recovered from the Killing vectors as

$$\mathring{g}_{mn}=2K_{\alpha\beta\,m}K_{\gamma\delta\,n}\delta^{\alpha\gamma}\delta^{\beta\delta}\,, \qquad \mathring{\tilde{g}}_{ij}=2\widetilde{K}_{\tilde{\alpha}\tilde{\beta}\,i}\widetilde{K}_{\tilde{\gamma}\tilde{\delta}\,j}\delta^{\tilde{\alpha}\tilde{\gamma}}\delta^{\tilde{\beta}\tilde{\delta}}\,, \tag{116}$$

and the vectors can then be split into $SO(4)\simeq SO(3)_L\times SO(3)_R$ as

$$\begin{cases} L_{\bar{m}}{}^m=\left(K_{4\bar{m}\,n}+\dfrac{1}{2}\,\varepsilon_{4\bar{m}\bar{n}\bar{p}}\,K_{\bar{n}\bar{p}\,n}\right)\mathring{g}^{nm}\,, \\[2mm] R_{\bar{m}}{}^m=\left(K_{4\bar{m}\,n}-\dfrac{1}{2}\,\varepsilon_{4\bar{m}\bar{n}\bar{p}}\,K_{\bar{n}\bar{p}\,n}\right)\mathring{g}^{nm}\,, \end{cases} \tag{117}$$

normalised so that

$$\mathcal{L}_{L_{\bar{m}}}L_{\bar{n}}=\varepsilon_{\bar{m}\bar{n}\bar{p}}\,L_{\bar{p}}\,, \qquad \mathcal{L}_{L_{\bar{m}}}R_{\bar{n}}=0\,, \qquad \mathcal{L}_{R_{\bar{m}}}R_{\bar{n}}=-\varepsilon_{\bar{m}\bar{n}\bar{p}}\,R_{\bar{p}}\,, \tag{118}$$

and analogously for the tilded counterparts.

The different blocks in the twist matrix (113) are then given by the $SO(3,3)\subset SO(4,4)$ vectors

$$\begin{aligned} \mathcal{K}_{\bar{m}}{}^m&=L_{\bar{m}}{}^m+R_{\bar{m}}{}^m\,, \\ \mathcal{K}^{\bar{m}m}&=(R_{\bar{n}}{}^m-L_{\bar{n}}{}^m)\,\delta^{\bar{n}\bar{m}}\,, \end{aligned} \tag{119}$$

and one-forms

$$\begin{aligned} \mathcal{Z}_{\bar{m}m}&=\delta_{\bar{m}\bar{n}}\mathcal{K}^{\bar{n}n}\mathring{g}_{nm}-2\sqrt{\mathring{g}}\,\mathcal{K}_{\bar{m}}{}^n\varepsilon_{mnp}\,\mathring{\xi}^p\,, \\ \mathcal{Z}^{\bar{m}}{}_m&=\delta^{\bar{m}\bar{n}}\mathcal{K}_{\bar{n}}{}^n\mathring{g}_{nm}-2\sqrt{\mathring{g}}\,\mathcal{K}^{\bar{m}n}\varepsilon_{mnp}\,\mathring{\xi}^p\,, \end{aligned} \tag{120}$$

with $\mathring{\xi}$ a vector satisfying $\mathring{\nabla}_m\mathring{\xi}^m=1$ with respect to the Levi-Civita connection associated to the metric (116). The analogous objects $\widetilde{\mathcal{K}}_{\bar{i}}{}^i$, $\widetilde{\mathcal{K}}^{\bar{i}i}$, $\widetilde{\mathcal{Z}}_{\bar{i}i}$, $\widetilde{\mathcal{Z}}^{\bar{i}}{}_i$ and $\mathring{\tilde{\xi}}^i$ are defined similarly for $\widetilde{S}^3$.

Together with the scaling function

$$\rho=\alpha^3\,\mathring{g}^{-1/2}\,\mathring{\tilde{g}}^{-1/2}\,, \tag{121}$$

these objects recover the embedding tensor (39) via eq. (67). Moreover, if we parameterise the ExFT coordinates $Y^{i,0}$ in (61) as

$$\begin{aligned} Y^{m,0}:\quad &Y^{1,0}=\cos(\theta)\cos(\varphi_1)\,, \quad Y^{2,0}=\cos(\theta)\sin(\varphi_1)\,, \quad Y^{3,0}=\sin(\theta)\cos(\varphi_2)\,, \\ Y^{i,0}:\quad &Y^{4,0}=\cos(\widetilde{\theta})\cos(\widetilde{\varphi}_1)\,, \quad Y^{5,0}=\cos(\widetilde{\theta})\sin(\widetilde{\varphi}_1)\,, \quad Y^{6,0}=\sin(\widetilde{\theta})\cos(\widetilde{\varphi}_2)\,, \\ &Y^{7,0}=y^7\,, \end{aligned} \tag{122}$$

with

$$0\le\theta,\widetilde{\theta}\le\frac{\pi}{2}\,, \qquad 0\le\varphi_i,\widetilde{\varphi}_i\le2\pi\,, \qquad 0\le y^7\le1\,, \tag{123}$$

with $i\in\{1,2\}$, and using the dictionary in (62), we can write the AdS$_3\times S^3\times\widetilde{S}^3\times S^1$ solution as

$$\begin{aligned} e^{\hat{\Phi}}&=1\,, \\ d\hat{s}^2&=\ell_{\text{AdS}}^2\,ds^2(\text{AdS}_3)+d\theta^2+\cos^2(\theta)\,d\varphi_1^2+\sin^2(\theta)\,d\varphi_2^2+\alpha^{-2}\big(d\widetilde{\theta}^2+\cos^2(\widetilde{\theta})\,d\widetilde{\varphi}_1^2+\sin^2(\widetilde{\theta})\,d\widetilde{\varphi}_2^2\big)+(dy^7)^2\,, \\ \hat{H}_{(3)}&=2\ell_{\text{AdS}}^2\,\text{vol}(\text{AdS}_3)+2\sin(\theta)\cos(\theta)\,d\theta\wedge d\varphi_1\wedge d\varphi_2+2\alpha^{-2}\sin(\widetilde{\theta})\cos(\widetilde{\theta})\,d\widetilde{\theta}\wedge d\widetilde{\varphi}_1\wedge d\widetilde{\varphi}_2\,, \\ \hat{G}_{(1)}&=\hat{G}_{(3)}=\hat{G}_{(5)}=0\,, \end{aligned} \tag{124}$$

both in the string and Einstein frames owing to the vanishing dilaton. Here and throughout, $ds^2(\text{AdS}_3)$ denotes the unit-radius metric on AdS$_3$, $\ell_{\text{AdS}}$ is the AdS length in (48), and vol(AdS$_3$) its associated volume form. In (124), $S^3$ has unit radius, whereas $\widetilde{S}^3$ has radius $\alpha^{-1}$. For later convenience, we will choose a gauge such that the local two-form potential leading to the internal part of the Kalb-Ramond three-form in (124) is given by

$$\hat{B}_{(2)}=\sin^2(\theta)\,d\varphi_1\wedge d\varphi_2+\alpha^{-2}\sin^2(\widetilde{\theta})\,d\widetilde{\varphi}_1\wedge d\widetilde{\varphi}_2\,. \tag{125}$$

**Twist matrix for $S^3 \times T^4$** The twist matrix for $S^3 \times T^4$ can be similarly parameterised as

$$(U^{-1})_{\bar{M}}{}^{M} = \begin{pmatrix} \rho & 0 & 2\mathring{\xi}^m & 0 & 0 \\ 0 & \rho^{-1} & 0 & 0 & 0 \\ 0 & -2\rho^{-1}\mathcal{Z}_{\bar{m}m}\mathring{\xi}^m & \mathcal{K}_{\bar{m}}{}^{m} & \mathcal{Z}_{\bar{m}m} & 0 \\ 0 & -2\rho^{-1}\mathcal{Z}^{\bar{m}}{}_{m}\mathring{\xi}^m & \mathcal{K}^{\bar{m}m} & \mathcal{Z}^{\bar{m}}{}_{m} & 0 \\ 0 & 0 & 0 & 0 & \mathbb{1}_8 \end{pmatrix}, \tag{126}$$

in terms of the $\mathcal{K}$, $\mathcal{Z}$ and $\mathring{\xi}$ tensors above, and the scaling function

$$\rho = \mathring{g}^{-1/2}. \tag{127}$$

Embedding the $S^3 \times T^4$ coordinates in ExFT as

$$Y^{1,0} = \cos(\theta)\cos(\varphi_1), \quad Y^{2,0} = \cos(\theta)\sin(\varphi_1), \quad Y^{3,0} = \sin(\theta)\cos(\varphi_2), \quad Y^{a,0} = y^a, \tag{128}$$

the AdS$_3 \times S^3 \times T^4$ solution reads

$$\begin{aligned} e^{\hat{\Phi}} &= 1, \\ d\hat{s}^2 &= \ell_{\text{AdS}}^2 \, ds^2(\text{AdS}_3) + d\theta^2 + \cos^2(\theta)\,d\varphi_1^2 + \sin^2(\theta)\,d\varphi_2^2 + \delta_{ab}\,dy^a dy^b, \\ \hat{H}_{(3)} &= 2\ell_{\text{AdS}}^2 \, \text{vol}(\text{AdS}_3) + 2\sin(\theta)\cos(\theta)\,d\theta \wedge d\varphi_1 \wedge d\varphi_2, \\ \hat{G}_{(1)} &= \hat{G}_{(3)} = \hat{G}_{(5)} = 0, \end{aligned} \tag{129}$$

with the coordinates now ranging as

$$0 \leq \theta \leq \frac{\pi}{2}, \qquad 0 \leq \varphi_1, \varphi_2 \leq 2\pi, \qquad 0 \leq y^a \leq 1, \tag{130}$$

and the local two-form potential simply given by

$$\hat{B}_{(2)} = \sin^2(\theta)d\varphi_1 \wedge d\varphi_2. \tag{131}$$

## 4.2 Sphere harmonics

For the products of spheres under consideration, the composite index $\mathbf{\Lambda}$ in (96) and (106) labels representations in the infinite-dimensional towers

$$\begin{aligned} S^3 \times T^4 &: \quad \bigoplus_{p_a \in \mathbb{Z}} \bigoplus_{m=0}^{\infty} \left(\tfrac{m}{2}, \tfrac{m}{2}\right)_{(p_4, p_5, p_6, p_7)} && \text{of } \text{SO}(4) \times \text{SO}(2)^4, \\ S^3 \times \widetilde{S}^3 \times S^1 &: \quad \bigoplus_{p_7 \in \mathbb{Z}} \bigoplus_{m=0}^{\infty} \bigoplus_{\tilde{m}=0}^{\infty} \left(\tfrac{m}{2}, \tfrac{m}{2}; \tfrac{\tilde{m}}{2}, \tfrac{\tilde{m}}{2}\right)_{p_7} && \text{of } \text{SO}(4) \times \text{SO}(4) \times \text{SO}(2), \end{aligned} \tag{132}$$

and the corresponding harmonics factorise as

$$\begin{aligned} \mathcal{Y}^{\mathbf{\Lambda}} &= \mathcal{Y}^{\Lambda} e^{2\pi i \sum p_a y_a}, && \text{for } S^3 \times T^4, \\ \mathcal{Y}^{\mathbf{\Lambda}} &= \mathcal{Y}^{\Lambda} \mathcal{Y}^{\tilde{\Lambda}} e^{2\pi i p_7 y_7}, && \text{for } S^3 \times \widetilde{S}^3 \times S^1, \end{aligned} \tag{133}$$

with each one-cycle having length 1. The SO(4) harmonics,

$$\mathcal{Y}^{\Lambda} = \left\{1, \, \mathcal{Y}^{\alpha}, \, \mathcal{Y}^{\{\alpha}\mathcal{Y}^{\beta\}}, \ldots\right\}, \qquad \alpha \in [\![1,4]\!], \tag{134}$$

and similarly for $\mathcal{Y}^{\tilde{\Lambda}}$, correspond to symmetric-traceless products of the level-one harmonics for the round $S^3$'s, which we choose as

$$\mathcal{Y}^{\alpha} = \left\{y^1, \, y^2, \, y^3, \, \sqrt{1-(y^1)^2-(y^2)^2-(y^3)^2}\right\}, \tag{135}$$

and analogously for $\mathcal{Y}^{\tilde{\alpha}}$.

Following (133) the matrices in (97) can in turn be decomposed as (*c.f.* $S^5 \times S^1$ S-fold solutions in [21, 23])

$$\mathring{\mathcal{T}}_{\bar{M}\bar{N}}{}^{(p_a)\Lambda\Sigma} = \mathring{\mathcal{T}}_{\bar{M}\bar{N}}{}^{\Lambda\Sigma} + \delta^{\Lambda\Sigma}\mathring{\mathcal{T}}_{\bar{M}\bar{N}}{}^{(p_a)}, \qquad\qquad \text{for } S^3 \times \mathrm{T}^4, $$
$$\mathring{\mathcal{T}}_{\bar{M}\bar{N}}{}^{(p_7)\Lambda\tilde{\Lambda}\Sigma\tilde{\Sigma}} = \delta^{\tilde{\Lambda}\tilde{\Sigma}}\mathring{\mathcal{T}}_{\bar{M}\bar{N}}{}^{\Lambda\Sigma} + \delta^{\Lambda\Sigma}\mathring{\mathcal{T}}_{\bar{M}\bar{N}}{}^{\tilde{\Lambda}\tilde{\Sigma}} + \delta^{\Lambda\Sigma}\delta^{\tilde{\Lambda}\tilde{\Sigma}}\mathring{\mathcal{T}}_{\bar{M}\bar{N}}{}^{(p_7)}, \quad \text{for } S^3 \times \widetilde{S}^3 \times S^1, \quad (136)$$

analogously for their maximal counterparts in (107).

For the $S^3 \times \mathrm{T}^4$ background (126), the SO(4) matrix $\mathring{\mathcal{T}}_{\bar{M}\bar{N}}{}^{\Lambda\Sigma}$ has non-vanishing components

$$\mathring{\mathcal{T}}_{\bar{m}\bar{0}}{}^{\alpha\beta} = \delta_4^{[\alpha}\delta_{\bar{m}}^{\beta]}, \qquad\qquad \mathring{\mathcal{T}}^{\bar{m}}{}_{\bar{0}}{}^{\alpha\beta} = \frac{1}{2}\varepsilon^{\bar{m}4\alpha\beta}, \qquad (137)$$

when acting on the level $m = 1$ harmonics in eq. (135). Similarly, in the $S^3 \times \widetilde{S}^3 \times S^1$ case we have

$$\mathring{\mathcal{T}}_{\bar{m}\bar{0}}{}^{\alpha\beta} = \delta_4^{[\alpha}\delta_{\bar{m}}^{\beta]}, \quad \mathring{\mathcal{T}}^{\bar{m}}{}_{\bar{0}}{}^{\alpha\beta} = \frac{1}{2}\varepsilon^{\bar{m}4\alpha\beta}, \quad \mathring{\mathcal{T}}_{\bar{m}\bar{0}}{}^{\tilde{\alpha}\tilde{\beta}} = \alpha\,\delta_4^{[\tilde{\alpha}}\delta_{\bar{m}}^{\tilde{\beta}]}, \quad \mathring{\mathcal{T}}^{\bar{m}}{}_{\bar{0}}{}^{\tilde{\alpha}\tilde{\beta}} = \frac{\alpha}{2}\varepsilon^{\bar{m}4\tilde{\alpha}\tilde{\beta}}. \quad (138)$$

At higher levels, the tensors $\mathring{\mathcal{T}}_{\bar{M}\bar{N}}{}^{\Lambda\Sigma}$ can be constructed recursively from (137) and (138) as

$$(\mathring{\mathcal{T}}_{\bar{M}\bar{N}})_{\alpha_1\dots\alpha_m}{}^{\beta_1\dots\beta_m} = m(\mathring{\mathcal{T}}_{\bar{M}\bar{N}})_{\{\alpha_1}{}^{\{\beta_1}\delta_{\alpha_2}^{\beta_2}\dots\delta_{\alpha_m\}}^{\beta_m\}}, \qquad (139)$$

and analogously for $\mathring{\mathcal{T}}_{\bar{M}\bar{N}}{}^{\tilde{\Lambda}\tilde{\Sigma}}$. Similarly, the SO(2) blocks are simply given by

$$\mathring{\mathcal{T}}_{\bar{a}\bar{0}}{}^{(p_a)} = -\pi i\, p_a. \qquad (140)$$

In this conventions, the matrices in (136) are complex, and hermitian conjugations need to be introduced in the mass matrices. Equivalently, we could have used manifestly real objects at the price of introducing a two-fold degeneracy in the eigenvalues.

## 4.3 Spectra on the round solutions

As the AdS$_3$ isometry group $\mathrm{SO}(2,2) \simeq \mathrm{SL}(2,\mathbb{R}) \times \mathrm{SL}(2,\mathbb{R})$ is not simple, the superisometry group of AdS$_3$ background is in general a direct product of simple supergroups $\mathcal{G} = \mathcal{G}_\mathrm{L} \times \mathcal{G}_\mathrm{R}$. The spectrum of such backgrounds organises into representations of $\mathcal{G}$, with conformal dimension $\Delta = \Delta_\mathrm{L} + \Delta_\mathrm{R}$ built from the conformal dimensions of each $\mathcal{G}_\mathrm{L,R}$ factor. The spacetime spin $s$ of a field in a given representation is then given by $s = \Delta_\mathrm{R} - \Delta_\mathrm{L}$. In the following, we use the ExFT spectroscopy of sec. 3.4 to compute the masses $m_{(s)}$ of each Kaluza-Klein tower of spin $s$, and identify the corresponding conformal dimensions from [61–66]

$$\begin{cases} \Delta_{(0)}\left(\Delta_{(0)} - 2\right) = \left(m_{(0)}\ell_{\mathrm{AdS}}\right)^2, \\ \Delta_{(1)} = 1 + |m_{(1)}\ell_{\mathrm{AdS}}|, \end{cases} \quad \text{and} \quad \begin{cases} \Delta_{(1/2)} = 1 \pm m_{(1/2)}\ell_{\mathrm{AdS}}, \\ \Delta_{(3/2)} = 1 + |m_{(3/2)}\ell_{\mathrm{AdS}}|, \end{cases} \quad (141)$$

where the masses are normalised by the AdS length $\ell_{\mathrm{AdS}}$.

The Kaluza-Klein spectrum on the round $S^3 \times \widetilde{S}^3 \times S^1$ solution of type II supergravity, recently revisited in [19], organises into supermultiplets of

$$\mathcal{G}_{\alpha \neq 0} = \mathrm{D}(2,1|\alpha)_\mathrm{L} \times \mathrm{D}(2,1|\alpha)_\mathrm{R} \times \mathrm{U}(1), \qquad (142)$$

with $\mathrm{D}(2,1|\alpha)$ the large $\mathcal{N} = 4$ supergroup in three dimensions and $\alpha$ the ratio of the $S^3$ radii. The even part of (142),

$$\mathrm{SO}(2,2) \times \mathrm{SO}(4) \times \widetilde{\mathrm{SO}(4)} \times \mathrm{U}(1), \qquad (143)$$

now corresponds to the isometries of AdS$_3 \times S^3 \times \widetilde{S}^3 \times S^1$, with $\mathrm{SO}(2,2) \simeq \mathrm{SL}(2,\mathbb{R})_\mathrm{L} \times \mathrm{SL}(2,\mathbb{R})_\mathrm{R}$, $\mathrm{SO}(4) \simeq \mathrm{SU}(2)_\mathrm{L} \times \mathrm{SU}(2)_\mathrm{R}$ and similarly for the tilded counterparts. The long multiplets of each

D(2, 1|$\alpha$) can be labelled as $\left[h, j^+, j^-\right]$ [67] (see appendix B for a review), and the complete spectrum of type II supergravity then reads [67]

$$\mathcal{S} = \bigoplus_{\substack{\ell, \tilde{\ell} \geq 0 \\ p_7 \in \mathbb{Z}}} \left( [h_0, \ell, \tilde{\ell}] \otimes [h_0, \ell, \tilde{\ell}] \right)_{p_7}, \tag{144}$$

with $p_7$ the U(1) integer charge and

$$h_0 = -\frac{1}{2} + \frac{1}{2} \sqrt{1 + \frac{4\ell(\ell+1) + 4\alpha^2 \tilde{\ell}(\tilde{\ell}+1) + (2\pi p_7)^2}{1 + \alpha^2}}. \tag{145}$$

The dimension of the superconformal primary of $\left( [h_0, \ell, \tilde{\ell}] \otimes [h_0, \ell, \tilde{\ell}] \right)_{p_7}$ is then $\Delta = 2h_0$. For these multiplets, shortening occurs when $p_7 = 0$ and $\ell = \tilde{\ell}$ following equation (B.6) of appendix B.

The case of the heterotic string can be described using the half-maximal supergravity of sec. 2.1. Given the 16 vector fields coupled to the NSNS fields in ten-dimensions, the three-dimensional supergravity arising from compactification to three dimensions has a coset space in the class of eq. (1),

$$\frac{\mathrm{SO}(8, 24)}{\mathrm{SO}(8) \times \mathrm{SO}(24)}. \tag{146}$$

The heterotic gauged supergravity is then obtained by embedding the SO(8,8) tensors of sec. 2.3 in SO(8, 24). All vacua of the SO(8,8) theory are vacua from the SO(8, 24) theory. The supergroup organising the spectrum at the scalar origin is

$$\mathrm{SL}(2, \mathbb{R})_{\mathrm{L}} \times \mathrm{SU}(2)_{\mathrm{L}} \times \widetilde{\mathrm{SU}(2)}_{\mathrm{L}} \times \mathrm{D}(2, 1|\alpha)_{\mathrm{R}} \times \mathrm{U}(1) \times \mathrm{SO}(16). \tag{147}$$

The bosonic isometries of the background are built similarly as in the maximal case, with an additional SO(16) factor for the heterotic vector fields. The spectrum can be described based on that in eq. (144). For each term in the sum, the left factor $[h_0, \ell, \tilde{\ell}]$ breaks into

$$
\begin{array}{ll}
\mathrm{SL}(2, \mathbb{R}) & \mathrm{SU}(2)_{\mathrm{L}} \times \widetilde{\mathrm{SU}(2)}_{\mathrm{L}} \\
\hline
h_0 & (\ell, \tilde{\ell}) \\
1 + h_0 & (\ell+1, \tilde{\ell}) \oplus (\ell, \tilde{\ell}-1) \oplus (\ell, \tilde{\ell}) \oplus (\ell, \tilde{\ell}) \oplus (\ell-1, \tilde{\ell}) \oplus (\ell, \tilde{\ell}+1) \\
2 + h_0 & (\ell, \tilde{\ell})
\end{array}
\tag{148}
$$

and the spectrum is supplemented at each level by 16 copies of the multiplet

$$\left( (1 + h_0, \ell, \tilde{\ell}) \otimes [h_0, \ell, \tilde{\ell}] \right)_{p_7}, \tag{149}$$

transforming as a vector of SO(16).

Regarding the spectrum of the round $S^3 \times \mathrm{T}^4$ solution, it abides by the supergroup

$$\mathcal{G}_{\alpha=0} = \left[ \widetilde{\mathrm{SU}(2)}_{\mathrm{L}} \ltimes \mathrm{SU}(2|1, 1)_{\mathrm{L}} \right] \times \left[ \widetilde{\mathrm{SU}(2)}_{\mathrm{R}} \ltimes \mathrm{SU}(2|1, 1)_{\mathrm{R}} \right] \times \mathrm{U}(1)^4, \tag{150}$$

where SU(2|1, 1) is the small $\mathcal{N} = 4$ supergroup in three dimensions. The even part of the superisometry corresponds to the isometries of $\mathrm{AdS}_3 \times S^3 \times \mathrm{T}^4$,

$$\mathrm{SO}(2, 2) \times \mathrm{SO}(4) \times \mathrm{U}(1)^4, \tag{151}$$

where $\mathrm{SO}(2, 2) \simeq \mathrm{SL}(2, \mathbb{R})_{\mathrm{L}} \times \mathrm{SL}(2, \mathbb{R})_{\mathrm{R}}$ and $\mathrm{SO}(4) \simeq \mathrm{SU}(2)_{\mathrm{L}} \times \mathrm{SU}(2)_{\mathrm{R}}$ correspond to the $\mathrm{AdS}_3 \times S^3$ isometries, together with an extra global $\widetilde{\mathrm{SO}(4)} \simeq \widetilde{\mathrm{SU}(2)}_{\mathrm{L}} \times \widetilde{\mathrm{SU}(2)}_{\mathrm{R}}$ factor corresponding to

relabelling of the torus angles. We denote $\left[\Delta, j^+, j^-\right]$ the long multiplets of $\mathrm{SU}(2)_- \ltimes \mathrm{SU}(2|1,1)_+$ and $p_a$ the $\mathrm{U}(1)^4$ charges. See appendix B for a review of the multiplet content of this superalgebra. The spectrum is then given by

$$\mathcal{S} = \bigoplus_{\substack{\ell \geq 0 \\ p_a \in \mathbb{Z}^4}} \left(\left[h_0, \ell, 0\right] \otimes \left[h_0, \ell, 0\right]\right)_{p_4, p_5, p_6, p_7}, \tag{152}$$

where

$$h_0 = -\frac{1}{2} + \frac{1}{2}\sqrt{1 + 4\ell(\ell+1) + \sum_a (2\pi p_a)^2}. \tag{153}$$

The conformal dimension of the primary of each factor is then $\Delta = 2h_0$. The unitary bound (B.6) is saturated for $p_4 = p_5 = p_6 = p_7 = 0$, and the multiplets get shortened according to (B.7). Therefore, at levels with $\ell = 0$ equation (152) must be interpreted as a shorthand of

$$\left(\left[\tfrac{1}{2}, \tfrac{1}{2}\right]_S \oplus \left[0, 1\right]_S\right) \otimes \left(\left[\tfrac{1}{2}, \tfrac{1}{2}\right]_S \oplus \left[0, 1\right]_S\right) \oplus \bigoplus_{p_a \in \mathbb{Z}^4 \setminus \{0\}} \left(\left[\Delta_L, 0, 0\right] \otimes \left[\Delta_R, 0, 0\right]\right)_{p_4, p_5, p_6, p_7}. \tag{154}$$

In the heterotic case, the supergroup organising the spectrum is

$$\left[\mathrm{SL}(2, \mathbb{R})_L \times \widetilde{\mathrm{SU}(2)}_L \times \mathrm{SU}(2)_L\right] \times \left[\widetilde{\mathrm{SU}(2)}_R \ltimes \mathrm{SU}(2|1,1)_R\right] \times \mathrm{U}(1)^4 \times \mathrm{SO}(16). \tag{155}$$

The spectrum can again be described based on that in eq. (152). The left factor $\left[h_0, \ell, 0\right]$ of each term breaks into

$$\begin{array}{c|c} \Delta_L & \mathrm{SU}(2)_L \times \widetilde{\mathrm{SU}(2)}_L \\ \hline h_0 & (\ell, 0) \\ 1 + h_0 & (\ell, 1) \oplus (\ell+1, 0) \oplus (\ell, 0) \oplus (\ell-1, 0) \\ 2 + h_0 & (\ell, 0) \end{array} \tag{156}$$

with $h_0$ given in (153), and 16 additional copies of the multiplet

$$\left(\left(1 + h_0, \ell, 0\right) \otimes \left[\ell, 0\right]\right)_{p_4, p_5, p_6, p_7} \tag{157}$$

adds up to each level.

# 5 Deformations

The three-dimensional solutions in (51) can be uplifted to ten dimensions as deformations of the round $S^3 \times T^4$ and $S^3 \times \widetilde{S}^3 \times S^1$ configurations reviewed in the previous section. For clarity, we will refrain from presenting the entire 17-parameter family, and focus instead on some subfamilies that best exemplify different phenomena. We discuss only the solutions in type IIB, but all of them have vanishing RR fluxes so the heterotic case follows easily.

Given that all the moduli in (51) belong to $\mathrm{SO}(7,7)/\mathrm{SO}(7) \times \mathrm{SO}(7)$, it is interesting to analyse the solutions from a generalised geometry perspective in terms of a generalised metric

$$\mathcal{H} = \begin{pmatrix} \hat{g}_s - \hat{B} \hat{g}_s^{-1} \hat{B} & \hat{B} \hat{g}_s^{-1} \\ -\hat{g}_s^{-1} \hat{B} & \hat{g}_s^{-1} \end{pmatrix}, \tag{158}$$

with $\hat{g}_s$ and $\hat{B}$ the internal components of the string frame metric and two-form. The deformed solutions (51) can be described as transformations of the undeformed solution by a constant $\mathrm{SO}(7,7)$ element that depends on the marginal parameters,

$$\mathcal{H}_{\mathrm{def}} = \Gamma \cdot \mathcal{H}_{\mathrm{round}} \cdot \Gamma^t, \tag{159}$$

with $\Gamma \in SO(7,7)$. This element can be written as products of $GL(7,\mathbb{R})$ transformations $\mathcal{G}$, constant shifts of the $B$ field $\mathcal{B}$ and TsT transformations $\mathcal{T}$, which are a sequence of T duality, shifts in the $B$ field and T duality, where

$$\mathcal{G} = \begin{pmatrix} \rho & 0 \\ 0 & \rho^{-t} \end{pmatrix}, \qquad \mathcal{B} = \begin{pmatrix} \mathbb{1} & s \\ 0 & \mathbb{1} \end{pmatrix}, \qquad \mathcal{T} = \begin{pmatrix} \mathbb{1} & 0 \\ \beta & \mathbb{1} \end{pmatrix}, \tag{160}$$

with

$$\rho \in GL(7,\mathbb{R}), \qquad \beta, s \in \bigwedge^2 \mathbb{R}^7. \tag{161}$$

In the following, we choose to represent the $SO(7,7)$ elements as

$$\Gamma = \mathcal{T} \cdot \mathcal{G} \cdot \mathcal{B}. \tag{162}$$

These elements will be expressed in the bases

$$\begin{array}{ll} \{d\theta, d\varphi_1, d\varphi_2, d\widetilde{\theta}, d\widetilde{\varphi}_1, d\widetilde{\varphi}_2, dy^7\}, & \text{for AdS}_3 \times S^3 \times \widetilde{S}^3 \times S^1, \\ \{d\theta, d\varphi_1, d\varphi_2, dy^4, dy^5, dy^6, dy^7\}, & \text{for AdS}_3 \times S^3 \times T^4. \end{array} \tag{163}$$

## 5.1 Uplift of the $(\omega \chi \beta)$-family

The two-parameter family of AdS$_3 \times S^3$ solutions in six-dimensional $\mathcal{N} = (1,1)$ supergravity found in [41] through the uplift of (47) can be further lifted into the NSNS sector of type IIB supergravity. See app. D for a direct account on how to construct this Ansatz for AdS$_3 \times S^3 \times T^4$. More generally, using ExFT we can obtain its embedding both in AdS$_3 \times S^3 \times \widetilde{S}^3 \times S^1$ and AdS$_3 \times S^3 \times T^4$ [9], which reads

$$\begin{aligned} e^{\hat{\Phi}} &= \sqrt{\Delta}, \\ d\hat{s}_s^2 &= \ell_{\text{AdS}}^2 \, ds^2(\text{AdS}_3) + d\theta^2 + e^\omega \Delta \big( \cos^2\theta \, d\varphi_1^2 + (\zeta^2 + e^{-2\omega}) \sin^2\theta \, d\varphi_2^2 \big) + ds^2(\widetilde{M}_3) \\ &\quad + 2 e^\omega \zeta \Delta dy^7 \big( \cos^2\theta \, d\varphi_1 - \sin^2\theta \, d\varphi_2 \big) + (dy^7)^2, \\ \hat{H}_{(3)} &= 2\ell_{\text{AdS}}^2 \, \text{vol}(\text{AdS}_3) + 2H(\alpha) \, \text{vol}(\widetilde{M}^3) \\ &\quad + \sin(2\theta) \Delta^2 e^{2\omega} d\theta \wedge (d\varphi_1 + \zeta dy^7) \wedge \big( (\zeta^2 + e^{-2\omega}) d\varphi_2 - \zeta dy^7 \big), \\ \hat{G}_{(1)} &= \hat{G}_{(3)} = \hat{G}_{(5)} = 0, \end{aligned} \tag{164}$$

with

$$ds^2(\widetilde{M}_3) = \begin{cases} \alpha^{-2} \big( d\widetilde{\theta}^2 + \cos^2\widetilde{\theta} \, d\widetilde{\varphi}_1^2 + \sin^2\widetilde{\theta} \, d\widetilde{\varphi}_2^2 \big), & \text{for AdS}_3 \times S^3 \times \widetilde{S}^3 \times S^1, \\ \delta_{ij} dy^i dy^j, & \text{for AdS}_3 \times S^3 \times T^4, \end{cases} \tag{165}$$

the function

$$\Delta = \frac{e^{-\omega}}{1 + (\zeta^2 + e^{-2\omega} - 1)\cos^2\theta}, \tag{166}$$

and $H(\alpha)$ a Heaviside function with $H(0) = 0$. The angles parameterising this manifold range as in (123) and (130). The moduli $(\omega, \zeta)$ define a perturbatively stable solution if all scalars within the spectrum satisfy the Breithenlohner-Freedman (BF) bound $(m\ell_{\text{AdS}})^2 \geq -1$ [40]. This restricts the parameters as

$$e^{-\omega} \leq \frac{2}{\sqrt{3}}, \qquad \zeta^2 \geq \frac{\sqrt{3}}{2} e^{-\omega} - e^{-2\omega}. \tag{167}$$

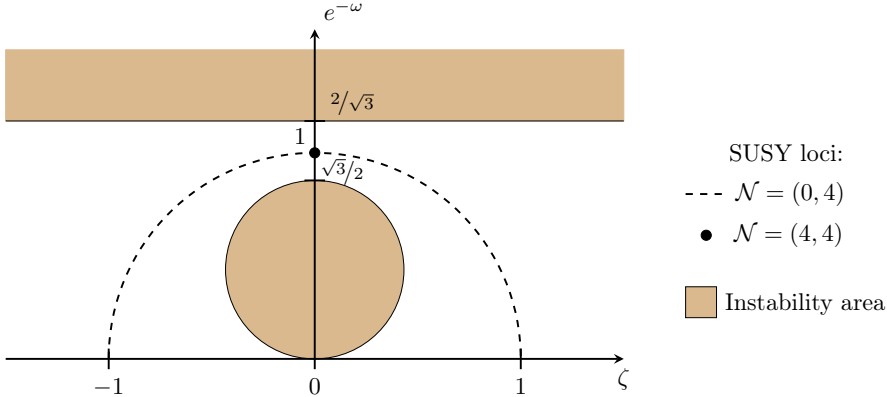

Figure 1: The solution (164) is perturbatively stable for any couple of parameters outside of the instability area, as given by eq. (167). The dashed line represents eq. (49), where $\mathcal{N} = (0,4)$ supersymmetries are preserved. At $\omega = \zeta = 0$ supersymmetry further enhances to $\mathcal{N} = (4,4)$.

See fig. 1 for a graphical representation. At the locus (49), where supersymmetric enhancement takes place, the configuration (164) becomes [9]

$$
\begin{aligned}
\hat{\Phi} &= -\frac{\omega}{2}, \\
d\hat{s}_s^2 &= \ell_{\text{AdS}}^2 \, ds^2(\text{AdS}_3) + ds^2(\mathbb{CP}^1) + e^{-2\omega}\eta^2 + ds^2(\tilde{M}_3) + \left(dy^7 + \sqrt{1-e^{-2\omega}}\,\eta\right)^2, \\
\hat{H}_{(3)} &= 2\ell_{\text{AdS}}^2 \, \text{vol}(\text{AdS}_3) + 2H(\alpha)\,\text{vol}(\tilde{M}^3) + 2\,\eta \wedge J + 2\sqrt{1-e^{-2\omega}}\, J \wedge dy^7, \\
\hat{G}_{(1)} &= \hat{G}_{(3)} = \hat{G}_{(5)} = 0,
\end{aligned}
\tag{168}
$$

with

$$
\eta = \cos^2\theta \, d\varphi_1 - \sin^2\theta \, d\varphi_2, \qquad\qquad J = \tfrac{1}{2}d\eta. \tag{169}
$$

In terms of the $SO(7,7)$ transformations in (162), the family of solutions in (164) is described by

$$
\beta = -\zeta \, d\varphi_1 \wedge dy^7 + (e^{-\omega}-1)d\varphi_1 \wedge d\varphi_2,
$$

$$
\rho = \begin{pmatrix} 1 & 0 & 0 & 0 \\ 0 & e^\omega & 0 & 0 \\ 0 & 0 & \mathbb{1}_4 & 0 \\ 0 & e^\omega\zeta & 0 & 1 \end{pmatrix}, \qquad s = 0. \tag{170}
$$

As apparent from (170), the family of solutions (164) cannot be generated via pure TsT transformations [20], since there is no value of the moduli $\omega$ and $\zeta$ for which both $\mathcal{G}$ and $\mathcal{B}$ reduce to the identity and $\mathcal{T}$ remains non-trivial. Nevertheless, we can achieve this by uncoupling the parameters $\chi_1$ and $\beta_1$ in (51). If we consider

$$
\mathcal{V}_{\bar{\mathcal{M}}}{}^{\mathcal{N}} = \exp\left[-\omega f^{\bar{3}}{}_{\bar{3}} - \frac{\omega}{1-e^{-\omega}}\left(\chi_1 f^{\bar{3}\bar{7}} + \beta_1 f^{\bar{3}}{}_{\bar{7}}\right)\right], \tag{171}
$$

the type IIB supergravity solution is

$$
\begin{aligned}
e^{\hat{\Phi}} &= \sqrt{\Delta}, \\
d\hat{s}_s^2 &= \ell_{\text{AdS}}^2 \, ds^2(\text{AdS}_3) + ds^2(M^3) + d\theta^2 \\
&\quad + e^\omega\Delta\left[\cos^2\theta\left(d\varphi_1 + \chi_1 \, dy^7\right)^2 + \sin^2\theta\left(\beta_1 \, d\varphi_2 + dy^7\right)^2 + e^{-2\omega}\sin^2\theta \, d\varphi_2^2 + e^{-2\omega}\cos^2\theta \, (dy^7)^2\right],
\end{aligned}
$$

$$\hat{H}_{(3)} = 2\ell_{\text{AdS}}^2 \, \text{vol}(\text{AdS}_3) + 2H(\alpha)\,\text{vol}(\tilde{M}^3)$$
$$+ \sin(2\theta)\Delta^2 e^{2\omega}\mathrm{d}\theta \wedge \left(\mathrm{d}\varphi_1 + \chi_1\,\mathrm{d}y^7\right) \wedge \left(\left(\beta_1^2 + e^{-2\omega}\right)\mathrm{d}\varphi_2 + \beta_1\,\mathrm{d}y^7\right),$$
$$\hat{G}_{(1)} = \hat{G}_{(3)} = \hat{G}_{(5)} = 0, \tag{172}$$

with

$$\Delta = \frac{e^{-\omega}}{1 + \left(\beta_1^2 + e^{-2\omega} - 1\right)\cos^2\theta}. \tag{173}$$

The deformation generically breaks the SO(4) factor in (143) and (151) to the Cartan subalgebra $U(1)_L \times U(1)_R$ and all supersymmetries. The relevant SO(7,7) transformation to construct (172) is given by

$$\beta = \beta_1\,\mathrm{d}\varphi_1 \wedge \mathrm{d}y^7 + (e^{-\omega} - 1)\,\mathrm{d}\varphi_1 \wedge \mathrm{d}\varphi_2,$$

$$\rho = \begin{pmatrix} 1 & 0 & 0 & 0 \\ 0 & e^{\omega} & 0 & 0 \\ 0 & 0 & \mathbb{1}_4 & 0 \\ 0 & e^{\omega}\chi_1 & 0 & 1 \end{pmatrix}, \qquad s = 0. \tag{174}$$

Therefore, the deformation is the combination of a coordinate redefinition coupling the angles $\varphi_1$ and $y^7$ and a rescaling of the $\varphi_1$ coordinate, both described by the GL(7, $\mathbb{R}$) transformation, and TsT transformations between $\varphi_1$ and $y^7$ on one hand, and $\varphi_1$ and $\varphi_2$ on the other. For this reason, the modulus $\chi_1$ is periodic, and taking spinors into account its period can be shown to be $\chi_1 \sim \chi_1 + 4\pi$.[12] We can describe a pure TsT transformation by turning off $\omega$ and $\chi_1$ while keeping a non-vanishing $\beta_1$. To the best of our knowledge, this is the first example of such a Lunin-Maldacena deformation captured among the modes of a consistent truncation down to a gauged maximal supergravity.

Before analysing generalisations of this solution, let us discuss its complete Kaluza-Klein spectrum. It can be obtained by shifting the dimensions (145) and (153) of each physical mode in (144) and (152) as

$$(2\pi p_7)^2 \longrightarrow \left(2\pi p_7 + \frac{1}{2}(q_L + q_R)(\chi_1 + \beta_1)\right)^2$$
$$+ \frac{e^{2\omega}}{4}\left((q_L - q_R) + (q_L + q_R)\left(e^{-2\omega} - \chi_1\beta_1\right) - 4\pi p_7 \beta_1\right)^2 - q_L^2, \tag{175}$$

for $q_L$ and $q_R$ the integer-normalised charges under the bosonic Cartan subalgebra sitting in the superalgebra, taking values

$$j \to \bigoplus_{k=0}^{2j} 2(k - j), \qquad \text{under} \qquad \text{SU}(2) \supset \text{U}(1). \tag{176}$$

Under a shift $\chi_1 \to \chi_1 + 4\pi$, the conformal dimensions following (175) map back to themselves modulo a shift of the $p_7$ number, as expected from the periodicity of the solution (172). For pure TsT deformations the spectrum reads

$$(2\pi p_7)^2 \longrightarrow \left(2\pi p_7 + \frac{1}{2}(q_L + q_R)\beta_1\right)^2 + \frac{1}{4}\left(2q_L - 4\pi p_7 \beta_1\right)^2 - q_L^2. \tag{177}$$

Even though it is not apparent from the Kaluza-Klein spectrum, the construction of this family using SO(7,7) transformations also indicates that the parameter $\beta_1$ is compact in the full string theory.

---

[12]See ref. [4] for an anologous discussion.

Conversely, for $\omega = \beta = 0$ eq. (175) reduces to

$$2\pi p_7 \longrightarrow 2\pi p_7 + \frac{1}{2}(q_L + q_R)\chi_1 \,, \tag{178}$$

following the pattern of other Wilson loop deformations in S-fold compactifications [21, 23, 24, 26].

The spectrum (175) can be used to determine potential supersymmetry enhancements within the three-dimensional moduli space, as well as the stability of the solutions. Supersymmetry enhancement points corresponds to combinations of the moduli such that some gravitini become massless, *i.e.* $\Delta_{(3/2)} = 3/2$. This can first happen within the $3d$ consistent truncation, by leaving the modes with $\Delta_{(3/2)} = 3/2$ unchanged. For the $(\omega, \chi_1, \beta_1)$ deformation, it occurs along the lines

$$\begin{cases} \chi_1 = \pm\sqrt{1 - e^{-2\omega}} \,, \\ \beta_1 = \mp\sqrt{1 - e^{-2\omega}} \,, \end{cases} \quad \text{and} \quad \begin{cases} \chi_1 = \pm\sqrt{1 - e^{-2\omega}} \,, \\ \beta_1 = \pm\sqrt{1 - e^{-2\omega}} \,, \end{cases} \tag{179}$$

where supersymmetry is enhanced to $\mathcal{N} = (0, 4)$ and $\mathcal{N} = (4, 0)$, respectively. Both cases reproduce the solution (168), with $\eta$ as in (169) or $\eta = \cos^2\theta \, d\varphi_1 + \sin^2\theta \, d\varphi_2$, respectively. We further find back the round $\mathcal{N} = (4, 4)$ solution at the origin $\omega = \beta_1 = \chi_1 = 0$. Alternatively some gravitini, originally massive, can become massless under the deformation. This happens here when

$$\begin{cases} \chi_1 = 4\pi q \pm \sqrt{1 - e^{-2\omega}} \,, \\ \beta_1 = \mp\sqrt{1 - e^{-2\omega}} \,, \end{cases} \quad \text{or} \quad \begin{cases} \chi_1 = 4\pi q \pm \sqrt{1 - e^{-2\omega}} \,, \\ \beta_1 = \pm\sqrt{1 - e^{-2\omega}} \,, \end{cases} \quad q \in \mathbb{Z} \,. \tag{180}$$

There are then four massless gravitini, two of them belonging to the multiplets $\ell = \widetilde{\ell} = p_{4,5,6} = 0$ and $p_7 = q$ of (144) and (152), and the two others in the multiplet with opposite charges. Supersymmetry is then enhanced to $\mathcal{N} = (0, 4)$ or $\mathcal{N} = (4, 0)$. The existence of these additionnal enhancement lines reflects the $4\pi$-periodicity in $\chi_1$.

Concerning the stability of the solutions, in both cases it is guaranteed if

$$e^{-\omega} \le \frac{2}{\sqrt{3}} \,, \quad \text{and} \quad (\chi_1 + \pi p_7)^2 \ge \frac{3}{4(1 + e^{2\omega}\beta_1^2)} - e^{-2\omega} \,, \quad \forall p_7 \in \mathbb{Z} \,. \tag{181}$$

There are such wide volumes inside the 3-dimensional parameter space inside which the perturbative stability of the deformations is ensured (see fig. 2). This is for example the case if

$$\begin{cases} e^{-\omega} \le \frac{2}{\sqrt{3}} \,, \\ \beta_1^2 \ge \frac{3}{4} - e^{-2\omega} \,, \end{cases} \tag{182}$$

and $\chi_1$ arbitrary.

The moduli space of this deformation is governed by the metric

$$ds_{\text{Zam.}}^2 = d\omega^2 + \frac{1}{2}e^{2\omega}\left(d\beta_1^2 + d\chi_1^2\right) \,, \tag{183}$$

which corresponds to the leading order in the large-$N$ limit of the Zamolodchikov metric of the holographic conformal manifold. Therefore, there are no infinite distances inside the family (171), and we have neither found them in its further genelisation in (51). This metric was deduced from the scalar kinetic term in eq. (21), with the scalar matrix parameterised by the moduli through the coset representative (171).

In the following, we describe two four-parameter families of solution mixing $S^3 \times M^3$ coordinates with $y^7$. The first family generalises the $\chi_1$ deformations in (174), whilst the second contains pure TsT deformations mixing $S^3 \times M^3$ and $S^1$ for both topologies. Later on, we also discuss deformations that mix $S^3$ with $M^3 = \widetilde{S}^3$.

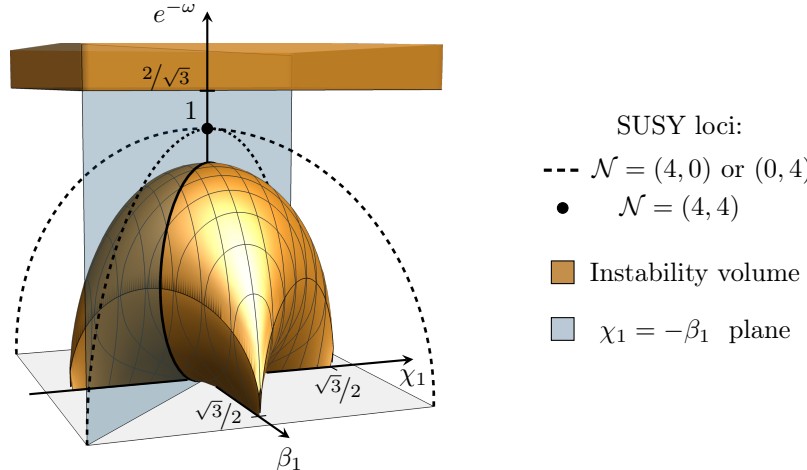

Figure 2: Instability volume (181) for the $(\omega, \beta_1, \chi_1)$ deformations at level $p_7 = 0$. Supersymmetry is enhanced along dashed lines, following eq. (179). Within the $\chi_1 = -\beta_1$ plane (in blue), this reproduces fig. 1 for the $(\omega, \zeta)$ family. For $p_7 \neq 0$, similar instability volumes are repeated with $\pi$ shifts along the $\chi$ axis. Those excluded volumes do not intersect.

## 5.2 Wilson loop deformations

Based on the previous example, we are now led to consider the representative

$$\mathcal{V}_{\tilde{M}}{}^{\tilde{N}} = \exp\left[ -\chi_1 f^{\bar{3}\bar{7}} - \chi_2 f_{\bar{3}}{}^{\bar{7}} - \widetilde{\chi}_1 f^{\bar{6}\bar{7}} - \widetilde{\chi}_2 f_{\bar{6}}{}^{\bar{7}} \right]. \tag{184}$$

At leading order in the large-$N$ expansion, the Zamolodchikov metric on this submanifold is given by

$$ds^2_{\text{Zam.}} = \frac{1}{2} \left( d\chi_1^2 + d\chi_2^2 + d\widetilde{\chi}_1^2 + d\widetilde{\chi}_2^2 \right). \tag{185}$$

The corresponding $10d$ configuration on $S^3 \times \widetilde{S}^3 \times S^1$ can be found in (C.4), and can be described in terms of (162) as

$$\beta = 0, \qquad \rho = \begin{pmatrix} 1 & 0 & 0 & 0 & 0 & 0 & 0 \\ 0 & 1 & 0 & 0 & 0 & 0 & 0 \\ 0 & 0 & 1 & 0 & 0 & 0 & 0 \\ 0 & 0 & 0 & 1 & 0 & 0 & 0 \\ 0 & 0 & 0 & 0 & 1 & 0 & 0 \\ 0 & 0 & 0 & 0 & 0 & 1 & 0 \\ 0 & \chi_1 & -\chi_2 & 0 & \alpha\,\widetilde{\chi}_1 & -\alpha\,\widetilde{\chi}_2 & 1 \end{pmatrix}, \qquad s = 0, \tag{186}$$

whilst the $S^3 \times T^4$ configuration can be found in (C.5) and is described by

$$\beta = 0, \qquad \rho = \begin{pmatrix} 1 & 0 & 0 & 0 & 0 & 0 & 0 \\ 0 & 1 & 0 & 0 & 0 & 0 & 0 \\ 0 & 0 & 1 & 0 & 0 & 0 & 0 \\ 0 & 0 & 0 & 1 & 0 & 0 & 0 \\ 0 & 0 & 0 & 0 & 1 & 0 & 0 \\ 0 & 0 & 0 & 0 & 0 & 1 & 0 \\ 0 & \chi_1 & -\chi_2 & 0 & 0 & \widetilde{\chi}_2 & 1 \end{pmatrix}, \qquad s = 0, \tag{187}$$

which shows that the parameter $\widetilde{\chi}_1$ in (184) is pure gauge in the $S^3 \times T^4$ reduction. In both cases, the deformation consists only in local coordinate redefinitions coupling the angles $\varphi_i, \widetilde{\varphi}_i$

and $y^6$ with $y^7$. They can be interpreted as Wilson loops along the $S^1$ with coordinate $y^7$. Generically, the only remaining isometries are

$$
\begin{aligned}
&\mathrm{SO}(2,2) \times \mathrm{U}(1)_\mathrm{L} \times \mathrm{U}(1)_\mathrm{R} \times \widetilde{\mathrm{U}(1)}_\mathrm{L} \times \widetilde{\mathrm{U}(1)}_\mathrm{R} \times \mathrm{U}(1), && \text{for AdS}_3 \times S^3 \times \widetilde{S}^3 \times S^1, \\
&\mathrm{SO}(2,2) \times \mathrm{U}(1)_\mathrm{L} \times \mathrm{U}(1)_\mathrm{R} \times \mathrm{U}(1)^4, && \text{for AdS}_3 \times S^3 \times \mathrm{T}^4,
\end{aligned}
\tag{188}
$$

and all supersymmetries are broken. The deformed $S^3 \times \widetilde{S}^3 \times S^1$ background can be identified with equation (6.10) in [19].

The spectrum for these solutions is deformed out of (144) through the replacement

$$
2\pi p_7 \longrightarrow 2\pi p_7 - \frac{1}{2}\Big[(q_\mathrm{L} + q_\mathrm{R})\chi_1 + (q_\mathrm{L} - q_\mathrm{R})\chi_2 + \alpha(\widetilde{q}_\mathrm{L} + \widetilde{q}_\mathrm{R})\widetilde{\chi}_1 + \alpha(\widetilde{q}_\mathrm{L} - \widetilde{q}_\mathrm{R})\widetilde{\chi}_2\Big], \tag{189}
$$

in the $S^3 \times \widetilde{S}^3 \times S^1$ case, and out of (152) through

$$
2\pi p_7 \longrightarrow 2\pi p_7 - \frac{1}{2}\Big[(q_\mathrm{L} + q_\mathrm{R})\chi_1 + (q_\mathrm{L} - q_\mathrm{R})\chi_2\Big] - 2\pi p_6\,\widetilde{\chi}_2, \tag{190}
$$

in the $S^3 \times \mathrm{T}^4$ one. They are invariant under

$$
\begin{cases}
\chi_i \to \chi_i + 4\pi q_i, \\
\widetilde{\chi}_i \to \widetilde{\chi}_i + 4\alpha^{-1}\pi \widetilde{q}_i,
\end{cases} \quad \text{for AdS}_3 \times S^3 \times \widetilde{S}^3 \times S^1,
$$

$$
\text{and} \quad
\begin{cases}
\chi_i \to \chi_i + 4\pi q_i, \\
\widetilde{\chi}_2 \to \widetilde{\chi}_2 + \widetilde{q}_2,
\end{cases} \quad \text{for AdS}_3 \times S^3 \times \mathrm{T}^4,
\tag{191}
$$

with $q_i, \widetilde{q}_i \in \mathbb{Z}$. Given the form of the deformations (189) and (190), both spectra are bounded from below by the masses of the round solutions, and pertubative stability is ensured for the entire 4-dimensional family of deformations.

The moduli space of the deformed $S^3 \times \widetilde{S}^3 \times S^1$ solutions enjoys numerous supersymmetry enhancements, as described in ref. [19]. The possible enhancements within the three-dimensional truncation are the following:

$$
\begin{aligned}
&\mathcal{N}=(2,0): && \chi_2 = \chi_1 \pm \alpha(\widetilde{\chi}_1 - \widetilde{\chi}_2), && \mathcal{N}=(0,2): && \chi_2 = -\chi_1 \pm \alpha(\widetilde{\chi}_1 + \widetilde{\chi}_2), \\
&\mathcal{N}=(4,0): && \begin{cases} \chi_2 = \chi_1, \\ \widetilde{\chi}_2 = \widetilde{\chi}_1, \end{cases} && \mathcal{N}=(0,4): && \begin{cases} \chi_2 = -\chi_1, \\ \widetilde{\chi}_2 = -\widetilde{\chi}_1, \end{cases} \\
&\mathcal{N}=(2,2): && \begin{cases} \chi_1 = \pm\alpha\widetilde{\chi}_1, \\ \chi_2 = \pm\alpha\widetilde{\chi}_2, \end{cases} && \mathcal{N}=(2,2): && \begin{cases} \chi_1 = \pm\alpha\widetilde{\chi}_2, \\ \chi_2 = \pm\alpha\widetilde{\chi}_1, \end{cases} \\
&\mathcal{N}=(4,2): && \begin{cases} \chi_1 = \chi_2 = \pm\alpha\widetilde{\chi}_1, \\ \widetilde{\chi}_2 = \widetilde{\chi}_1, \end{cases} && \mathcal{N}=(2,4): && \begin{cases} \chi_1 = -\chi_2 = \pm\alpha\widetilde{\chi}_1, \\ \widetilde{\chi}_2 = -\widetilde{\chi}_1. \end{cases}
\end{aligned}
\tag{192}
$$

SUSY enhancements at higher levels in the $p_7$ tower can be obtained from the periodicities (191). For the $\mathrm{T}^4$ background, supersymmetry is enhanced from $\mathcal{N} = (0,0)$ to $\mathcal{N} = (4,0)$ and $\mathcal{N} = (0,4)$ when

$$
\chi_2 = 4\pi q + \chi_1, \quad \text{and} \quad \chi_2 = 4\pi q - \chi_1, \quad q \in \mathbb{Z}, \tag{193}
$$

respectively. There are then two massless gravitini at level $p_7 = q$ and two other at level $p_7 = -q$, all other charges and SU(2) spins vanishing.

## 5.3 TsT deformations

The deformation

$$
\mathcal{V}_{\bar{M}}{}^{\bar{N}} = \exp\Big[ -\beta_1 f^{\bar{3}}{}_{\bar{7}} - \beta_2 f_{\bar{3}\bar{7}} - \widetilde{\beta}_1 f^{\bar{6}}{}_{\bar{7}} - \widetilde{\beta}_2 f_{\bar{6}\bar{7}} \Big], \tag{194}
$$

recovers a family previously obtained in [19]. As for (185), in this case the Zamolodchikov metric is also flat

$$ds^2_{\text{Zam.}} = \frac{1}{2}\left(d\beta_1^2 + d\beta_2^2 + d\widetilde{\beta}_1^2 + d\widetilde{\beta}_2^2\right). \tag{195}$$

From a $10d$ perspective, this family can be shown to uplift to a type IIB solution which can be constructed through the SO(7,7) transformation in (162), with

$$\beta = \beta_1 \, d\varphi_1 \wedge dy^7 + \alpha\widetilde{\beta}_1 \, d\widetilde{\varphi}_1 \wedge dy^7 - \beta_2 \, d\varphi_2 \wedge dy^7 - \alpha\widetilde{\beta}_2 \, d\widetilde{\varphi}_2 \wedge dy^7,$$

$$\rho = \begin{pmatrix} 1 & 0 & 0 & 0 & 0 & 0 \\ 0 & 1 & 0 & 0 & 0 & -\beta_2 \\ 0 & 0 & \mathbb{1}_2 & 0 & 0 & 0 \\ 0 & 0 & 0 & 1 & 0 & -\alpha^{-1}\widetilde{\beta}_2 \\ 0 & 0 & 0 & 0 & 1 & 0 \\ 0 & 0 & 0 & 0 & 0 & 1 \end{pmatrix}, \qquad s = 0, \tag{196}$$

on the $S^3 \times \widetilde{S}^3 \times S^1$ background, and

$$\beta = \beta_1 \, d\varphi_1 \wedge dy^7 - \beta_2 \, d\varphi_2 \wedge dy^7 + \widetilde{\beta}_2 \, dy^6 \wedge dy^7,$$

$$\rho = \begin{pmatrix} 1 & 0 & 0 & 0 & 0 & 0 \\ 0 & 1 & 0 & 0 & 0 & -\beta_2 \\ 0 & 0 & 1 & 0 & 0 & 0 \\ 0 & 0 & 0 & \mathbb{1}_2 & 0 & 0 \\ 0 & 0 & 0 & 0 & 1 & -\widetilde{\beta}_1 \\ 0 & 0 & 0 & 0 & 0 & 1 \end{pmatrix}, \qquad s = 0, \tag{197}$$

in the $S^3 \times T^4$ case. The detailed $D = 10$ solutions can be respectively found in (C.11) and (C.14). Again, the remaining isometries are those of eq. (188) and $\mathcal{N} = (0,0)$ for generic values of the parameters. These deformations consist in couplings between the angles $\varphi_i$, $\widetilde{\varphi}_i$ and $y^6$ with $y^7$, and TsT transformations between those same angles. Pure TsT deformations are obtained for the couples of cycles $(\varphi_1, y^7)$ and $(\widetilde{\varphi}_1, y^7)$ when $\beta_2 = \widetilde{\beta}_2 = 0$ in the $\widetilde{S}^3 \times S^1$ case and similarly when $\beta_2 = \widetilde{\beta}_1 = 0$ for $T^4$. Alternatively, the solutions can be generated from the SO(7,7) transformation in (162), with

$$\beta = \beta_1 \, d\varphi_1 \wedge dy^7 - \beta_2 \, d\varphi_2 \wedge dy^7 + \alpha\widetilde{\beta}_1 \, d\widetilde{\varphi}_1 \wedge dy^7 - \alpha\widetilde{\beta}_2 \, d\widetilde{\varphi}_2 \wedge dy^7,$$

$$\rho = \begin{pmatrix} \mathbb{1}_2 & 0 & 0 & 0 & 0 \\ 0 & 1 & 0 & 0 & \beta_1 \\ 0 & 0 & \mathbb{1}_2 & 0 & 0 \\ 0 & 0 & 0 & 1 & \alpha^{-1}\widetilde{\beta}_1 \\ 0 & 0 & 0 & 0 & 1 \end{pmatrix}, \qquad s = -d\varphi_1 \wedge \varphi_2 - \alpha^{-2} \, d\widetilde{\varphi}_1 \wedge \widetilde{\varphi}_2, \tag{198}$$

on $S^3 \times \widetilde{S}^3 \times S^1$, and

$$\beta = \beta_1 \, d\varphi_1 \wedge dy^7 - \beta_2 \, d\varphi_2 \wedge dy^7 + \widetilde{\beta}_2 \, dy^6 \wedge dy^7,$$

$$\rho = \begin{pmatrix} \mathbb{1}_2 & 0 & 0 & 0 & 0 \\ 0 & 1 & 0 & 0 & \beta_1 \\ 0 & 0 & \mathbb{1}_2 & 0 & 0 \\ 0 & 0 & 0 & 1 & -\widetilde{\beta}_1 \\ 0 & 0 & 0 & 0 & 1 \end{pmatrix}, \qquad s = -d\varphi_1 \wedge \varphi_2, \tag{199}$$

on $S^3 \times T^4$, giving rise to pure TsT when $\beta_1 = \widetilde{\beta}_1 = 0$. These two solutions differ from (196) and (197) by the gauge choice for the undeformed 2-form in (125) and (131), respectively.

The shift $s$ would be absent in (198) and (199) if the $\sin^2\theta$ and $\sin^2\widetilde{\theta}$ would have been replaced by $-\cos^2\theta$ and $-\cos^2\widetilde{\theta}$ in (125) and (131).

The spectrum of these deformations of $S^3 \times \widetilde{S}^3 \times S^1$ can be obtained from eq. (144) and (145) by shifting

$$
\begin{aligned}
(2\pi p_7)^2 \longrightarrow {}& (2\pi p_7)^2\left(1 + \beta_1^2 + \beta_2^2 + \widetilde{\beta}_1^2 + \widetilde{\beta}_2^2 + (\beta_1\beta_2 + \widetilde{\beta}_1\widetilde{\beta}_2)^2\right) \\
& - 2\pi p_7\Big( (1 + \beta_1\beta_2 + \widetilde{\beta}_1\widetilde{\beta}_2)\big(q_{\mathrm{L}}(\beta_1 + \beta_2) + \alpha\widetilde{q}_{\mathrm{L}}(\widetilde{\beta}_1 + \widetilde{\beta}_2)\big) \\
& \qquad + (-1 + \beta_1\beta_2 + \widetilde{\beta}_1\widetilde{\beta}_2)\big(q_{\mathrm{R}}(\beta_1 - \beta_2) + \alpha\widetilde{q}_{\mathrm{R}}(\widetilde{\beta}_1 - \widetilde{\beta}_2)\big)\Big) \\
& + \frac{1}{4}\big(\beta_1(q_{\mathrm{L}} + q_{\mathrm{R}}) + \beta_2(q_{\mathrm{L}} - q_{\mathrm{R}}) + \alpha\widetilde{\beta}_1(\widetilde{q}_{\mathrm{L}} + \widetilde{q}_{\mathrm{R}}) + \alpha\widetilde{\beta}_2(\widetilde{q}_{\mathrm{L}} - \widetilde{q}_{\mathrm{R}})\big)^2 .
\end{aligned}
\tag{200}
$$

Similarly, the spectrum for the deformed $S^3 \times \mathrm{T}^4$ background follows from eq. (152) and (153) by replacing

$$
\begin{aligned}
(2\pi p_7)^2 \longrightarrow {}& \left(\frac{1}{2}\big((q_{\mathrm{L}} + q_{\mathrm{R}})\beta_1 + (q_{\mathrm{L}} - q_{\mathrm{R}})\beta_2\big) - 2\pi p_6\widetilde{\beta}_2\right)^2 \\
& + (2\pi p_7)^2\left(1 + \beta_1^2 + \beta_2^2 + \widetilde{\beta}_1^2 + \widetilde{\beta}_2^2 + (\beta_1\beta_2 + \widetilde{\beta}_1\widetilde{\beta}_2)^2\right) \\
& - 2\pi p_7\Big(q_{\mathrm{L}}(\beta_1 + \beta_2)(1 + \beta_1\beta_2 + \widetilde{\beta}_1\widetilde{\beta}_2) + q_{\mathrm{R}}(\beta_1 - \beta_2)(-1 + \beta_1\beta_2 + \widetilde{\beta}_1\widetilde{\beta}_2)\Big) \\
& + 8\pi^2 p_6 p_7\left(\widetilde{\beta}_1 + \beta_1\beta_2\widetilde{\beta}_2 + \widetilde{\beta}_1\widetilde{\beta}_2^2\right).
\end{aligned}
\tag{201}
$$

For $p_7 = 0$, these turn out to be the exact same spectra as the ones for the $\chi$'s in sec. 5.2 up to matching $\chi_i \to \beta_i$ and $\widetilde{\chi}_i \to \widetilde{\beta}_i$. The solutions then enjoy the same supersymmetry enhancements as the $\chi$ deformations restricted to the $3d$ consistent truncation, see eq. (192) and (193) (for $q = 0$).

For $\widetilde{\beta}_1 = \widetilde{\beta}_2 = 0$, the solution is stable for any value of $\beta_1$ and $\beta_2$. The converse is not true, however, with instabilities present when $\beta_1 = \beta_2 = 0$ and $\widetilde{\beta}_1$ and $\widetilde{\beta}_2$ are non-vanishing. This apparent inequity is not in tension with the interchangeability between the two spheres, given that is a symmetry of the equations of motion only if the $S^1$ is also rescaled, as can be seen in (124). This rescaling can be parameterised by the modulus $\sigma_7$ in (51), and the configuration is then invariant under the transformation

$$
\beta_i \mapsto \widetilde{\beta}_i, \qquad \widetilde{\beta}_i \mapsto \beta_i, \qquad e^{-\sigma_7} \mapsto \alpha^{-1}e^{-\sigma_7}.
\tag{202}
$$

The precise stability range when the four $\beta$s are turned on needs further study, but perturbative stability is guaranteed in certain subregions by the existence of the supersymmetric loci.

## 5.4 Mixing of $S^3$ and $\widetilde{S}^3$

We now analyse the deformation

$$
\mathcal{V}_{\bar{M}}{}^{\bar{N}} = \exp\left[-\Xi_2 f^{\bar{3}}{}_{\bar{6}} - \Xi_4 f_{\bar{3}\bar{6}}\right],
\tag{203}
$$

with Zamolodchikov metric

$$
\mathrm{d}s_{\mathrm{Zam.}}^2 = \frac{1}{2}\left(\mathrm{d}\Xi_2^2 + \mathrm{d}\Xi_4^2\right).
\tag{204}
$$

On the $S^3 \times \mathrm{T}^4$ background, this is analogous to the deformations in sec. 5.2 and 5.3 up to relabelling of the torus coordinates. On the other hand, on the $S^3 \times \widetilde{S}^3 \times S^1$ background it corresponds to mixing the coordinates on the two spheres through the SO(7,7) transformation (162)

with

$$\beta = \alpha\left(-\Xi_2\,\mathrm{d}\varphi_1 + \Xi_4\,\mathrm{d}\varphi_2\right)\wedge \mathrm{d}\widetilde{\varphi}_2,$$

$$\rho = \begin{pmatrix} 1 & 0 & 0 & 0 & 0 & 0 & 0 \\ 0 & 1 & 0 & 0 & 0 & \alpha\Xi_4 & 0 \\ 0 & 0 & 1 & 0 & 0 & 0 & 0 \\ 0 & 0 & 0 & 1 & 0 & 0 & 0 \\ 0 & \alpha^{-1}\Xi_2 & -\alpha^{-1}\Xi_4 & 0 & 1 & 0 & \\ 0 & 0 & 0 & 0 & 0 & 1 & 0 \\ 0 & 0 & 0 & 0 & 0 & 0 & 1 \end{pmatrix}, \qquad s = 0, \qquad (205)$$

involving in particular TsT transformations coupling each $\varphi_i$ with $\widetilde{\varphi}_2$. The explicit solution in $D = 10$ can be found in eq. (C.20). Generically, the isometries are broken down to (188) and there is no remaining supersymmetry. At the points

$$\Xi_2 = -\Xi_4 = \pm\frac{2}{\alpha}, \quad \text{and} \quad \Xi_2 = \Xi_4 = \pm\frac{2}{\alpha}, \qquad (206)$$

SUSY enhances to $\mathcal{N} = (2,0)$ and $\mathcal{N} = (0,2)$, respectively. This can be observed from the deformed spectrum, given by eq. (144) and (145) by shifting

$$(2\pi p_7)^2 \longrightarrow (2\pi p_7)^2 + \frac{1}{4}\left(q_{\mathrm{L}}(\Xi_2 + \Xi_4) + q_{\mathrm{R}}(\Xi_4 - \Xi_2)\right)^2$$
$$+ \frac{\alpha^2}{4}(\widetilde{q}_{\mathrm{L}} - \widetilde{q}_{\mathrm{R}})\left(\widetilde{q}_{\mathrm{L}}(\Xi_4 - \Xi_2)^2 - \widetilde{q}_{\mathrm{R}}(\Xi_2 + \Xi_4)^2 + (\widetilde{q}_{\mathrm{L}} - \widetilde{q}_{\mathrm{R}})(\Xi_2\Xi_4)^2\right) \qquad (207)$$
$$- \frac{\alpha}{2}q_{\mathrm{L}}(\Xi_2 + \Xi_4)\left((\widetilde{q}_{\mathrm{L}} - \widetilde{q}_{\mathrm{R}})\Xi_2\Xi_4 - 2\widetilde{q}_{\mathrm{R}}\right) - \frac{\alpha}{2}q_{\mathrm{R}}(\Xi_4 - \Xi_2)\left((\widetilde{q}_{\mathrm{L}} - \widetilde{q}_{\mathrm{R}})\Xi_2\Xi_4 - 2\widetilde{q}_{\mathrm{L}}\right).$$

Regarding the perturbative stability of these deformations, analysis of the lowest Kaluza-Klein levels indicates that the region in parameter space with tachyonic modes can get arbitrary close to the SUSY lines (206). This feature is not apparent at low Kaluza-Klein levels ($\ell + \widetilde{\ell} < 1/2$), but already at level $(1/2, 1/2)$ we find modes whose region of instability ends on the SUSY enhancement lines, where the modes saturate the BF bound. This can be observed in fig. 3. This analysis is not conclusive about the region of stability around the origin $\Xi_1 = \Xi_2 = 0$.

## 6 Worldsheet and holographic descriptions

The deformed solutions built in the previous section are all purely NSNS and can therefore be described from the point of view of the worldsheet action. In this section, we start by reviewing the WZW models relevant to the round $\mathrm{AdS}_3 \times S^3 \times \mathrm{T}^4$ and $\mathrm{AdS}_3 \times S^3 \times S^3 \times S^1$ worldsheet realizations. We then show that the deformations built in sec. 5 can be described in this formulation by $J\bar{J}$ deformations.

The group manifolds for the $\mathcal{N} = 1$ superconformal WZW models we consider are [14, 15]

$$\begin{aligned} \mathrm{SL}(2,\mathbb{R}) \times \mathrm{SU}(2) \times \mathrm{U}(1)^4, &\qquad \text{for } \mathrm{AdS}_3 \times S^3 \times \mathrm{T}^4, \\ \mathrm{SL}(2,\mathbb{R}) \times \mathrm{SU}(2) \times \widetilde{\mathrm{SU}(2)} \times \mathrm{U}(1), &\qquad \text{for } \mathrm{AdS}_3 \times S^3 \times \tilde{S}^3 \times S^1. \end{aligned} \qquad (208)$$

The undeformed WZW action at level $k \in \mathbb{N}$ for each factor is given by [68, 69]

$$S = \frac{k}{4\pi}\int_\Sigma dzd\bar{z}\,\mathrm{Tr}\left(\partial g\,\bar{\partial}g^{-1}\right) + \frac{k}{6\pi}\int_\Omega \mathrm{d}^3x\,\epsilon^{ijk}\,\mathrm{Tr}\left[(g^{-1}\partial_i g)(g^{-1}\partial_j g)(g^{-1}\partial_k g)\right], \qquad (209)$$

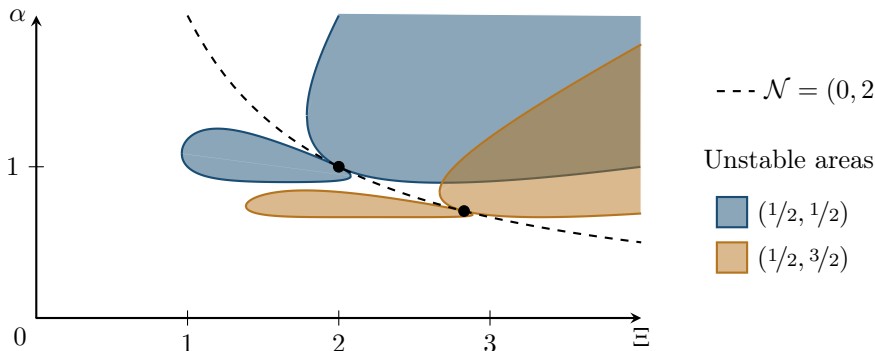

Figure 3: Parameter space for the deformation (203) with $\Xi_1 = \Xi_2 = \Xi$. Supersymmetry is enhanced to $\mathcal{N} = (2,0)$ along the dashed line (see eq. (206)) and totally broken otherwise for non-vanishing $\Xi$. There are subregions of the parameter space for which some modes become unstable, as presented at levels $(\ell, \widetilde{\ell}) = (1/2, 1/2)$ and $(1/2, 3/2)$ for $p_7 = 0$. At the border of these regions, the modes have masses saturating the BF bound. At level $(1/2, \widetilde{\ell})$, the potentially unstable modes are those arising from the deformation of the $SO(4) \times \widetilde{SO(4)}$ scalars $(3/2, \widetilde{\ell}, 1/2, \widetilde{\ell}+1)$ and $(1/2, \widetilde{\ell}+1, 1/2, \widetilde{\ell}+1)$ with extremal charges.

with $\Omega$ such that $\Sigma = \partial \Omega$. As customary, the action is written in Euclidean signature and in terms of a complex coordinate $z$ with the shorthands $\partial = \partial_z$ and $\bar{\partial} = \partial_{\bar{z}}$. The entire model is superconformal if the levels of the different factors in (208) are related as

$$\frac{1}{k_0} = \frac{1}{k} + \frac{1}{\tilde{k}}, \tag{210}$$

for $k_0$ the $SL(2, \mathbb{R})$ level and $k$, $\tilde{k}$ corresponding to the spheres. The $AdS_3 \times S^3 \times T^4$ case is given by the limit $1/\tilde{k} = 0$, which in terms of the geometric radii

$$k_0 = 4\pi^2 \ell_{AdS}^2, \qquad k = 4\pi^2 \ell_{S^3}^2, \qquad \tilde{k} = 4\pi^2 \ell_{\tilde{S}^3}^2 = 4\pi^2 \alpha^{-2} \ell_{S^3}^2, \tag{211}$$

corresponds to the limit $\alpha \to 0$. With these identifications, the level matching condition (210) reproduces the supergravity result (48) with normalisation $\ell_{S^3} = 1$.

As the deformations we consider preserve the conformal algebra, in the following we will omit the $SL(2, \mathbb{R})$ factors. We parameterise the $SU(2)$ elements in terms of Euler angles as

$$g = e^{i(\varphi_1 + \varphi_2)\sigma_3/2} e^{i\theta\sigma_1} e^{i(\varphi_1 - \varphi_2)\sigma_3/2}, \tag{212}$$

with $\sigma_i$ the Pauli matrices, and similarly for $\widetilde{SU(2)}$ in terms of the tilded angles on $\tilde{S}^3$. For the circle directions, the representative is simply

$$g_a = e^{2\pi i y^a}. \tag{213}$$

The angles are here understood as fields on the worldsheet depending on the coordinates $z, \bar{z}$. In the $SU(2) \times U(1)^4$ case, eq. (209) reads

$$S_{SU(2) \times T^4} = \frac{k}{2\pi} \int_{\Sigma} \left( \partial\theta\bar{\partial}\theta + \cos^2\theta\, \partial\varphi_1\bar{\partial}\varphi_1 + \sin^2\theta\, \partial\varphi_2\bar{\partial}\varphi_2 + \sin^2\theta\left(\partial\varphi_1\bar{\partial}\varphi_2 - \partial\varphi_2\bar{\partial}\varphi_1\right)\right)$$

$$+ \frac{1}{2\pi} \int_{\Sigma} \delta_{ab}\partial y^a\bar{\partial}y^b. \tag{214}$$

In this theory, the currents associated with the translations in $y^a$, as well as the $\text{SU}(2)_\text{L} \times \text{SU}(2)_\text{R}$ currents

$$
\begin{aligned}
j_1^\text{L} + i j_2^\text{L} &= \left[\partial\theta - \tfrac{i}{2}\sin 2\theta(\partial\varphi_1 + \partial\varphi_2)\right]e^{i(\varphi_1-\varphi_2)}, &\quad j_3^\text{L} &= \cos^2\theta\,\partial\varphi_1 - \sin^2\theta\,\partial\varphi_2, \\
j_1^\text{R} + i j_2^\text{R} &= \left[\bar\partial\theta + \tfrac{i}{2}\sin 2\theta(\bar\partial\varphi_1 - \bar\partial\varphi_2)\right]e^{-i(\varphi_1+\varphi_2)}, &\quad j_3^\text{R} &= \cos^2\theta\,\bar\partial\varphi_1 + \sin^2\theta\,\bar\partial\varphi_2,
\end{aligned}
\tag{215}
$$

are (anti-)holomorphic per the equations of motion,

$$
\bar\partial\partial y^a = 0, \qquad \text{and} \qquad \bar\partial j_i^\text{L} = \partial j_i^\text{R} = 0. \tag{216}
$$

This conservation is preserved at the quantum level, and the algebraic structure determines the current algebra OPE [68]

$$
j_i^\text{L}(z)j_j^\text{L}(w) \sim \frac{k\delta_{ij}}{(z-w)^2} + \sum_k \frac{f_{ij}{}^k j_k^\text{L}(z)}{z-w}, \tag{217}
$$

with $k$ as in (214) and $f_{ij}{}^k$ the SU(2) structure constants, and similarly for the translations and anti-holomorphic currents.

Similarly, for $\text{SU}(2) \times \widetilde{\text{SU}(2)} \times \text{U}(1)$, the action reads

$$
\begin{aligned}
S_{\text{SU}(2)^2\times\text{U}(1)} = {}& \frac{k}{2\pi}\int_\Sigma \partial\theta\bar\partial\theta + \cos^2\theta\,\partial\varphi_1\bar\partial\varphi_1 + \sin^2\theta\,\partial\varphi_2\bar\partial\varphi_2 + \sin^2\theta\big(\partial\varphi_1\bar\partial\varphi_2 - \partial\varphi_2\bar\partial\varphi_1\big) \\
&+ \frac{\tilde k}{2\pi}\int_\Sigma \partial\tilde\theta\bar\partial\tilde\theta + \cos^2\tilde\theta\,\partial\tilde\varphi_1\bar\partial\tilde\varphi_1 + \sin^2\tilde\theta\,\partial\tilde\varphi_2\bar\partial\tilde\varphi_2 + \sin^2\tilde\theta\big(\partial\tilde\varphi_1\bar\partial\tilde\varphi_2 - \partial\tilde\varphi_2\bar\partial\tilde\varphi_1\big) \\
&+ \frac{1}{2\pi}\int_\Sigma \partial y^7\bar\partial y^7,
\end{aligned}
\tag{218}
$$

with now, besides the $y^7$ translations, a group $\text{SU}(2)^2 \times \widetilde{\text{SU}(2)}^2$ worth of symmetries generated by (215) and their tilded counterparts. In both eq. (214) and (218), the internal components of the metric and $B$ field can be read off from

$$
S = \int_\Sigma \partial y^i \bar\partial y^j\, E_{ij}, \tag{219}
$$

with $E_{ij} = (\hat g_s)_{ij} + \hat B_{ij}$.

## 6.1 Deformations around generic points

For every solution in sec. 5, the worldsheet action is defined by eq. (219). Let us now show that infinitesimal deformations around generic points of the families discussed in that section are current-current deformations of this worldsheet action. The currents that participate in the deformations can be expressed in terms of $E_{ij}$ as [70]

$$
j_p = \partial y^i E_{ij} k_p^j, \qquad \bar j_p = k_p^i E_{ij} \bar\partial y^j, \tag{220}
$$

where $k_p$ are abelian Killing vectors of the round geometries in eq. (208).[13] In particular, for the cases under consideration,

$$
\begin{aligned}
&k_{\varphi_1} = \partial_{\varphi_1}, \quad k_{\varphi_2} = \partial_{\varphi_2}, \quad k_{y^a} = \partial_{y^a}, &&\text{for } \text{AdS}_3\times S^3\times \text{T}^4, \\
&k_{\varphi_1} = \partial_{\varphi_1}, \quad k_{\varphi_2} = \partial_{\varphi_2}, \quad k_{\tilde\varphi_1} = \partial_{\tilde\varphi_1}, \quad k_{\tilde\varphi_2} = \partial_{\tilde\varphi_2}, \quad k_{y^7} = \partial_{y^7}, &&\text{for } \text{AdS}_3\times S^3\times \tilde S^3\times S^1.
\end{aligned}
\tag{221}
$$

---

[13]These Killing vectors leave the metric invariant and change the $B$-field only by a gauge transformation, i.e. $\mathcal{L}_{k_p} g = 0$ and $\mathcal{L}_{k_p} B = d\lambda$.

For the deformed solutions in sec. 5, the $E_{ij}$ matrix can be read off from (158) and (159). Infinitesimal variations of the parameters around the deformed solutions can be expressed in terms of the currents in (220). For instance, for the three-parameter family of solutions in (172) the relevant currents are

$$
\begin{aligned}
j_{\varphi_1} &= \frac{\left(\partial\varphi_1 + (\chi_1 + \beta_1)\partial y^7\right)\cos^2\theta - \sin^2\theta\,\partial\varphi_2}{1 + \left(e^{-2\omega} + \beta_1^2 - 1\right)\cos^2\theta}\,, \\
\bar{j}_{\varphi_1} &= \frac{\left(\bar\partial\varphi_1 + (\chi_1 - \beta_1)\bar\partial y^7\right)\cos^2\theta + \sin^2\theta\,\bar\partial\varphi_2}{1 + \left(e^{-2\omega} + \beta_1^2 - 1\right)\cos^2\theta}\,, \\
j_{y^7} &= \frac{\left(1 + \left(e^{-2\omega} + \chi_1^2 - 1\right)\cos^2\theta\right)\partial y^7 + (\chi_1 - \beta_1)\left(\cos^2\theta\,\partial\varphi_1 - \sin^2\theta\,\partial\varphi_2\right)}{1 + \left(e^{-2\omega} + \beta_1^2 - 1\right)\cos^2\theta}\,, \\
\bar{j}_{y^7} &= \frac{\left(1 + \left(e^{-2\omega} + \chi_1^2 - 1\right)\cos^2\theta\right)\bar\partial y^7 + (\chi_1 + \beta_1)\left(\cos^2\theta\,\bar\partial\varphi_1 + \sin^2\theta\,\bar\partial\varphi_2\right)}{1 + \left(e^{-2\omega} + \beta_1^2 - 1\right)\cos^2\theta}\,,
\end{aligned}
\tag{222}
$$

which respectively reduce to $j_3^{\mathrm{L}}$, $j_3^{\mathrm{R}}$, $\partial y^7$ and $\bar\partial y^7$ in (215) when $\omega = \beta_1 = \chi_1 = 0$. Upon the equations of motion stemming from eq. (219) for the background (172), these currents are (anti-)holomorphic (*c.f.* app. C)

$$
\bar\partial j_{\varphi_1} = \bar\partial j_{y^7} = 0\,, \qquad \partial\bar{j}_{\varphi_1} = \partial\bar{j}_{y^7} = 0\,,
\tag{223}
$$

and the results at $(\omega + \delta\omega, \beta_1 + \delta\beta_1, \chi_1 + \delta\chi_1)$ and $(\omega, \beta_1, \chi_1)$ for the family in (172) are related as

$$
\delta E = 2(e^{-2\omega}\delta\omega - \chi_1\delta\chi_1)j_{\varphi_1}\otimes\bar{j}_{\varphi_1} + (\delta\chi_1 - \delta\beta_1)j_{\varphi_1}\otimes\bar{j}_{y^7} + (\delta\chi_1 + \delta\beta_1)j_{y^7}\otimes\bar{j}_{\varphi_1}\,,
\tag{224}
$$

to linear order in $(\delta\omega, \delta\beta_1, \delta\chi_1)$ and for both cases in (208). Regarding the ten-dimensional dilaton, it changes as a compensator of the variation of the metric so as to keep the generalised dilaton $\hat{d} = \hat\Phi - \frac{1}{4}\log\hat{g}_s$ invariant under the deformations, as required by marginality [69, 71].

The linear variation $\delta E$ for all the instances discussed in sec. 5 can be expressed in terms of products of currents which are (anti-)holomorphic upon imposing the equations of motion, as can be found in app. C. For the $4\chi$ and $4\beta$ families in (184) and (194) the infinitesimal variations read

$$
\begin{aligned}
\delta E &= (\delta\chi_1 + \delta\chi_2)j_{\varphi_1}\otimes\bar\partial y^7 + (\delta\chi_1 - \delta\chi_2)\partial y^7\otimes\bar{j}_{\varphi_1} \\
&\quad + \alpha(\delta\tilde\chi_1 + \delta\tilde\chi_2)j_{\tilde\varphi_1}\otimes\bar\partial y^7 + \alpha(\delta\tilde\chi_1 - \delta\tilde\chi_2)\partial y^7\otimes\bar{j}_{\tilde\varphi_1}\,,
\end{aligned}
\tag{225}
$$

$$
\begin{aligned}
\delta E &= -(\delta\beta_1 + \delta\beta_2)j_{\varphi_1}\otimes\bar{j}_{y^7} + (\delta\beta_1 - \delta\beta_2)j_{y^7}\otimes\bar{j}_{\varphi_1} \\
&\quad - \alpha(\delta\tilde\beta_1 + \delta\tilde\beta_2)j_{\tilde\varphi_1}\otimes\bar{j}_{y^7} + \alpha(\delta\tilde\beta_1 - \delta\tilde\beta_2)j_7\otimes\bar{j}_{\tilde\varphi_1} - 2(\beta_2\delta\beta_2 + \tilde\beta_2\delta\tilde\beta_2)j_{y^7}\otimes\bar{j}_{y^7}\,,
\end{aligned}
\tag{226}
$$

for the $\mathrm{AdS}_3 \times S^3 \times \tilde S^3 \times S^1$ topology, and

$$
\delta E = (\delta\chi_1 + \delta\chi_2)j_{\varphi_1}\otimes\bar\partial y^7 + (\delta\chi_1 - \delta\chi_2)\partial y^7\otimes\bar{j}_{\varphi_1} + \delta\tilde\chi_2(j_{y^6}\otimes\bar\partial y^7 + \partial y^7\otimes\bar{j}_{y^6})\,,
\tag{227}
$$

$$
\begin{aligned}
\delta E &= -(\delta\beta_1 + \delta\beta_2)j_{\varphi_1}\otimes\bar{j}_{y^7} + (\delta\beta_1 - \delta\beta_2)j_{y^7}\otimes\bar{j}_{\varphi_1} \\
&\quad - (\delta\tilde\beta_1 + \delta\tilde\beta_2)j_{y^6}\otimes\bar{j}_{y^7} - (\delta\tilde\beta_1 - \delta\tilde\beta_2)j_{y^7}\otimes\bar{j}_{y^6} - 2(\beta_2\delta\beta_2 + \tilde\beta_1\delta\tilde\beta_1)j_{y^7}\otimes\bar{j}_{y^7}\,,
\end{aligned}
\tag{228}
$$

for the $\mathrm{AdS}_3 \times S^3 \times \mathrm{T}^4$. In (225) and (227), the forms $\partial y^7$ and $\bar\partial y^7$ have definite chirality (i.e. the field $y^7$ is free), and gauge transformations of the $B$ field have been omitted. Finally, the $\Xi$ deformation (203) gives rise to

$$
\begin{aligned}
\delta E &= (\delta\Xi_2 + \delta\Xi_4)\left(j_{\varphi_1} - \alpha\Xi_4 j_{\tilde\varphi_2}\right)\otimes\left(\alpha\bar{j}_{\tilde\varphi_1} + \Xi_4\bar{j}_{\varphi_2}\right) + (\delta\Xi_2 - \delta\Xi_4)\left(\alpha j_{\tilde\varphi_1} + \Xi_4 j_{\varphi_2}\right)\otimes\left(\bar{j}_{\varphi_1} - \alpha\Xi_4\bar{j}_{\tilde\varphi_2}\right) \\
&\quad - 2\Xi_2\delta\Xi_2\left(j_{\varphi_1} - \alpha\Xi_4 j_{\tilde\varphi_2}\right)\otimes\left(\bar{j}_{\varphi_1} - \alpha\Xi_4\bar{j}_{\tilde\varphi_2}\right)\,,
\end{aligned}
\tag{229}
$$

up to gauge transformations of $B$. In this case, the conservation laws are

$$\bar{\partial}\left(j_{\varphi_1} - \alpha \Xi_4 j_{\tilde{\varphi}_2}\right) = \bar{\partial}\left(\alpha j_{\tilde{\varphi}_1} + \Xi_4 j_{\varphi_2}\right) = 0, \quad \partial\left(\alpha \bar{j}_{\tilde{\varphi}_1} + \Xi_4 \bar{j}_{\varphi_2}\right) = \partial\left(\bar{j}_{\varphi_1} - \alpha \Xi_4 \bar{j}_{\tilde{\varphi}_2}\right) = 0, \quad (230)$$

even though the currents $j_{\varphi_1}$, $j_{\tilde{\varphi}_2}$, etc are not separately conserved.

Around the origin, the deformations in (224)–(228) simplify drastically and can be written in terms of the WZW currents in (215). For the $(\omega, \beta_1, \chi_1)$ family, we get

$$(d\hat{s}_s^2 + \hat{B}_{(2)}) = (d\hat{s}_s^2 + \hat{B}_{(2)})_0 + 2\delta\omega\, j_3^{\mathrm{L}} \otimes j_3^{\mathrm{R}} + (\delta\chi_1 - \delta\beta_1)\, j_3^{\mathrm{L}} \otimes \bar{\partial} y^7 + (\delta\chi_1 + \delta\beta_1)\, \partial y^7 \otimes j_3^{\mathrm{R}}. \quad (231)$$

The $4\chi$ and $4\beta$ deformations read

$$\begin{aligned}
(d\hat{s}_s^2 + \hat{B}_{(2)}) = (d\hat{s}_s^2 + \hat{B}_{(2)})_0 &+ (\delta\chi_1 + \delta\chi_2)\, j_3^{\mathrm{L}} \otimes \bar{\partial} y^7 + (\delta\chi_1 - \delta\chi_2)\, \partial y^7 \otimes j_3^{\mathrm{R}} \\
&+ \alpha^{-1}(\delta\tilde{\chi}_1 + \delta\tilde{\chi}_2)\, \tilde{j}_3^{\mathrm{L}} \otimes \bar{\partial} y^7 + \alpha^{-1}(\delta\tilde{\chi}_1 - \delta\tilde{\chi}_2)\, \partial y^7 \otimes \tilde{j}_3^{\mathrm{R}},
\end{aligned} \quad (232)$$

$$\begin{aligned}
(d\hat{s}_s^2 + \hat{B}_{(2)}) = (d\hat{s}_s^2 + \hat{B}_{(2)})_0 &- (\delta\beta_1 + \delta\beta_2)\, j_3^{\mathrm{L}} \otimes \bar{\partial} y^7 + (\delta\beta_1 - \delta\beta_2)\, \partial y^7 \otimes j_3^{\mathrm{R}} \\
&- \alpha^{-1}(\delta\tilde{\beta}_1 + \delta\tilde{\beta}_2)\, \tilde{j}_3^{\mathrm{L}} \otimes \bar{\partial} y^7 + \alpha^{-1}(\delta\tilde{\beta}_1 - \delta\tilde{\beta}_2)\, \partial y^7 \otimes \tilde{j}_3^{\mathrm{R}},
\end{aligned} \quad (233)$$

for $\mathrm{AdS}_3 \times S^3 \times \tilde{S}^3 \times S^1$, and

$$\begin{aligned}
(d\hat{s}_s^2 + \hat{B}_{(2)}) = (d\hat{s}_s^2 + \hat{B}_{(2)})_0 &+ (\delta\chi_1 + \delta\chi_2)\, j_3^{\mathrm{L}} \otimes \bar{\partial} y^7 + (\delta\chi_1 - \delta\chi_2)\, \partial y^7 \otimes j_3^{\mathrm{R}} \\
&+ \delta\tilde{\chi}_2(\partial y^6 \otimes \bar{\partial} y^7 + \partial y^7 \otimes \bar{\partial} y^6),
\end{aligned} \quad (234)$$

$$\begin{aligned}
(d\hat{s}_s^2 + \hat{B}_{(2)}) = (d\hat{s}_s^2 + \hat{B}_{(2)})_0 &- (\delta\beta_1 + \delta\beta_2)\, j_3^{\mathrm{L}} \otimes \bar{\partial} y^7 + (\delta\beta_1 - \delta\beta_2)\, \partial y^7 \otimes j_3^{\mathrm{R}} \\
&- \delta\tilde{\beta}_1(\partial y^6 \otimes \bar{\partial} y^7 + \partial y^7 \otimes \bar{\partial} y^6),
\end{aligned} \quad (235)$$

in the $\mathrm{AdS}_3 \times S^3 \times \mathrm{T}^4$ case. For each topology, these expressions match at the linearised level up to a straightforward redefinition of the parameters given by

$$\delta\chi_1 \mapsto -\delta\beta_2, \qquad \delta\chi_2 \mapsto -\delta\beta_1, \qquad \delta\tilde{\chi}_1 \mapsto -\delta\tilde{\beta}_2, \qquad \delta\tilde{\chi}_2 \mapsto -\delta\tilde{\beta}_1. \quad (236)$$

The $\Xi$ deformation (203) also simplifies to

$$(d\hat{s}_s^2 + \hat{B}_{(2)}) = (d\hat{s}_s^2 + \hat{B}_{(2)})_0 + \alpha^{-1}(\delta\Xi_2 + \delta\Xi_4)\, j_3^{\mathrm{L}} \otimes \tilde{j}_3^{\mathrm{R}} + \alpha^{-1}(\delta\Xi_2 - \delta\Xi_4)\, \tilde{j}_3^{\mathrm{L}} \otimes j_3^{\mathrm{R}}. \quad (237)$$

## 6.2 Comments on the CFT dual

For all cases (224)–(229), the deformations are described by products of commuting (anti-)holomorphic currents, and are thus exactly marginal [7]. Further checks of this exact marginality can be already be made in supergravity out of the three-point functions for the Kaluza-Klein modes following ref. [37] so as to study the vanishing of the beta-functions in conformal perturbation theory [8]. We plan to return to this question in the future.

From a holographic perspective, the identification of the WZW currents around the origin in (231)–(237) allows us to conjecture that the marginal operators in the holographic conformal field theories are also of $j\bar{j}$ type. In the symmetric orbifold theories,

$$\mathrm{Sym}^N(M^4), \quad (238)$$

one can identify two SU(2) factors corresponding to the left- and right-moving currents associated to the R-symmetry, and extra flavour symmetries realised on every copy of $M^4 = \mathrm{U}(1)^4$ or $M^4 = \mathrm{SU}(2) \times \mathrm{U}(1)$. The relevant "single trace" operators [72] on the orbifold are given by the projection

$$\mathcal{O} \sim \sum_{k}^{N} (j\bar{j})_k, \quad (239)$$

with $k$ an index on each of the copies.

# 7 Discussion

This note focused on the construction and study of new marginal deformations of the $AdS_3 \times S^3 \times S^3 \times S^1$ and $AdS_3 \times S^3 \times T^4$ solutions of heterotic and type IIB supergravities using exceptional field theory. These solutions are of particular relevance for the $AdS_3/CFT_2$ correspondence and we built a general framework that unifies the description of those backgrounds in both theories. The rich structure of marginal deformations thus revealed is in sharp contrast with what happens in higher dimensions. These deformations include Lunin-Maldacena TsT transformations and Wilson loops among more general deformations. However, our search of moduli is far from being exhaustive and needs to be generalised, for example to include mixing of TsT transformation between the sphere and multiple directions on the torus, or couplings between TsT and Wilson loop deformations. The integrability of the WZW models describing the round solutions [73] allows to describe our deformed solutions as Yang-Baxter deformations along the lines of [74]. Integrability could provide powerful tools to study these solutions in more detail.

All the deformation parameters we considered belong to three-dimensional consistent truncations. This makes it possible to use the ExFT's Kaluza-Klein spectrometer to compute the effect of the deformations on the full Kaluza-Klein tower of excitations. We used these deformation-dependent spectra to study the perturbative stability of some non-supersymmetric vacua, and demonstrated that there is a vast subregion of parameter space where the solutions are free from perturbative instabilities. The complete stability of these solutions has to be tested against potential non-perturbative decay channels, as brane-jet instabilities [75–77] and nucleations of bubbles [78–83]. This would require building their associated brane configurations. It would also be very interesting to study the existence of positive energy theorems in the lines of ref. [84].

Among the directions in the conformal manifold, the possibility of describing TsT deformations acting on spheres in a consistent truncation is a three-dimensional peculiarity, as in higher-dimensions the moduli triggering those transformations sit within higher Kaluza-Klein levels [20,57,58]. Similarly to what happens for Wilson loop deformations, even though TsT seems to be composed of symmetry transformations of string theory (T duality, shifts in coordinates and T duality), our results demonstrate that such deformations affect the Kaluza-Klein spectrum. This is because the coordinate shift couples directions with non-compatible periodicities, and therefore the transformations are not globally well-defined for generic values of the deformation parameters. It would be very interesting to study if these three-dimensional results could provide insights on the Kaluza-Klein spectra for TsT deformations in higher dimensions.

The transformations in (51) do not excite RR fluxes. We took advantage of this property to describe them as current-current couplings of the WZW worldsheet actions describing the $AdS_3 \times S^3 \times S^3 \times S^1$ and $AdS_3 \times S^3 \times T^4$ backgrounds. This suggests that the holographically dual deformations are single-trace $J\bar{J}$. It will be interesting to analyse these deformed holographic duals, and the fact that some of these deformations preserve some supersymmetries for both left- and right-movers (see *e.g.* (192)) suggests that some subfamilies should be amenable to the CFT analysis. Nevertheless, it would also be of interest to study whether some new deformations could also excite RR fluxes. Given that U duality encompasses both T and S dualities, the ExFT framework can also be used to generate transformations that excite them. Of particular interest are the S dual rotation mapping the NS5-F1 and D1-D5 configurations, as well as the S duality orbits of the pure NSNS deformations described above. If such deformations belong to a three-dimensional consistent truncation to a gauged maximal supergravity, one should expect to find them among the $E_{8(8)}$ generators in the **128** representation of $SO(8,8)$ in the decomposition (26). Describing the uplift to ten-dimensional supergravity would then require the construction of the full dictionary between $E_{8(8)}$ ExFT and type IIB supergravity,

generalising eq. (62). Such new deformations mixing NSNS and RR fluxes could make contact with the recent families of AdS$_3$ solutions constructed in ref. [85–87].

Given that both the AdS$_3 \times S^3 \times S^3 \times S^1$ and AdS$_3 \times S^3 \times$ T$^4$ spectra feature massless scalar modes at higher Kaluza-Klein levels, one could wonder if these backgrounds also feature moduli outside of their consistent truncations to $3d$. This could be investigated by applying the generalised geometry techniques developed in ref. [88–90]. Similar methods have recently been applied in ExFT to relevant deformations in ref. [91].

# Acknowledgments

We are grateful to Michele Galli, Georgios Itsios and Emanuel Malek for collaboration on related projects. We would also like to thank Alexandre Belin, Nikolay Bobev, Riccardo Borsato, Anamaria Font, Henning Samtleben and Linus Wulf for useful discussions. GL wants to thank Universidad de Oviedo, Vrije Universiteit Brussel and the Albert Einstein Institute for hospitality in the late stages of this project.

**Funding information** CE is supported by the FWO-Vlaanderen through the project G006119N and by the Vrije Universiteit Brussel through the Strategic Research Program "High-Energy Physics". GL is supported by endowment funds from the Mitchell Family Foundation.

# A  Orthogonal decompositions and projectors of E$_{8(8)}$

This appendix brings to our notation the construction of $\mathfrak{e}_{8(8)}$ based on $\mathfrak{so}(8,8)$ and $\mathfrak{so}(16)$ found in [56]. In sec. A.3, we also detail some projectors used in the main text.

## A.1  SO(8, 8) decomposition of E$_{8(8)}$

Following (26), $\mathfrak{e}_{8(8)}$ is comprised by the 120 generators of $\mathfrak{so}(8,8)$ together with 128 extra generators transforming as spinors under the orthogonal group and closing back into it according to the commutators[14]

$$
\begin{aligned}
[t_{MN}, t_{PQ}] &= 2\,\eta_{MP} t_{NQ} - 2\,\eta^{NP} t^{MQ} - 2\,\eta^{MQ} t_{NP} + 2\,\eta^{NQ} t^{MP}\,, \\
[t_{MN}, t_{\mathcal{A}}] &= \tfrac{1}{2}(\Gamma_{MN})_{\mathcal{A}}{}^{\mathcal{B}} t_{\mathcal{B}}\,, \qquad [t_{\mathcal{A}}, t_{\mathcal{B}}] = -\tfrac{1}{2}\Gamma^{MN}{}_{\mathcal{A}\mathcal{B}} t_{MN}\,.
\end{aligned}
\tag{A.1}
$$

Indices are raised and lowered using the invariant metrics $\eta_{MN}$ and $\eta_{\mathcal{A}\mathcal{B}}$. Here, we use a basis where the SO(8, 8) invariant metric $\eta_{MN}$ is diagonal and given by

$$
\eta^{(\text{diag})} = \begin{pmatrix} -\delta_{\hat{I}\hat{J}} & 0 \\ 0 & \delta_{IJ} \end{pmatrix}.
\tag{A.2}
$$

The charge conjugation matrices are then given by

$$
\eta_{\mathcal{A}\mathcal{B}} = \begin{pmatrix} \delta_{AB}\delta_{\dot{A}\dot{B}} & 0 \\ 0 & -\delta_{\dot{A}\dot{B}}\delta_{AB} \end{pmatrix}, \qquad
\eta_{\dot{\mathcal{A}}\dot{\mathcal{B}}} = \begin{pmatrix} \delta_{AC}\delta_{BD} & 0 \\ 0 & -\delta_{\dot{A}\dot{C}}\delta_{\dot{B}\dot{D}} \end{pmatrix},
\tag{A.3}
$$

under the SO(8) $\times$ SO(8) breaking

$$
\begin{aligned}
\text{SO}(8,8) &\supset \text{SO}(8) \times \text{SO}(8)\,, \\
\mathbf{16} &\to (\mathbf{8}_\text{v}, \mathbf{1}) \oplus (\mathbf{1}, \mathbf{8}_\text{v})\,, \\
\mathbf{128}_\text{s} &\to (\mathbf{8}_\text{s}, \mathbf{8}_\text{c}) \oplus (\mathbf{8}_\text{c}, \mathbf{8}_\text{s})\,, \\
\mathbf{128}_\text{c} &\to (\mathbf{8}_\text{s}, \mathbf{8}_\text{s}) \oplus (\mathbf{8}_\text{c}, \mathbf{8}_\text{c})\,,
\end{aligned}
\tag{A.4}
$$

---

[14]The discussion in this appendix applies both to the global E$_{8(8)}$ in $D = 3$ as well as to its ExFT counterpart, and we have chosen to present the formulae with unbarred objects. In sec. 2.2, all indices here acquire overbars.

with indices decomposing as $X_M = \{X_{\hat{I}}, X_I\}$, $Y_{\mathcal{A}} = \{Y_{\hat{A}\dot{B}}, Y_{\hat{A}B}\}$ and $Y_{\dot{\mathcal{A}}} = \{Y_{\hat{A}B}, Y_{\hat{A}\dot{B}}\}$ for $\dot{\mathcal{A}} \in [\![1, 128]\!]$ the $\mathbf{128}_c$ index and $A, \dot{A} \in [\![1, 8]\!]$ and their hatted counterparts respectively labelling the $\mathbf{8}_s$ and $\mathbf{8}_c$ of each SO(8) factor.

The last ingredient in (A.1) are the generators of $\mathfrak{so}(8, 8)$ in the $\mathbf{128}_s$ representation, which are proportional to

$$\Gamma^{MN}{}_{\mathcal{A}}{}^{\mathcal{B}} = \tfrac{1}{2}\big(\Gamma^M{}_{\mathcal{A}}{}^{\dot{\mathcal{C}}}\,\bar{\Gamma}^N{}_{\dot{\mathcal{C}}}{}^{\mathcal{B}} - \Gamma^N{}_{\mathcal{A}}{}^{\dot{\mathcal{C}}}\,\bar{\Gamma}^M{}_{\dot{\mathcal{C}}}{}^{\mathcal{B}}\big), \tag{A.5}$$

for $\bar{\Gamma}^M$ the transpose of $\Gamma^M$. These chiral SO(8, 8) gamma matrices satisfy

$$\Gamma^M{}_{\mathcal{A}}{}^{\dot{\mathcal{C}}}\,\bar{\Gamma}^N{}_{\dot{\mathcal{C}}}{}^{\mathcal{B}} + \Gamma^N{}_{\mathcal{A}}{}^{\dot{\mathcal{C}}}\,\bar{\Gamma}^M{}_{\dot{\mathcal{C}}}{}^{\mathcal{B}} = 2\,\eta^{MN}\delta_{\mathcal{A}}{}^{\mathcal{B}}, \tag{A.6}$$

and are conveniently parametrised in terms of SO(8) gamma matrices as

$$
\begin{aligned}
\Gamma^{\hat{I}}{}_{\hat{A}B}{}^{\hat{\dot{C}}\dot{D}} &= -\delta_{\dot{B}\dot{D}}\,\gamma^{\hat{I}}{}_{\hat{A}\hat{C}}, & \Gamma^{\hat{I}}{}_{\hat{A}B}{}^{\hat{C}D} &= \delta_{BD}\,\gamma^{\hat{I}}{}_{\hat{C}\hat{A}}, \\
\Gamma^{I}{}_{\hat{A}B}{}^{\hat{C}D} &= \delta_{\hat{A}\hat{C}}\,\gamma^{I}{}_{C\dot{A}}, & \Gamma^{I}{}_{\hat{A}B}{}^{\hat{\dot{C}}\dot{D}} &= -\delta_{\hat{A}\hat{C}}\,\gamma^{I}{}_{B\dot{D}}.
\end{aligned}
\tag{A.7}
$$

These chiral SO(8) gamma matrices satisfy Clifford identities analogous to (A.6) and are chosen so that the charge conjugation matrices, $\eta_{AB}$, $\eta_{\hat{A}\hat{B}}$, etc, are just the identity matrix. Explicit expressions fulfilling these requirements are given by

$$\gamma^I = \{\gamma^{IJ}, \gamma^+, \gamma^-\}, \tag{A.8}$$

with

$$\gamma^{\mathbb{I}\mathbb{J}} = \begin{pmatrix} \epsilon^{IJ} & 2\delta^{IJ} \\ -2\delta^{IJ} & \epsilon^{IJ} \end{pmatrix}, \qquad \gamma^+ = \begin{pmatrix} \mathbb{1} & 0 \\ 0 & -\mathbb{1} \end{pmatrix}, \qquad \Gamma^- = \begin{pmatrix} 0 & \mathbb{1} \\ \mathbb{1} & 0 \end{pmatrix}, \tag{A.9}$$

in terms of SO(4) $\subset$ SU(4) $\subset$ SO(8) invariant tensors $(\epsilon^{IJ})^{KL} = \epsilon^{IJKL}$ and $(\delta^{IJ})^{KL} = \delta^{I[K}\delta^{L]J}$ under the splitting $I = \{[IJ], +, -\}$, $A = \{I, J\}$ and $\dot{A} = \{I, J\}$ with $I, J \in [\![1, 4]\!]$, and analogously for hatted indices.

For completeness, we also include expressions for the generators of $\mathfrak{so}(8, 8)$ in the $\mathbf{128}_c$ representation as well as for the other higher-order products of SO(8, 8) gamma matrices that play a rôle in the main text:

$$
\begin{aligned}
\bar{\Gamma}^{MN}{}_{\dot{\mathcal{A}}}{}^{\dot{\mathcal{B}}} &= \tfrac{1}{2}\big(\bar{\Gamma}^M{}_{\dot{\mathcal{A}}}{}^{\mathcal{C}}\,\Gamma^N{}_{\mathcal{C}}{}^{\dot{\mathcal{B}}} - \bar{\Gamma}^N{}_{\dot{\mathcal{A}}}{}^{\mathcal{C}}\,\Gamma^M{}_{\mathcal{C}}{}^{\dot{\mathcal{B}}}\big), \\
\Gamma^{MNP}{}_{\mathcal{A}}{}^{\dot{\mathcal{B}}} &= \Gamma^{[MN}{}_{\mathcal{A}}{}^{\mathcal{C}}\,\Gamma^{P]}{}_{\mathcal{C}}{}^{\dot{\mathcal{B}}}, & \bar{\Gamma}^{MNP}{}_{\dot{\mathcal{A}}}{}^{\mathcal{B}} &= \bar{\Gamma}^{[MN}{}_{\dot{\mathcal{A}}}{}^{\dot{\mathcal{C}}}\,\bar{\Gamma}^{P]}{}_{\dot{\mathcal{C}}}{}^{\mathcal{B}}, \\
\Gamma^{MNPQ}{}_{\mathcal{A}}{}^{\mathcal{B}} &= \Gamma^{[MNP}{}_{\mathcal{A}}{}^{\dot{\mathcal{C}}}\,\Gamma^{Q]}{}_{\dot{\mathcal{C}}}{}^{\mathcal{B}}, & \bar{\Gamma}^{MNPQ}{}_{\dot{\mathcal{A}}}{}^{\dot{\mathcal{B}}} &= \bar{\Gamma}^{[MNP}{}_{\dot{\mathcal{A}}}{}^{\mathcal{C}}\,\bar{\Gamma}^{Q]}{}_{\mathcal{C}}{}^{\dot{\mathcal{B}}}.
\end{aligned}
\tag{A.10}
$$

In terms of these objects, the structure constants of $E_{8(8)}$ are given by [56]

$$
\begin{aligned}
f_{MN,PQ}{}^{RS} &= -8\,\delta_{[M}{}^{[R}\eta_{N][P}\delta_{Q]}{}^{S]}, \\
f_{MN,\mathcal{A}}{}^{\mathcal{B}} &= \frac{1}{2}(\Gamma_{MN})_{\mathcal{A}}{}^{\mathcal{B}}, \\
f_{\mathcal{A}\mathcal{B}}{}^{MN} &= -\frac{1}{2}\Gamma^{MN}{}_{\mathcal{A}\mathcal{B}},
\end{aligned}
\tag{A.11}
$$

and the Cartan-Killing metric,

$$\kappa_{\mathcal{M}\mathcal{N}} = \frac{1}{60}f_{\mathcal{M}\mathcal{P}}{}^{\mathcal{Q}}f_{\mathcal{N}\mathcal{Q}}{}^{\mathcal{P}}, \tag{A.12}$$

decomposes as

$$
\begin{aligned}
\kappa_{M_1 M_2, N_1 N_2} &= -2\,\eta_{M_1[N_1}\eta_{N_2]M_2}, & \kappa_{\mathcal{A}\mathcal{B}} &= \eta_{\mathcal{A}\mathcal{B}}, \\
\kappa^{M_1 M_2, N_1 N_2} &= -2\,\eta^{M_1[N_1}\eta^{N_2]M_2}, & \kappa^{\mathcal{A}\mathcal{B}} &= \eta^{\mathcal{A}\mathcal{B}}.
\end{aligned}
\tag{A.13}
$$

## A.2 SO(16) decomposition of $E_{8(8)}$

$E_{8(8)}$ can be decomposed under SO(16) analogously to (26),

$$
\begin{aligned}
E_{8(8)} &\supset & SO(16)\,, \\
\mathbf{248} &\to & \mathbf{120} + \mathbf{128}_s\,, \\
t_{\mathcal{M}} &\to & \{t_{[MN]},\, t_{\mathcal{A}}\}\,,
\end{aligned}
\tag{A.14}
$$

with indices M and $\mathcal{A}$ now labelling the vector and spinorial representations of SO(16). The $E_{8(8)}$ structure constants in this basis are

$$
\begin{aligned}
f_{MN,PQ}{}^{RS} &= -8\,\delta_{[M}{}^{[R}\eta_{N][P}\delta_{Q]}{}^{S]}\,, \\
f_{MN,\mathcal{A}}{}^{\mathcal{B}} &= \frac{1}{2}\,\Gamma_{MN\mathcal{A}}{}^{\mathcal{B}}\,, \\
f_{\mathcal{A}\mathcal{B}}{}^{MN} &= -\frac{1}{2}\,\Gamma^{MN}{}_{\mathcal{A}\mathcal{B}}\,,
\end{aligned}
\tag{A.15}
$$

and the Cartan-Killing form (A.12) decomposes as

$$
\begin{aligned}
\kappa_{M_1M_2,N_1N_2} &= -2\,\eta_{M_1[N_1}\eta_{N_2]M_2}\,, & \kappa_{\mathcal{A}\mathcal{B}} &= \eta_{\mathcal{A}\mathcal{B}}\,, \\
\kappa^{M_1M_2,N_1N_2} &= -2\,\eta^{M_1[N_1}\eta^{N_2]M_2}\,, & \kappa^{\mathcal{A}\mathcal{B}} &= \eta^{\mathcal{A}\mathcal{B}}\,,
\end{aligned}
\tag{A.16}
$$

for $\Gamma^M$ the SO(16) gamma matrices and the invariant metric $\eta_{MN}$ and charge conjugation matrices $\eta_{\mathcal{A}\mathcal{B}}$ and $\eta_{\dot{\mathcal{A}}\dot{\mathcal{B}}}$ given by identity matrices in the respective dimensions. For this reason, the upstairs vs downstairs position of these indices lacks significance. The gamma matrices are most easily defined by breaking SO(16) down to SO(8) × SO(8),

$$
\begin{aligned}
SO(16) &\supset & SO(8) \times SO(8)\,, \\
\mathbf{16} &\to & (\mathbf{8}_c,\mathbf{1}) \oplus (\mathbf{1},\mathbf{8}_s)\,, \\
\mathbf{128}_s &\to & (\mathbf{8}_v,\mathbf{8}_v) \oplus (\mathbf{8}_s,\mathbf{8}_c)\,, \\
\mathbf{128}_c &\to & (\mathbf{8}_s,\mathbf{8}_v) \oplus (\mathbf{8}_v,\mathbf{8}_c)\,,
\end{aligned}
\tag{A.17}
$$

with indices decomposing as $X_M = \{X_{\hat{A}}, X_A\}$, $Y_{\mathcal{A}} = \{Y_{\hat{I}J}, Y_{\hat{A}\dot{B}}\}$ and $Y_{\dot{\mathcal{A}}} = \{Y_{\hat{A}I}, \hat{Y}_{\dot{J}\dot{B}}\}$. Then,

$$
\begin{aligned}
\Gamma^{\hat{A}}{}_{\hat{I}J,\hat{B}K} &= \delta_{JK}\,\gamma^{\hat{I}}{}_{\hat{B}\hat{A}}\,, & \Gamma^{\hat{A}}{}_{\hat{B}\dot{C},\hat{I}\dot{D}} &= -\delta_{\dot{C}\dot{D}}\,\gamma^{\hat{I}}{}_{\hat{B}\hat{A}}\,, \\
\Gamma^{A}{}_{\hat{I}J,\hat{K}\dot{B}} &= \delta_{\hat{I}\hat{K}}\,\gamma^{J}{}_{A\dot{B}}\,, & \Gamma^{A}{}_{\hat{B}\dot{C},\hat{D}I} &= \delta_{\hat{B}\hat{D}}\,\gamma^{I}{}_{A\dot{C}}\,,
\end{aligned}
\tag{A.18}
$$

and higher-order products follow (A.10).

One can finally map the SO(8, 8) and SO(16) representations appearing in the decomposition (26) and (A.14), $\{X_{MN}, Y_{\mathcal{A}}\}$ and $\{X'_{MN}, Y'_{\mathcal{A}}\}$, via their respective SO(8) × SO(8) breakings:

$$
\begin{aligned}
2X'_{\hat{A}\hat{B}} &= -\frac{1}{4}\gamma^{\hat{I}\hat{J}}{}_{\hat{A}\hat{B}}X_{\hat{I}J}\,, & X'_{\hat{A}B} &= -Y_{\hat{A}\dot{B}}\,, & X'_{A\hat{B}} &= Y_{\hat{A}\dot{B}}\,, & X'_{AB} &= \frac{1}{4}\gamma^{IJ}{}_{AB}X_{IJ}\,, \\
Y'_{\hat{I}J} &= -X_{\hat{I}J}\,, & Y'_{\hat{A}\dot{B}} &= Y_{\hat{A}\dot{B}}\,.
\end{aligned}
\tag{A.19}
$$

## A.3 $E_{8(8)}$ projectors

Some of the representations in the product

$$
\mathbf{248} \otimes \mathbf{248} \to \mathbf{1} \oplus \mathbf{248} \oplus \mathbf{3875} \oplus \mathbf{27000} \oplus \mathbf{30380}\,,
\tag{A.20}
$$

play a prominent rôle in supergravity and ExFT. The projectors onto these irreducible representations are given by [38,92]

$$
\begin{aligned}
(\mathbb{P}_1)_{\mathcal{MN}}{}^{\mathcal{KL}} &= \tfrac{1}{248}\kappa_{\mathcal{MN}}\,\kappa^{\mathcal{KL}}\,, \\
(\mathbb{P}_{248})_{\mathcal{MN}}{}^{\mathcal{KL}} &= \tfrac{1}{60}f_{\mathcal{MNP}}f^{\mathcal{PKL}}\,, \\
(\mathbb{P}_{3875})_{\mathcal{MN}}{}^{\mathcal{KL}} &= \tfrac{1}{7}\delta^{\mathcal{K}}_{(\mathcal{M}}\delta^{\mathcal{L}}_{\mathcal{N})} - \tfrac{1}{56}\kappa_{\mathcal{MN}}\,\kappa^{\mathcal{KL}} - \tfrac{1}{14}f^{\mathcal{P}}{}_{\mathcal{M}}{}^{(\mathcal{K}}f_{\mathcal{PN}}{}^{\mathcal{L})}\,, \\
(\mathbb{P}_{27000})_{\mathcal{MN}}{}^{\mathcal{KL}} &= \tfrac{6}{7}\delta^{\mathcal{K}}_{(\mathcal{M}}\delta^{\mathcal{L}}_{\mathcal{N})} + \tfrac{3}{217}\kappa_{\mathcal{MN}}\,\kappa^{\mathcal{KL}} + \tfrac{1}{14}f^{\mathcal{P}}{}_{\mathcal{M}}{}^{(\mathcal{K}}f_{\mathcal{PN}}{}^{\mathcal{L})}\,, \\
(\mathbb{P}_{30380})_{\mathcal{MN}}{}^{\mathcal{KL}} &= \delta^{\mathcal{K}}_{[\mathcal{M}}\delta^{\mathcal{L}}_{\mathcal{N}]} - \tfrac{1}{60}f_{\mathcal{MNP}}f^{\mathcal{PKL}}\,.
\end{aligned}
\tag{A.21}
$$

In particular, given that the embedding tensor of maximal supergravity has index structure

$$
(\mathbf{248}\otimes\mathbf{248})_{\mathrm{sym}} \to \mathbf{1}\oplus\mathbf{3875}\oplus\mathbf{27000}\,,
\tag{A.22}
$$

the linear constraint on the embedding tensor alluded to in sec. 2.2 can be phrased as

$$
(\mathbb{P}_{27000}X)_{\bar{\mathcal{M}}\bar{\mathcal{K}}} = 0\,.
\tag{A.23}
$$

# B   D(2, 1|α) vs SU(2) ⋉ SU(2|1, 1) superalgebras

The superalgebras $\mathrm{SU}(2)\ltimes\mathrm{SU}(2|1,1)$ and $\mathrm{D}(2,1|\alpha)$ coincide as vector spaces. They are generated by bosonic elements $L_m$ with $m=0,\pm1$, and $A_i^{\pm}$ with $i=1,2,3$, which respectively generate $\mathrm{SL}(2,\mathbb{R})$ and two copies of $\mathrm{SU}(2)$, and their fermionic counterparts $G_r^a$ with $r=\pm\tfrac{1}{2}$ and $a=1,2,3,4$, transforming in the bi-fundamental representation of $\mathrm{SU}(2)_-\times\mathrm{SU}(2)_+$. For $\mathrm{D}(2,1|\alpha)$, the super-Lie bracket is [67,93]

$$
\begin{aligned}
&[L_m, L_n] = (m-n)L_{m+n}\,, && [A_i^{\pm}, A_j^{\pm}] = i\epsilon_{ijk}A_k^{\pm}\,, && [L_m, A_i^{\pm}] = 0\,, \\
&[L_m, G_r^a] = \left(\tfrac{m}{2}-r\right)G_r^a\,, && [A_i^{\pm}, G_r^a] = i\alpha_{ab}^{\pm i}G_r^b\,, \\
&\{G_r^a, G_s^b\} = 2\delta^{ab}L_{r+s} + 4i(r-s)\Big[\frac{\alpha^2}{1+\alpha^2}\alpha_{ab}^{+i}A_i^+ + \frac{1}{1+\alpha^2}\alpha_{ab}^{-i}A_i^-\Big]\,,
\end{aligned}
\tag{B.1}
$$

with

$$
\alpha_{ab}^{\pm i} = \pm\delta_{i+1,[a}\delta_{b]1} + \tfrac{1}{2}\epsilon_{i,a-1,b-1,4}\,.
\tag{B.2}
$$

For $\mathrm{SU}(2)\ltimes\mathrm{SU}(2|1,1)$, only the fermionic anti-commutator is modified into

$$
\{G_r^a, G_s^b\} = 2\delta^{ab}L_{r+s} + 4i(r-s)\alpha_{ab}^{-i}A_i^-\,,
\tag{B.3}
$$

following the limit

$$
\lim_{\alpha\to0}\mathrm{D}(2,1|\alpha) = \mathrm{SU}(2|1,1)\rtimes\mathrm{SU}(2)_+\,.
\tag{B.4}
$$

Therefore, $\mathrm{SU}(2|1,1)\supset\mathrm{SL}(2,\mathbb{R})\times\mathrm{SU}(2)_-$ is an ideal of the non-semisimple $\mathrm{SU}(2)\ltimes\mathrm{SU}(2|1,1)$ superalgebra.

The limit (B.4) does not affect the matter content of long multiplets, whose states can be given in terms of the weights under the bosonic subalgebras as $(h, j^-, j^+)$, with $h$ denoting the $\mathrm{SL}(2,\mathbb{R})$ dimension and $j^{\pm}$ being half-integer spins for $\mathrm{SU}(2)_{\pm}$. Supermultiplets are then determined by a primary state which is annihilated by all $G_{\frac{1}{2}}^a$ and $L_1$. A supermultiplet with superconformal primary $(h, j^-, j^+)$ will be denoted $[h, j^-, j^+]$, and its descendants can be

obtained by successively applying antisymmetric products of the $G^a_{-\frac{1}{2}}$ generators, which live in representations

$$G^a_{-\frac{1}{2}} \in (\tfrac{1}{2}, \tfrac{1}{2}, \tfrac{1}{2}), \qquad G^{[a}_{-\frac{1}{2}} G^{b]}_{-\frac{1}{2}} \in (1,1,0) \oplus (1,0,1),$$
$$G^{[a}_{-\frac{1}{2}} G^b_{-\frac{1}{2}} G^{c]}_{-\frac{1}{2}} \in (\tfrac{3}{2}, \tfrac{1}{2}, \tfrac{1}{2}), \qquad G^1_{-\frac{1}{2}} G^2_{-\frac{1}{2}} G^3_{-\frac{1}{2}} G^4_{-\frac{1}{2}} \in (2,0,0), \tag{B.5}$$

of $\mathrm{SL}(2,\mathbb{R}) \times \mathrm{SU}(2)_- \times \mathrm{SU}(2)_+$. Applying (B.5) onto a superconformal primary with charges $(h, j^-, 0)$ one recovers the states in (A.20) of [41], whilst equation (A.17) therein applies whenever the superconformal primary has both $j^-$ and $j^+$ greater than one.

Shortening of the long multiplets occurs when the superconformal primaries saturate the BPS bounds

$$\begin{array}{ll} h \geq \frac{1}{1+\alpha^2} j^- + \frac{\alpha^2}{1+\alpha^2} j^+, & \text{for } \mathrm{D}(2,1|\alpha), \\ h \geq j^-, & \text{for } \mathrm{SU}(2|1,1) \rtimes \mathrm{SU}(2). \end{array} \tag{B.6}$$

The missing factor in (B.3) implies that short multiplets of $\mathrm{SU}(2|1,1) \rtimes \mathrm{SU}(2)$ are shorter than those of $\mathrm{D}(2,1|\alpha)$ with the same charges, since all states for which the $\mathrm{SU}(2)_-$ weight rises become null at the BPS bound. Therefore, the breaking rules are

$$\begin{array}{ll} [\frac{1}{1+\alpha^2}(j^- + \alpha^2 j^+) + \epsilon, j^-, j^+] \xrightarrow{\epsilon \to 0} [j^-, j^+]_s + [j^- + \tfrac{1}{2}, j^+ + \tfrac{1}{2}]_s, & \text{for } \mathrm{D}(2,1|\alpha), \\ [j^- + \epsilon, j^-, j^+] \xrightarrow{\epsilon \to 0} [j^-, j^+]_s + [j^- + \tfrac{1}{2}, j^+ + \tfrac{1}{2}]_s & \\ \qquad\qquad + [j^- - \tfrac{1}{2}, j^+ + \tfrac{1}{2}]_s + [j^-, j^+ + 1]_s, & \text{for } \mathrm{SU}(2|1,1) \rtimes \mathrm{SU}(2), \end{array} \tag{B.7}$$

where the conformal dimensions of the short multiplets accord to (B.6) and have been omitted. Note also that for both superalgebras the multiplet $[0,0]_s$ is unphysical, as it has $h = 0$. Explicit expressions for the state content of the short multiplets of $\mathrm{D}(2,1|\alpha)$ can be found in [19], and for $\mathrm{SU}(2|1,1) \rtimes \mathrm{SU}(2)$ in [41].

In the following, we tabulate the states of a few of these multiplets for the convenience of the reader.

Table 1: Long multiplet $[h, 0, 0]$.

| $\mathrm{SL}(2,\mathbb{R})$ | $\mathrm{SU}(2)_- \times \mathrm{SU}(2)_+$ |
|---|---|
| $h$ | $(0,0)$ |
| $h + 1/2$ | $(1/2, 1/2)$ |
| $h + 1$ | $(1,0) \oplus (0,1)$ |
| $h + 3/2$ | $(1/2, 1/2)$ |
| $h + 2$ | $(0,0)$ |

Table 2: Long multiplet $[h, 1/2, 0]$.

| $\mathrm{SL}(2,\mathbb{R})$ | $\mathrm{SU}(2)_- \times \mathrm{SU}(2)_+$ |
|---|---|
| $h$ | $(1/2, 0)$ |
| $h + 1/2$ | $(1, 1/2) \oplus (0, 1/2)$ |
| $h + 1$ | $(1/2, 1) \oplus (3/2, 0) \oplus (1/2, 0)$ |
| $h + 3/2$ | $(1, 1/2) \oplus (0, 1/2)$ |
| $h + 2$ | $(1/2, 0)$ |

Table 3: Long multiplet $[h, 1/2, 1/2]$.

| $\mathrm{SL}(2,\mathbb{R})$ | $\mathrm{SU}(2)_- \times \mathrm{SU}(2)_+$ |
|---|---|
| $h$ | $(1/2, 1/2)$ |
| $h + 1/2$ | $(1,1) \oplus (1,0) \oplus (0,1) \oplus (0,0)$ |
| $h + 1$ | $(3/2, 1/2) \oplus 2(1/2, 1/2) \oplus (1/2, 3/2)$ |
| $h + 3/2$ | $(1,1) \oplus (1,0) \oplus (0,1) \oplus (0,0)$ |
| $h + 2$ | $(1/2, 1/2)$ |

Table 4: Long multiplet $[h, 1, 0]$.

| $\mathrm{SL}(2,\mathbb{R})$ | $\mathrm{SU}(2)_- \times \mathrm{SU}(2)_+$ |
|---|---|
| $h$ | $(1,0)$ |
| $h + 1/2$ | $(1/2, 1/2) \oplus (3/2, 1/2)$ |
| $h + 1$ | $(1,1) \oplus (2,0) \oplus (1,0) \oplus (0,0)$ |
| $h + 3/2$ | $(1/2, 1/2) \oplus (3/2, 1/2)$ |
| $h + 2$ | $(1,0)$ |

Table 5: Short multiplet $\left[1/2, 1/2\right]_{\mathrm{s}}$ for D(2, 1|α) and SU(2|1, 1) ⋊ SU(2).

| D(2, 1\|α) | SU(2\|1, 1) ⋊ SU(2) |
|:---:|:---:|
| $(1/2, 1/2)$ | $(1/2, 1/2)$ |
| $(1,0) \oplus (0,1) \oplus (0,0)$ | $(0,1) \oplus (0,0)$ |
| $(1/2, 1/2)$ | — |
| $(0,0)$ | — |

Table 6: Short multiplet $\left[1, 1\right]_{\mathrm{s}}$ for D(2, 1|α) and SU(2|1, 1) ⋊ SU(2).

| D(2, 1\|α) | SU(2\|1, 1) ⋊ SU(2) |
|:---:|:---:|
| $(1,1)$ | $(1,1)$ |
| $(3/2, 1/2) \oplus (1/2, 1/2) \oplus (1/2, 3/2)$ | $(1/2, 3/2) \oplus (1/2, 1/2)$ |
| $(1,1) \oplus (1,0) \oplus (0,1)$ | $(0,1)$ |
| $(1/2, 1/2)$ | — |

# C   Solution details

In this appendix, we present the detailed $10d$ configurations corresponding to the solutions presented in sec. 5, uplifted from $3d$ through (62) and (64). As discussed in sec. 6, the linear variations around a vacuum in these conformal manifolds can be parameterised in terms of the Noether currents introduced in (220). The conservation of the Noether current $\mathcal{J}_p$ associated to the shift of the angle $y^p$ in (221) reads

$$\mathrm{d} * \mathcal{J}_p = [\bar{\partial} j_p + \partial \bar{j}_p] dz \wedge d\bar{z} = 0 \,. \tag{C.1}$$

The components of $\mathcal{J}_p$ are not necessarily (anti-)holomorphic, but in the following we show that the linear combinations relevant for the different backgrounds are chiral upon imposing the equations of motion

$$\mathcal{E}_y = \partial\left(\frac{\delta\mathcal{L}}{\delta\partial y}\right) + \bar{\partial}\left(\frac{\delta\mathcal{L}}{\delta\bar{\partial} y}\right) - \frac{\delta\mathcal{L}}{\delta y} \,, \tag{C.2}$$

for the worldsheet Lagrangian in (219). For the three-parameter family of solutions in (172) and the currents in (220), this amounts to

$$
\begin{aligned}
\bar{\partial} j_{\varphi_1} &= \frac{1}{2}\left(1 - e^{2\omega}\beta_1\chi_1\right)\mathcal{E}_{\varphi_1} - \frac{e^{2\omega}}{2}\mathcal{E}_{\varphi_2} + \frac{e^{2\omega}}{2}\beta_1\mathcal{E}_{y^7}\,, \\
\bar{\partial} j_{y^7} &= -\frac{1}{2}\beta_1\left(1 + e^{2\omega}\chi_1^2\right)\mathcal{E}_{\varphi_1} - \frac{e^{2\omega}}{2}\chi_1\mathcal{E}_{\varphi_2} + \frac{1}{2}\left(1 + e^{2\omega}\beta_1\chi_1\right)\mathcal{E}_{y^7}\,, \\
\partial \bar{j}_{\varphi_1} &= \frac{1}{2}\left(1 + e^{2\omega}\beta_1\chi_1\right)\mathcal{E}_{\varphi_1} + \frac{e^{2\omega}}{2}\mathcal{E}_{\varphi_2} - \frac{e^{2\omega}}{2}\beta_1\mathcal{E}_{y^7}\,, \\
\partial \bar{j}_{y^7} &= \frac{1}{2}\beta_1\left(1 + e^{2\omega}\chi_1^2\right)\mathcal{E}_{\varphi_1} + \frac{e^{2\omega}}{2}\chi_1\mathcal{E}_{\varphi_2} + \frac{1}{2}\left(1 - e^{2\omega}\beta_1\chi_1\right)\mathcal{E}_{y^7}\,,
\end{aligned}
\tag{C.3}
$$

for both the $S^3 \times S^1$ and $T^4$ backgrounds. In fact, one can show that $y^7$ is a free field satisfying $\bar{\partial}\partial y^7 = 0$.

## C.1 Wilson loop deformations

For $\text{AdS}_3 \times S^3 \times \widetilde{S}^3 \times S^1$, the four-axion geometry corresponding to (184) is given by [19]

$$
\begin{aligned}
e^{\hat{\Phi}} &= 1, \\
\mathrm{d}\hat{s}_s^2 &= \ell_{\text{AdS}}^2 \mathrm{d}s^2(\text{AdS}_3) + (\mathrm{d}y^7)^2 \\
&\quad + \mathrm{d}\theta^2 + \cos^2(\theta)\big(\mathrm{d}\varphi_1 + \chi_1\,\mathrm{d}y^7\big)^2 + \sin^2(\theta)\big(\mathrm{d}\varphi_2 - \chi_2\,\mathrm{d}y^7\big)^2 \\
&\quad + \alpha^{-2}\Big(\mathrm{d}\widetilde{\theta}^2 + \cos^2(\widetilde{\theta})\big(\mathrm{d}\widetilde{\varphi}_1 + \alpha\,\widetilde{\chi}_1\,\mathrm{d}y^7\big)^2 + \sin^2(\widetilde{\theta})\big(\mathrm{d}\widetilde{\varphi}_2 - \alpha\,\widetilde{\chi}_2\,\mathrm{d}y^7\big)^2\Big), \\
\hat{H}_{(3)} &= 2\ell_{\text{AdS}}^2 \,\text{vol}(\text{AdS}_3) + \sin(2\theta)\,\mathrm{d}\theta \wedge \big(\mathrm{d}\varphi_1 + \chi_1\,\mathrm{d}y^7\big) \wedge \big(\mathrm{d}\varphi_2 - \chi_2\,\mathrm{d}y^7\big) \\
&\quad + \alpha^{-2}\sin(2\widetilde{\theta})\,\mathrm{d}\widetilde{\theta} \wedge \big(\mathrm{d}\widetilde{\varphi}_1 + \alpha\,\widetilde{\chi}_1\,\mathrm{d}y^7\big) \wedge \big(\mathrm{d}\widetilde{\varphi}_2 - \alpha\,\widetilde{\chi}_2\,\mathrm{d}y^7\big), \\
\hat{G}_{(1)} &= \hat{G}_{(3)} = \hat{G}_{(5)} = 0,
\end{aligned}
\tag{C.4}
$$

and for $\text{AdS}_3 \times S^3 \times \text{T}^4$, it is realised by

$$
\begin{aligned}
e^{\hat{\Phi}} &= 1, \\
\mathrm{d}\hat{s}_s^2 &= \ell_{\text{AdS}}^2 \mathrm{d}s^2(\text{AdS}_3) + \mathrm{d}\theta^2 + \cos^2(\theta)\big(\mathrm{d}\varphi_1 + \chi_1\mathrm{d}y^7\big)^2 + \sin^2(\theta)\big(\mathrm{d}\varphi_2 - \chi_2\mathrm{d}y^7\big)^2 \\
&\quad + (\mathrm{d}y^4)^2 + (\mathrm{d}y^5)^2 + \big(\mathrm{d}y^6 + \widetilde{\chi}_2\mathrm{d}y^7\big)^2 + (\mathrm{d}y^7)^2, \\
\hat{H}_{(3)} &= 2\ell_{\text{AdS}}^2 \,\text{vol}(\text{AdS}_3) + \sin(2\theta)\,\mathrm{d}\theta \wedge \big(\mathrm{d}\varphi_1 + \chi_1\,\mathrm{d}y^7\big) \wedge \big(\mathrm{d}\varphi_2 - \chi_2\,\mathrm{d}y^7\big), \\
\hat{G}_{(1)} &= \hat{G}_{(3)} = \hat{G}_{(5)} = 0.
\end{aligned}
\tag{C.5}
$$

The currents taking part in (225) and (227) are respectively

$$
\begin{aligned}
j_{\varphi_1} &= \big(\partial\varphi_1 + \chi_1\partial y^7\big)\cos^2\theta - \big(\partial\varphi_2 - \chi_2\partial y^7\big)\sin^2\theta, \\
j_{\widetilde{\varphi}_1} &= \alpha^{-2}\big(\partial\varphi_1 + \alpha\widetilde{\chi}_1\partial y^7\big)\cos^2\widetilde{\theta} - \alpha^{-2}\big(\partial\varphi_2 - \alpha\widetilde{\chi}_2\partial y^7\big)\sin^2\widetilde{\theta}, \\
\bar{j}_{\varphi_1} &= \big(\bar{\partial}\varphi_1 + \chi_1\bar{\partial}y^7\big)\cos^2\theta + \big(\bar{\partial}\varphi_2 - \chi_2\bar{\partial}y^7\big)\sin^2\theta, \\
\bar{j}_{\widetilde{\varphi}_1} &= \alpha^{-2}\big(\bar{\partial}\varphi_1 + \alpha\widetilde{\chi}_1\bar{\partial}y^7\big)\cos^2\widetilde{\theta} + \alpha^{-2}\big(\bar{\partial}\varphi_2 - \alpha\widetilde{\chi}_2\bar{\partial}y^7\big)\sin^2\widetilde{\theta},
\end{aligned}
\tag{C.6}
$$

and

$$
\begin{aligned}
j_{\varphi_1} &= \big(\partial\varphi_1 + \chi_1\partial y^7\big)\cos^2\theta - \big(\partial\varphi_2 - \chi_2\partial y^7\big)\sin^2\theta, & j_{y^6} &= \partial y^6 + \widetilde{\chi}_2\partial y^7, \\
\bar{j}_{\varphi_1} &= \big(\bar{\partial}\varphi_1 + \chi_1\bar{\partial}y^7\big)\cos^2\theta + \big(\bar{\partial}\varphi_2 - \chi_2\bar{\partial}y^7\big)\sin^2\theta, & \bar{j}_{y^6} &= \bar{\partial}y^6 + \widetilde{\chi}_2\bar{\partial}y^7.
\end{aligned}
\tag{C.7}
$$

Furthermore, $\partial y^7$ can be expressed in terms of the currents in both cases. These currents satisfy

$$
\bar{\partial}j_{\varphi_1} = \tfrac{1}{2}\mathcal{E}_{\varphi_1} - \tfrac{1}{2}\mathcal{E}_{\varphi_2}, \qquad \partial\bar{j}_{\varphi_1} = \tfrac{1}{2}\mathcal{E}_{\varphi_1} + \tfrac{1}{2}\mathcal{E}_{\varphi_2},
\tag{C.8}
$$

for both configurations, and

$$
\begin{aligned}
\bar{\partial}j_{\widetilde{\varphi}_1} &= \tfrac{1}{2}\mathcal{E}_{\widetilde{\varphi}_1} - \tfrac{1}{2}\mathcal{E}_{\widetilde{\varphi}_2}, \qquad \partial\bar{j}_{\widetilde{\varphi}_1} = \tfrac{1}{2}\mathcal{E}_{\widetilde{\varphi}_1} + \tfrac{1}{2}\mathcal{E}_{\widetilde{\varphi}_2}, \\
\bar{\partial}\partial y^7 &= -\tfrac{1}{2}\chi_1\mathcal{E}_{\varphi_1} + \tfrac{1}{2}\chi_2\mathcal{E}_{\varphi_2} - \tfrac{\alpha}{2}\widetilde{\chi}_1\mathcal{E}_{\widetilde{\varphi}_1} + \tfrac{1}{2}\widetilde{\chi}_2\mathcal{E}_{\widetilde{\varphi}_2} + \tfrac{1}{2}\mathcal{E}_{y^7},
\end{aligned}
\tag{C.9}
$$

or

$$
\begin{aligned}
\bar{\partial}j_{y^6} &= \tfrac{1}{2}\mathcal{E}_{y^6}, \qquad \partial\bar{j}_{y^6} = \tfrac{1}{2}\mathcal{E}_{y^6}, \\
\bar{\partial}\partial y^7 &= -\tfrac{1}{2}\chi_1\mathcal{E}_{\varphi_1} + \tfrac{1}{2}\chi_2\mathcal{E}_{\varphi_2} - \tfrac{1}{2}\widetilde{\chi}_2\mathcal{E}_{y^6} + \tfrac{1}{2}\mathcal{E}_{y^7}.
\end{aligned}
\tag{C.10}
$$

depending on the topology.

## C.2 TsT deformations

### C.2.1 4$\beta$-family

For AdS$_3 \times S^3 \times \widetilde{S}^3 \times S^1$, (194) uplifts to

$$e^{\mathring{\Phi}} = \sqrt{\Delta}\,,$$

$$\begin{aligned}
\mathrm{d}\hat{s}_s^2 &= \ell_{\mathrm{AdS}}^2 \mathrm{d}s^2(\mathrm{AdS}_3) + \mathrm{d}\theta^2 + \cos^2(\theta)\mathrm{d}\varphi_1^2 + \sin^2(\theta)\mathrm{d}\varphi_2^2 + \alpha^{-2}\Big(\mathrm{d}\widetilde{\theta}^2 + \cos^2(\widetilde{\theta})\mathrm{d}\widetilde{\varphi}_1^2 + \sin^2(\widetilde{\theta})\mathrm{d}\widetilde{\varphi}_2^2\Big) \\
&\quad -\Delta\big(\beta_1 \cos^2(\theta)\mathrm{d}\varphi_1 - \beta_2 \sin^2(\theta)\mathrm{d}\varphi_2 + \alpha^{-1}\widetilde{\beta}_1 \cos^2(\widetilde{\theta})\mathrm{d}\widetilde{\varphi}_1 - \alpha^{-1}\widetilde{\beta}_2 \sin^2(\widetilde{\theta})\mathrm{d}\widetilde{\varphi}_2\big)^2 \\
&\quad +\Delta\big(\mathrm{d}y^7 - \beta_2 \cos^2(\theta)\mathrm{d}\varphi_1 + \beta_1 \sin^2(\theta)\mathrm{d}\varphi_2 - \alpha^{-1}\widetilde{\beta}_2 \cos^2(\widetilde{\theta})\mathrm{d}\widetilde{\varphi}_1 + \alpha^{-1}\widetilde{\beta}_1 \sin^2(\widetilde{\theta})\mathrm{d}\widetilde{\varphi}_2\big)^2\,,
\end{aligned}$$

$$\hat{H}_{(3)} = 2\ell_{\mathrm{AdS}}^2 \mathrm{vol}(\mathrm{AdS}_3) + \sin(2\theta)\,\mathrm{d}\theta \wedge v_1 \wedge v_2 + \sin(2\widetilde{\theta})\,\mathrm{d}\widetilde{\theta} \wedge \widetilde{v}_1 \wedge \widetilde{v}_2\,,$$

$$\hat{G}_{(1)} = \hat{G}_{(3)} = \hat{G}_{(5)} = 0\,, \tag{C.11}$$

with the warping factor

$$\Delta = \frac{1}{1 + \beta_1^2 \cos^2(\theta) + \beta_2^2 \sin^2(\theta) + \widetilde{\beta}_1^2 \cos^2(\widetilde{\theta}) + \widetilde{\beta}_2^2 \sin^2(\widetilde{\theta})}\,, \tag{C.12}$$

and the one-forms

$$\begin{aligned}
v_1 &= \big(1 + \Delta(\beta_2^2 - \beta_1^2)\cos^2(\theta)\big)\mathrm{d}\varphi_1 - \Delta\Big[\beta_2 \mathrm{d}y^7 + (\beta_1\widetilde{\beta}_1 - \beta_2\widetilde{\beta}_2)\cos^2(\widetilde{\theta})\frac{\mathrm{d}\widetilde{\varphi}_1}{\alpha} + (\beta_2\widetilde{\beta}_1 - \beta_1\widetilde{\beta}_2)\sin^2(\widetilde{\theta})\frac{\mathrm{d}\widetilde{\varphi}_2}{\alpha}\Big]\,, \\
v_2 &= \big(1 + \Delta(\beta_1^2 - \beta_2^2)\sin^2(\theta)\big)\mathrm{d}\varphi_2 + \Delta\Big[\beta_1 \mathrm{d}y^7 + (\beta_2\widetilde{\beta}_1 - \beta_1\widetilde{\beta}_2)\cos^2(\widetilde{\theta})\frac{\mathrm{d}\widetilde{\varphi}_1}{\alpha} + (\beta_1\widetilde{\beta}_1 - \beta_2\widetilde{\beta}_2)\sin^2(\widetilde{\theta})\frac{\mathrm{d}\widetilde{\varphi}_2}{\alpha}\Big]\,, \\
\widetilde{v}_1 &= \big(1 + \Delta(\widetilde{\beta}_2^2 - \widetilde{\beta}_1^2)\cos^2(\widetilde{\theta})\big)\frac{\mathrm{d}\widetilde{\varphi}_1}{\alpha} - \Delta\Big[\widetilde{\beta}_2 \mathrm{d}y^7 + (\beta_1\widetilde{\beta}_1 - \beta_2\widetilde{\beta}_2)\cos^2(\theta)\mathrm{d}\varphi_1 - (\beta_2\widetilde{\beta}_1 - \beta_1\widetilde{\beta}_2)\sin^2(\theta)\mathrm{d}\varphi_2\Big]\,, \\
\widetilde{v}_2 &= \big(1 + \Delta(\widetilde{\beta}_1^2 - \widetilde{\beta}_2^2)\sin^2(\widetilde{\theta})\big)\frac{\mathrm{d}\widetilde{\varphi}_2}{\alpha} + \Delta\Big[\widetilde{\beta}_1 \mathrm{d}y^7 - (\beta_2\widetilde{\beta}_1 - \beta_1\widetilde{\beta}_2)\cos^2(\theta)\mathrm{d}\varphi_1 + (\beta_1\widetilde{\beta}_1 - \beta_2\widetilde{\beta}_2)\sin^2(\theta)\mathrm{d}\varphi_2\Big]\,. \tag{C.13}
\end{aligned}$$

For AdS$_3 \times S^3 \times \mathrm{T}^4$, its uplift reads

$$e^{\mathring{\Phi}} = \sqrt{\Delta}\,,$$

$$\begin{aligned}
\mathrm{d}\hat{s}_s^2 &= \ell_{\mathrm{AdS}}^2 \mathrm{d}s^2(\mathrm{AdS}_3) + \mathrm{d}\theta^2 + \cos^2(\theta)\mathrm{d}\varphi_1^2 + \sin^2(\theta)\mathrm{d}\varphi_2^2 + (\mathrm{d}y^4)^2 + (\mathrm{d}y^5)^2 + (\mathrm{d}y^6)^2 \\
&\quad -\Delta\big(\beta_1 \cos^2(\theta)\mathrm{d}\varphi_1 - \beta_2 \sin^2(\theta)\mathrm{d}\varphi_2 + \widetilde{\beta}_2 \mathrm{d}y^6\big)^2 \\
&\quad +\Delta\big(\mathrm{d}y^7 - \beta_2 \cos^2(\theta)\mathrm{d}\varphi_1 + \beta_1 \sin^2(\theta)\mathrm{d}\varphi_2 - \widetilde{\beta}_1 \mathrm{d}y^6\big)^2\,,
\end{aligned}$$

$$\begin{aligned}
\hat{H}_{(3)} &= 2\ell_{\mathrm{AdS}}^2 \mathrm{vol}(\mathrm{AdS}_3) + \Delta^2 \sin(2\theta)\mathrm{d}\theta \wedge \big((1 + \beta_2^2 + \widetilde{\beta}_2^2)\mathrm{d}\varphi_1 + (\beta_2\widetilde{\beta}_1 - \beta_1\widetilde{\beta}_2)\mathrm{d}y^6 - \beta_2 \mathrm{d}y^7\big) \\
&\qquad\qquad\qquad\qquad\qquad \wedge \big((1 + \beta_1^2 + \widetilde{\beta}_2^2)\mathrm{d}\varphi_2 - (\beta_1\widetilde{\beta}_1 - \beta_2\widetilde{\beta}_2)\mathrm{d}y^6 + \beta_1 \mathrm{d}y^7\big)\,,
\end{aligned}$$

$$\hat{G}_{(1)} = \hat{G}_{(3)} = \hat{G}_{(5)} = 0\,, \tag{C.14}$$

with the warping factor now being

$$\Delta = \frac{1}{1 + \widetilde{\beta}_2^2 + \beta_1^2 \cos^2(\theta) + \beta_2^2 \sin^2(\theta)}\,. \tag{C.15}$$

The currents taking part in (228) and (225) are respectively

$$\begin{aligned}
j_{\varphi_1} &= \Delta\Big[\big(\Delta^{-1} + (\beta_2^2 - \beta_1^2)\cos^2(\theta)\big)j_3^{\mathrm{L}} - \alpha^{-1}(\beta_1 - \beta_2)(\widetilde{\beta}_1 + \widetilde{\beta}_2)\cos^2(\theta)\widetilde{j}_3^{\mathrm{L}} + (\beta_1 - \beta_2)\cos^2(\theta)\partial y^7\Big]\,, \\
j_{\widetilde{\varphi}_1} &= \alpha^{-1}\Delta\Big[\alpha^{-1}\big(\Delta^{-1} + (\widetilde{\beta}_2^2 - \widetilde{\beta}_1^2)\cos^2(\widetilde{\theta})\big)\widetilde{j}_3^{\mathrm{L}} - (\beta_1 + \beta_2)(\widetilde{\beta}_1 - \widetilde{\beta}_2)\cos^2(\widetilde{\theta})j_3^{\mathrm{L}} + (\widetilde{\beta}_1 - \widetilde{\beta}_2)\cos^2(\widetilde{\theta})\partial y^7\Big]\,, \\
j_{y^7} &= \Delta\Big[\partial y^7 - (\beta_1 + \beta_2)j_3^{\mathrm{L}} - \alpha^{-1}(\widetilde{\beta}_1 + \widetilde{\beta}_2)\widetilde{j}_3^{\mathrm{L}}\Big]\,, \\
\bar{j}_{\varphi_1} &= \Delta\Big[\big(\Delta^{-1} + (\beta_2^2 - \beta_1^2)\cos^2(\theta)\big)j_3^{\mathrm{R}} - \alpha^{-1}(\beta_1 + \beta_2)(\widetilde{\beta}_1 - \widetilde{\beta}_2)\cos^2(\theta)\widetilde{j}_3^{\mathrm{R}} - (\beta_1 + \beta_2)\cos^2(\theta)\bar{\partial}y^7\Big]\,, \\
\bar{j}_{\widetilde{\varphi}_1} &= \alpha^{-1}\Delta\Big[\alpha^{-1}\big(\Delta^{-1} + (\widetilde{\beta}_2^2 - \widetilde{\beta}_1^2)\cos^2(\widetilde{\theta})\big)\widetilde{j}_3^{\mathrm{R}} - (\beta_1 - \beta_2)(\widetilde{\beta}_1 + \widetilde{\beta}_2)\cos^2(\widetilde{\theta})j_3^{\mathrm{R}} - (\widetilde{\beta}_1 + \widetilde{\beta}_2)\cos^2(\widetilde{\theta})\bar{\partial}y^7\Big]\,, \\
\bar{j}_{y^7} &= \Delta\Big[\bar{\partial}y^7 + (\beta_1 - \beta_2)j_3^{\mathrm{R}} + \alpha^{-1}(\widetilde{\beta}_1 - \widetilde{\beta}_2)\widetilde{j}_3^{\mathrm{R}}\Big]\,, \tag{C.16}
\end{aligned}$$

with $\Delta$ in (C.12), and

$$j_{\varphi_1} = \Delta\Big[\big(1+\beta_2^2+\tilde{\beta}_2^2\big)j_3^{\mathrm{L}} + (\beta_1-\beta_2)(\partial y^7 - (\tilde{\beta}_1+\tilde{\beta}_2)\partial y^6)\cos^2(\theta)\Big],$$

$$j_{y^6} = \Delta\Big[\big(\tilde{\beta}_1^2-\tilde{\beta}_2^2+\Delta^{-1}\big)\partial y^6 - (\tilde{\beta}_1-\tilde{\beta}_2)\partial y^7 + (\beta_1+\beta_2)(\tilde{\beta}_1-\tilde{\beta}_2)j_3^{\mathrm{L}}\Big],$$

$$j_{y^7} = \Delta\Big[\partial y^7 - (\tilde{\beta}_1+\tilde{\beta}_2)\partial y^6 - (\beta_1+\beta_2)j_3^{\mathrm{L}}\Big],$$

$$\bar{j}_{\varphi_1} = \Delta\Big[\big(1+\beta_2^2+\tilde{\beta}_2^2\big)j_3^{\mathrm{R}} - (\beta_1+\beta_2)(\bar{\partial} y^7 - (\tilde{\beta}_1-\tilde{\beta}_2)\bar{\partial} y^6)\cos^2(\theta)\Big],$$

$$\bar{j}_{y^6} = \Delta\Big[\big(\tilde{\beta}_1^2-\tilde{\beta}_2^2+\Delta^{-1}\big)\bar{\partial} y^6 - (\tilde{\beta}_1+\tilde{\beta}_2)\bar{\partial} y^7 - (\beta_1-\beta_2)(\tilde{\beta}_1+\tilde{\beta}_2)j_3^{\mathrm{R}}\Big],$$

$$\bar{j}_{y^7} = \Delta\Big[\bar{\partial} y^7 - (\tilde{\beta}_1-\tilde{\beta}_2)\bar{\partial} y^6 + (\beta_1-\beta_2)j_3^{\mathrm{R}}\Big], \tag{C.17}$$

with $\Delta$ in (C.15). Here, $j_3^{\mathrm{L}}$ and $j_3^{\mathrm{R}}$ are the currents in (215), and $\tilde{j}_3^{\mathrm{L}}$ and $\tilde{j}_3^{\mathrm{R}}$ their tilded counterparts. The deformed currents satisfy

$$\bar{\partial} j_{\varphi_1} = \tfrac{1}{2}(1+\beta_1\beta_2)\,\mathcal{E}_{\varphi_1} - \tfrac{1}{2}(1+\beta_2^2)\,\mathcal{E}_{\varphi_2} + \tfrac{\alpha}{2}\beta_2\tilde{\beta}_1\,\mathcal{E}_{\tilde{\varphi}_1} - \tfrac{\alpha}{2}\beta_2\tilde{\beta}_2\,\mathcal{E}_{\tilde{\varphi}_2} + \tfrac{1}{2}\big(\beta_1+\beta_1\beta_2^2+\beta_2\tilde{\beta}_1\tilde{\beta}_2\big)\,\mathcal{E}_{y^7},$$

$$\bar{\partial} j_{\tilde{\varphi}_1} = \tfrac{1}{2\alpha}\beta_1\tilde{\beta}_2\,\mathcal{E}_{\varphi_1} - \tfrac{1}{2\alpha}\beta_2\tilde{\beta}_2\,\mathcal{E}_{\varphi_2} + \tfrac{1}{2}\big(1+\tilde{\beta}_1\tilde{\beta}_2\big)\,\mathcal{E}_{\tilde{\varphi}_1} - \tfrac{1}{2}\big(1+\tilde{\beta}_2^2\big)\,\mathcal{E}_{\tilde{\varphi}_2} + \tfrac{1}{2\alpha}\big(\tilde{\beta}_1+\tilde{\beta}_1\tilde{\beta}_2^2+\beta_1\beta_2\tilde{\beta}_2\big)\,\mathcal{E}_{y^7},$$

$$\bar{\partial} j_{y^7} = -\tfrac{1}{2}\beta_1\,\mathcal{E}_{\varphi_1} + \tfrac{1}{2}\beta_2\,\mathcal{E}_{\varphi_2} - \tfrac{\alpha}{2}\tilde{\beta}_1\,\mathcal{E}_{\tilde{\varphi}_1} + \tfrac{\alpha}{2}\tilde{\beta}_2\,\mathcal{E}_{\tilde{\varphi}_2} + \tfrac{1}{2}\big(1-\beta_1\beta_2-\tilde{\beta}_1\tilde{\beta}_2\big)\,\mathcal{E}_{y^7},$$

$$\partial \bar{j}_{\varphi_1} = \tfrac{1}{2}(1-\beta_1\beta_2)\,\mathcal{E}_{\varphi_1} + \tfrac{1}{2}\big(1+\beta_2^2\big)\,\mathcal{E}_{\varphi_2} - \tfrac{\alpha}{2}\beta_2\tilde{\beta}_1\,\mathcal{E}_{\tilde{\varphi}_1} + \tfrac{\alpha}{2}\beta_2\tilde{\beta}_2\,\mathcal{E}_{\tilde{\varphi}_2} - \tfrac{1}{2}\big(\beta_1+\beta_1\beta_2^2+\beta_2\tilde{\beta}_1\tilde{\beta}_2\big)\,\mathcal{E}_{y^7}$$

$$\partial \bar{j}_{\tilde{\varphi}_1} = -\tfrac{1}{2\alpha}\beta_1\tilde{\beta}_2\,\mathcal{E}_{\varphi_1} + \tfrac{1}{2\alpha}\beta_2\tilde{\beta}_2\,\mathcal{E}_{\varphi_2} + \tfrac{1}{2}\big(1-\tilde{\beta}_1\tilde{\beta}_2\big)\,\mathcal{E}_{\tilde{\varphi}_1} + \tfrac{1}{2}\big(1+\tilde{\beta}_2^2\big)\,\mathcal{E}_{\tilde{\varphi}_2} - \tfrac{1}{2\alpha}\big(\tilde{\beta}_1+\tilde{\beta}_1\tilde{\beta}_2^2+\beta_1\beta_2\tilde{\beta}_2\big)\,\mathcal{E}_{y^7},$$

$$\partial \bar{j}_{y^7} = \tfrac{1}{2}\beta_1\,\mathcal{E}_{\varphi_1} - \tfrac{1}{2}\beta_2\,\mathcal{E}_{\varphi_2} + \tfrac{\alpha}{2}\tilde{\beta}_1\,\mathcal{E}_{\tilde{\varphi}_1} - \tfrac{\alpha}{2}\tilde{\beta}_2\,\mathcal{E}_{\tilde{\varphi}_2} + \tfrac{1}{2}\big(1+\beta_1\beta_2+\tilde{\beta}_1\tilde{\beta}_2\big)\,\mathcal{E}_{y^7}, \tag{C.18}$$

for the $S^3 \times S^1$ topology, and

$$\bar{\partial} j_{\varphi_1} = \tfrac{1}{2}\big(1+\beta_1\beta_2\big)\mathcal{E}_{\varphi_1} - \tfrac{1}{2}\big(1+\beta_2^2\big)\mathcal{E}_{\varphi_2} + \tfrac{1}{2}\beta_2\tilde{\beta}_2\mathcal{E}_{y^6} + \tfrac{1}{2}\big(\beta_1+\beta_1\beta_2^2+\beta_2\tilde{\beta}_1\tilde{\beta}_2\big)\mathcal{E}_{y^7},$$

$$\bar{\partial} j_{y^6} = \tfrac{1}{2}\beta_1\tilde{\beta}_1\mathcal{E}_{\varphi_1} - \tfrac{1}{2}\beta_2\tilde{\beta}_1\mathcal{E}_{\varphi_2} + \tfrac{1}{2}\big(1+\tilde{\beta}_1\tilde{\beta}_2\big)\mathcal{E}_{y^6} + \tfrac{1}{2}\big(\tilde{\beta}_2+\tilde{\beta}_2\tilde{\beta}_1^2+\beta_1\beta_2\tilde{\beta}_1\big)\mathcal{E}_{y^7},$$

$$\bar{\partial} j_{y^7} = -\tfrac{1}{2}\beta_1\mathcal{E}_{\varphi_1} + \tfrac{1}{2}\beta_2\mathcal{E}_{\varphi_2} - \tfrac{1}{2}\tilde{\beta}_2\mathcal{E}_{y^6} + \tfrac{1}{2}\big(1-\beta_1\beta_2-\tilde{\beta}_1\tilde{\beta}_2\big)\mathcal{E}_{y^7},$$

$$\partial \bar{j}_{\varphi_1} = \tfrac{1}{2}\big(1-\beta_1\beta_2\big)\mathcal{E}_{\varphi_1} + \tfrac{1}{2}\big(1+\beta_2^2\big)\mathcal{E}_{\varphi_2} - \tfrac{1}{2}\beta_2\tilde{\beta}_2\mathcal{E}_{y^6} - \tfrac{1}{2}\big(\beta_1+\beta_1\beta_2^2+\beta_2\tilde{\beta}_1\tilde{\beta}_2\big)\mathcal{E}_{y^7}$$

$$\partial \bar{j}_{y^6} = -\tfrac{1}{2}\beta_1\tilde{\beta}_1\mathcal{E}_{\varphi_1} + \tfrac{1}{2}\beta_2\tilde{\beta}_1\mathcal{E}_{\varphi_2} + \tfrac{1}{2}\big(1-\tilde{\beta}_1\tilde{\beta}_2\big)\mathcal{E}_{y^6} - \tfrac{1}{2}\big(\tilde{\beta}_2+\tilde{\beta}_2\tilde{\beta}_1^2+\beta_1\beta_2\tilde{\beta}_1\big)\mathcal{E}_{y^7},$$

$$\partial \bar{j}_{y^7} = \tfrac{1}{2}\beta_1\mathcal{E}_{\varphi_1} - \tfrac{1}{2}\beta_2\mathcal{E}_{\varphi_2} + \tfrac{1}{2}\tilde{\beta}_2\mathcal{E}_{y^6} + \tfrac{1}{2}\big(1+\beta_1\beta_2+\tilde{\beta}_1\tilde{\beta}_2\big)\mathcal{E}_{y^7}, \tag{C.19}$$

in the $\mathrm{T}^4$ background case. Again, in both cases $\bar{\partial}\partial y^7 = 0$ on shell.

### C.2.2 Between $S^3$ and $S^3$

The family (203) can be embedded in $10d$ as

$$e^{\hat{\Phi}} = \sqrt{\Delta},$$

$$d\hat{s}^2 = ds^2(\mathrm{AdS}_3) + (dy^7)^2 + d\theta^2 + \alpha^{-2}d\tilde{\theta}^2$$

$$+ \Delta\bigg(\cos^2(\theta)\cos^2(\tilde{\theta})\Big(d\varphi_1 + \frac{\Xi_2}{\alpha}d\tilde{\varphi}_1\Big)^2 + \sin^2(\theta)\cos^2(\tilde{\theta})\Big(d\varphi_2 - \frac{\Xi_4}{\alpha}d\tilde{\varphi}_1\Big)^2$$

$$+ \cos^2(\theta)\sin^2(\tilde{\theta})\Big(\Xi_4\,d\varphi_1 + \frac{1}{\alpha}d\tilde{\varphi}_2\Big)^2 + \sin^2(\theta)\sin^2(\tilde{\theta})\Big(\Xi_2\,d\varphi_2 - \frac{1}{\alpha}d\tilde{\varphi}_2\Big)^2$$

$$+ \cos^2(\theta)\sin^2(\tilde{\theta})\,d\varphi_1^2 + \sin^2(\theta)\sin^2(\tilde{\theta})\,d\varphi_2^2 + \frac{1}{\alpha^2}\cos^2(\tilde{\theta})\,d\tilde{\varphi}_1^2\bigg),$$

$$\hat{H}_{(3)} = 2\,\ell_{\text{AdS}}^2\,\text{vol}(\text{AdS}_3)$$

$$+\Delta^2\sin(2\theta)\,d\theta \wedge \left( \left(1 + \Xi_2^4\sin^2(\widetilde{\theta})\right)d\varphi_1 + \Xi_2\cos^2(\widetilde{\theta})\frac{d\widetilde{\varphi}_1}{\alpha} + \Xi_4\sin^2(\widetilde{\theta})\frac{d\widetilde{\varphi}_2}{\alpha} \right)$$

$$\wedge \left( \left(1 + \Xi_2^2\sin^2(\widetilde{\theta})\right)d\varphi_2 - \Xi_4\cos^2(\widetilde{\theta})\frac{d\widetilde{\varphi}_1}{\alpha} - \Xi_2\sin^2(\widetilde{\theta})\frac{d\widetilde{\varphi}_2}{\alpha} \right)$$

$$+\Delta^2\sin(2\widetilde{\theta})\,d\widetilde{\theta} \wedge \left( \left(1 + \Xi_2^2\cos^2(\theta) + \Xi_4^2\sin^2(\theta)\right)\frac{d\widetilde{\varphi}_1}{\alpha} + \Xi_2\cos^2(\theta)\,d\varphi_1 - \Xi_4\sin^2(\theta)\,d\varphi_1 \right)$$

$$\wedge \left( \frac{d\widetilde{\varphi}_2}{\alpha} + \Xi_4\cos^2(\theta)\,d\varphi_1 - \Xi_2\sin^2(\widetilde{\theta})\,d\varphi_2 \right),$$

$$\hat{G}_{(1)} = \hat{G}_{(3)} = \hat{G}_{(5)} = 0, \tag{C.20}$$

with

$$\Delta = \frac{1}{1 + \left(\Xi_2^2\cos^2(\theta) + \Xi_4^2\sin^2(\theta)\right)\sin^2(\widetilde{\theta})}. \tag{C.21}$$

The currents taking part in (229) read

$$j_{\varphi_1} - \alpha\Xi_4 j_{\widetilde{\varphi}_2} = \Delta\left[\left(1 - \Xi_2\Xi_4\sin^2(\widetilde{\theta})\right)j_3^{\text{L}} + \alpha^{-1}\left(\Xi_2\cos^2(\theta) + \Xi_4\sin^2(\theta)\right)\widetilde{j}_3^{\text{L}}\right],$$

$$\alpha j_{\widetilde{\varphi}_1} + \Xi_4 j_{\varphi_2} = \Delta\left[\left(\Xi_2\cos^2(\widetilde{\theta}) - \Xi_4(1+\Xi_2^2)\sin^2(\widetilde{\theta})\right)j_3^{\text{L}} + \alpha^{-1}\left(1 + \Xi_2(\Xi_2\cos^2(\theta) + \Xi_4\sin^2(\theta))\right)\widetilde{j}_3^{\text{L}}\right],$$

$$\bar{j}_{\varphi_1} - \alpha\Xi_4\bar{j}_{\widetilde{\varphi}_2} = \Delta\left[\left(\Xi_2\Xi_4\sin^2(\widetilde{\theta}) + 1\right)j_3^{\text{R}} + \alpha^{-1}\left(\Xi_2\cos^2(\theta) - \Xi_4\sin^2(\theta)\right)\widetilde{j}_3^{\text{R}}\right],$$

$$\alpha\bar{j}_{\widetilde{\varphi}_1} + \Xi_4\bar{j}_{\varphi_2} = \Delta\left[\left(\Xi_2\cos^2(\widetilde{\theta}) + \Xi_4(1+\Xi_2^2)\sin^2(\widetilde{\theta})\right)j_3^{\text{R}} + \alpha^{-1}\left(1 + \Xi_2(\Xi_2\cos^2(\theta) - \Xi_4\sin^2(\theta))\right)\widetilde{j}_3^{\text{R}}\right], \tag{C.22}$$

with $\Delta$ in (C.21) and $j_3^{\text{L}}$, $j_3^{\text{R}}$, $\widetilde{j}_3^{\text{L}}$ and $\widetilde{j}_3^{\text{R}}$ as in (C.16)-(C.17). In this case, the currents $j_{\varphi_1}$, $j_{\varphi_2}$, $j_{\widetilde{\varphi}_1}$ and $j_{\widetilde{\varphi}_2}$ in (220) are not separately holomorphic, but for the combinations above,

$$\bar{\partial}\left(j_{\varphi_1} - \alpha\Xi_4 j_{\widetilde{\varphi}_2}\right) = \tfrac{1}{2}\mathcal{E}_{\varphi_1} - \tfrac{1}{2}\mathcal{E}_{\varphi_2} - \tfrac{\alpha}{2}\left(\Xi_4 + \Xi_2\right)\mathcal{E}_{\widetilde{\varphi}_2},$$

$$\bar{\partial}\left(\alpha j_{\widetilde{\varphi}_1} + \Xi_4 j_{\varphi_2}\right) = \tfrac{1}{2}\left(\Xi_4 - \Xi_2\right)\mathcal{E}_{\varphi_2} + \tfrac{\alpha}{2}\mathcal{E}_{\widetilde{\varphi}_1} - \tfrac{\alpha}{2}\left(1+\Xi_2^2\right)\mathcal{E}_{\widetilde{\varphi}_2},$$

$$\partial\left(\bar{j}_{\varphi_1} - \alpha\Xi_4\bar{j}_{\widetilde{\varphi}_2}\right) = \tfrac{1}{2}\mathcal{E}_{\varphi_1} + \tfrac{1}{2}\mathcal{E}_{\varphi_2} + \tfrac{\alpha}{2}\left(\Xi_2 - \Xi_4\right)\mathcal{E}_{\widetilde{\varphi}_2},$$

$$\partial\left(\alpha\bar{j}_{\widetilde{\varphi}_1} + \Xi_4\bar{j}_{\varphi_2}\right) = \tfrac{1}{2}\left(\Xi_4 + \Xi_2\right)\mathcal{E}_{\varphi_2} + \tfrac{\alpha}{2}\mathcal{E}_{\widetilde{\varphi}_1} + \tfrac{\alpha}{2}\left(1+\Xi_2^2\right)\mathcal{E}_{\widetilde{\varphi}_2}. \tag{C.23}$$

# D  Uplift of 6d $\mathcal{N} = (1,1)$ supergravity

In this appendix we give a self-contained account of the consistent truncation of the NSNS sector of type II supergravity on a four-torus down $\mathcal{N} = (1,1)$ supergravity in six dimensions. The bosonic fields of the latter comprise the metric, the dilaton, four one-forms and a two-form,

$$\{g_{\text{mn}},\ \phi,\ A_{\text{m}}{}^a,\ B_{\text{mn}}\}, \tag{D.1}$$

with indices $\text{m},\text{n} \in [\![0,5]\!]$ and $a \in [\![4,7]\!]$. The field strengths associated to the vectors and two-form are

$$F_{\text{mn}}{}^a = 2\,\partial_{[\text{m}}A_{\text{n}]}{}^a, \qquad \text{and} \qquad H_{\text{mnp}} = 3\,\partial_{[\text{m}}B_{\text{np}]} - \frac{3}{2}\,\delta_{ab}A_{[\text{m}}{}^a F_{\text{np}]}{}^b. \tag{D.2}$$

The action is given by

$$S = \int d^6x\,\sqrt{-g}\left(R - \partial_{\text{m}}\phi\,\partial^{\text{m}}\phi - \frac{1}{4}e^{-\phi}\delta_{ab}F_{\text{mn}}{}^a F^{\text{mn}\,b} - \frac{1}{12}e^{-2\phi}H_{\text{mnp}}H^{\text{mnp}}\right). \tag{D.3}$$

This six dimensional theory can be obtained from the NSNS sector of all superstring theories. The bosonic field content of this sector is given by a metric, a dilaton and a two-form,

$$\{\hat{g}_{s\,\hat{\mu}\hat{\nu}},\ \hat{\Phi},\ \hat{B}_{\hat{\mu}\hat{\nu}}\},\tag{D.4}$$

with indices $\hat{\mu},\hat{\nu}\in[\![0,9]\!]$. The field strength associated to the two-form is

$$\hat{H}_{\hat{\mu}\hat{\nu}\hat{\rho}}=3\,\partial_{[\hat{\mu}}\hat{B}_{\hat{\nu}\hat{\rho}]}\,.\tag{D.5}$$

The action in string frame is given by

$$S=\int \mathrm{d}^{10}x\,\sqrt{-\hat{g}_{s}}\,e^{-2\hat{\Phi}}\left(\hat{R}_{s}+4\,\partial_{\hat{\mu}}\hat{\Phi}\partial^{\hat{\mu}}\hat{\Phi}-\frac{1}{12}\hat{H}_{\hat{\mu}\hat{\nu}\hat{\rho}}\hat{H}^{\hat{\mu}\hat{\nu}\hat{\rho}}\right).\tag{D.6}$$

To compactify on the four-dimensional torus $\mathrm{T}^{4}$, we use the index split $X^{\hat{\mu}}=\{x^{\mathrm{m}},y^{a}\}$, with index ranges as before, and drop the dependence of all fields on the internal coordinates $y^{a}$. We consider the following Kaluza-Klein Ansätze:

$$\mathrm{d}\hat{s}_{s}^{2}=\tilde{g}_{\mathrm{mn}}\mathrm{d}x^{\mathrm{m}}\mathrm{d}x^{\mathrm{n}}+G_{ab}\left(\mathrm{d}y^{a}+A_{\mathrm{m}}^{(1)\,a}\mathrm{d}x^{\mathrm{m}}\right)\left(\mathrm{d}y^{b}+A_{\mathrm{n}}^{(1)\,b}\mathrm{d}x^{\mathrm{n}}\right),\tag{D.7}$$
$$\hat{B}_{(2)}=\tfrac{1}{2}B_{\mathrm{mn}}\mathrm{d}x^{\mathrm{m}}\wedge \mathrm{d}x^{\mathrm{n}}+A_{\mathrm{m}b}^{(2)}\mathrm{d}x^{\mathrm{m}}\wedge\left(\mathrm{d}y^{b}+A_{\mathrm{n}}^{(1)\,b}\mathrm{d}x^{\mathrm{n}}\right)+\tfrac{1}{2}B_{ab}\left(\mathrm{d}y^{a}+A_{\mathrm{m}}^{(1)\,a}\mathrm{d}x^{\mathrm{m}}\right)\wedge\left(\mathrm{d}y^{b}+A_{\mathrm{n}}^{(1)\,b}\mathrm{d}x^{\mathrm{n}}\right),$$

in terms of a six-dimensional metric $\tilde{g}_{\mathrm{mn}}$, two-form $B_{\mathrm{mn}}$, vector fields $A_{\mathrm{m}}^{(1)\,a}$ and $A_{\mathrm{m}a}^{(2)}$, and scalar fields $G_{ab}=G_{ba}$ and $B_{ab}=-B_{ba}$. From a six-dimensional perspective, upon reducing on $\mathrm{T}^{4}$, the ten-dimensional gravity multiplet (D.4) gives rise to a six-dimensional gravity multiplet coupled to four vector multiplets

$$\mathrm{GRAV}_{10}\to\mathrm{GRAV}_{6}\oplus 4\times\mathrm{VEC}_{6}\,.\tag{D.8}$$

The reduced action can be cast in the SO(4,4)-covariant form:

$$S=\int \mathrm{d}^{6}x\,\sqrt{-\tilde{g}}\,e^{-2\Phi}\left(\tilde{R}+4\,\partial_{\mathrm{m}}\Phi\,\partial^{\mathrm{m}}\Phi+\frac{1}{8}\,\partial_{\mathrm{m}}\mathcal{H}_{\mathbb{A}\mathbb{B}}\partial^{\mathrm{m}}\mathcal{H}^{\mathbb{A}\mathbb{B}}-\frac{1}{4}\mathcal{H}_{\mathbb{A}\mathbb{B}}\mathcal{F}_{\mathrm{mn}}{}^{\mathbb{A}}\mathcal{F}^{\mathrm{mn}\,\mathbb{B}}-\frac{1}{12}H_{\mathrm{mnp}}H^{\mathrm{mnp}}\right),\tag{D.9}$$

with $\Phi=\hat{\Phi}-\ln(\det(G_{ab}))/2$, the vector fields joined into a single SO(4,4) vector $\mathcal{A}_{\mathrm{m}}{}^{\mathbb{A}}$ and the scalar fields parameterising the coset SO(4,4)/(SO(4)×SO(4)) through the SO(4,4) matrix $\mathcal{H}_{\mathbb{A}\mathbb{B}}$. The three-form is given by

$$H_{\mathrm{mnp}}=3\,\partial_{[\mathrm{m}}B_{\mathrm{np}]}-\frac{3}{2}\eta_{\mathbb{A}\mathbb{B}}\mathcal{A}_{[\mathrm{m}}{}^{\mathbb{A}}\mathcal{F}_{\mathrm{np}]}{}^{\mathbb{B}}\,.\tag{D.10}$$

In a basis where the SO(4,4) invariant matrix $\eta_{\mathbb{M}\mathbb{N}}$ takes the form

$$\eta_{\mathbb{A}\mathbb{B}}=\begin{pmatrix}0 & \delta_{a}{}^{b}\\ \delta^{b}{}_{a} & 0\end{pmatrix},\tag{D.11}$$

the SO(4,4) fields are parameterised as follows in terms of the Ansätze (D.7)

$$\mathcal{A}_{\mathrm{m}}{}^{\mathbb{A}}=\begin{pmatrix}A_{\mathrm{m}}^{(1)\,a}\\ A_{\mathrm{m}a}^{(2)}\end{pmatrix},\tag{D.12}$$

$$\mathcal{H}_{\mathbb{A}\mathbb{B}}=\begin{pmatrix}G_{ab}-B_{ac}G^{cd}B_{db} & B_{ac}G^{cb}\\ -G^{ac}B_{cb} & G^{ab}\end{pmatrix}.\tag{D.13}$$

If we further move to the Einstein frame by redefining $\tilde{g}_{\mathrm{mn}}\to g_{\mathrm{E\,mn}}=e^{-\Phi}\tilde{g}_{\mathrm{mn}}$, then

$$S=\int \mathrm{d}^{6}x\,\sqrt{-g_{\mathrm{E}}}\left(R_{\mathrm{E}}-\partial_{\mathrm{m}}\Phi\,\partial^{\mathrm{m}}\Phi+\frac{1}{8}\,\partial_{\mathrm{m}}\mathcal{H}_{\mathbb{A}\mathbb{B}}\partial^{\mathrm{m}}\mathcal{H}^{\mathbb{A}\mathbb{B}}-\frac{1}{4}e^{-\Phi}\mathcal{H}_{\mathbb{A}\mathbb{B}}\mathcal{F}_{\mathrm{mn}}{}^{\mathbb{A}}\mathcal{F}^{\mathrm{mn}\,\mathbb{B}}-\frac{1}{12}e^{-2\Phi}H_{\mathrm{mnp}}H^{\mathrm{mnp}}\right).\tag{D.14}$$

To reduce (D.14) down to the minimal $\mathcal{N} = (1,1)$ theory (D.3), we need to truncate the four vector multiplets in (D.8). Therefore, we consider the truncation to SO(4) singlets in SO(4, 4). There are two possible SO(4) factors in SO(4, 4), denoted SO(4)$_\pm$:[15]

$$\mathrm{SO(4)}_+ \times \mathrm{SO(4)}_- \subset \mathrm{SO(4,4)}\,, \tag{D.15}$$

which respectively rotate $\pm\mathbb{1}$ in the basis in which the invariant SO(4,4) metric is diagonal. Both truncations (either to singlets of SO(4)$_+$ or the ones of SO(4)$_-$) leave 4 vectors in $\mathcal{A}_\mathrm{m}{}^\mathbb{A}$ and no scalar in $\mathcal{H}_{\mathbb{AB}}$ ($\mathcal{H}_{\mathbb{AB}} = \delta_{\mathbb{AB}}$), thus reproducing the field content (D.1). The truncation to SO(4)$_-$ singlets then matches the field-strengths (D.2) and the action (D.3) upon identifying

$$g_{\mathrm{E\,mn}} = g_{\mathrm{mn}}\,, \qquad \Phi = \phi\,. \tag{D.16}$$

In the basis where $\eta_{\mathbb{AB}}$ takes the off-diagonal form (D.11), $\mathcal{A}_\mathrm{m}{}^\mathbb{A}$ and $\mathcal{H}_{\mathbb{AB}}$ are given by

$$\mathcal{A}_\mathrm{m}{}^\mathbb{A} = \frac{1}{\sqrt{2}}\begin{pmatrix} A_\mathrm{m}{}^a \\ A_\mathrm{m}{}^a \delta_{ab} \end{pmatrix}\,, \tag{D.17}$$

$$\mathcal{H}_{\mathbb{AB}} = \delta_{\mathbb{AB}}\,, \tag{D.18}$$

and the field strength (D.10) reduces to (D.2). Therefore, the embedding of the minimal $\mathcal{N} = (1,1)$ theory in 6$d$ into ten dimensions reads

$$\begin{aligned}
\hat{\Phi} &= \phi\,, \\
\mathrm{d}\hat{s}_s^2 &= e^\phi g_{\mathrm{mn}}\,\mathrm{d}x^\mathrm{m}\mathrm{d}x^\mathrm{n} + \delta_{ab}\Big(\mathrm{d}y^a + \tfrac{1}{\sqrt{2}}A_\mathrm{m}{}^a\,\mathrm{d}x^\mathrm{m}\Big)\Big(\mathrm{d}y^b + \tfrac{1}{\sqrt{2}}A_\mathrm{n}{}^b\,\mathrm{d}x^\mathrm{n}\Big)\,, \\
\hat{B}_{(2)} &= \tfrac{1}{2}B_{\mathrm{mn}}\,\mathrm{d}x^\mathrm{m} \wedge \mathrm{d}x^\mathrm{n} + \tfrac{1}{\sqrt{2}}\delta_{ab}A_\mathrm{m}{}^a\,\mathrm{d}x^\mathrm{m} \wedge \mathrm{d}y^b\,.
\end{aligned} \tag{D.19}$$

The global SO(4) symmetry of $D = 6$ $\mathcal{N} = (1,1)$ supergravity can then be understood as coordinate-independent rotations preserving $\delta_{ab}$, and thus becomes a gauge symmetry in the full ten dimensional description.

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
