# Peer review of "Charting the Conformal Manifold of Holographic CFT$_2$'s"

_SciPost Physics, doi:SciPost Phys. 17, 123 (2024)_

## Round 1 · Referee Report · Riccardo Borsato (Referee 1) · 2024-7-26

Report

The discussion on the worldsheet description of the deformations and the interpretation as current-current deformations should be clarified and possibly corrected. See the requested changes below.

Requested changes

1 - The parameterisation of the group element of SU(2) in (6.5) does not seem right. In fact, both a left and a right transformation generated by σ3 would affect just the combination φ1+φ2. Another way to put it is that it seems that there is no SU(2) transformation capable of affecting φ1φ2. On one of the two sides of g, then, the relative sign of the angles should change.

2 - The definition of the currents in (6.12) is not clear. On the one hand, because they are defined in terms of the Killing vectors, they should probably be interpreted as the Noether currents. It is unclear, however, why the authors use a notation in which it appears that these currents are chiral/antichiral. In general, the Noether currents will not have a definite chirality, and in particular they will be non-vanishing for both values of the worldsheet indices. See for example eq. (5.7) of ref. [69]. For the understanding of (6.12), it does not help that the Killing vectors ki and ˉki have not been defined, for example to obtain the currents in (6.13). I suspect (I hope to be corrected if I'm wrong) that the authors are taking ki=ˉki, and that ji and ˉji in (6.12) are not two different currents, but just the two worldsheet components of the same Noether current. If that is the case, I would ask the authors to stress in the text that, despite the notation, those j,ˉj are not necessarily chiral/antichiral, and that they are just the two components of the same Noether current. I would also ask them not to use k and ˉk but just k, for example, because otherwise the definition becomes very confusing.

3 - The marginal current-current deformations constructed in ref. [7, 68,70] are of the form J1ˉJ2 where J1 is chiral and ˉJ2 is antichiral. If my interpretation in point 2 above is correct, the authors are not describing current-current deformations of that type, so they cannot really make any statement about the relation to the marginal current-current deformations of [7, 68,70]. Following for example the discussion in section 5.2.2 of ref. [69], it is possible to rewrite the Noether currents as chiral/antichiral currents plus "improvement terms", and to expose the form of the current-current deformations in terms of the chiral/antichiral currents. I would ask the authors to either do this explicitly in one of their example, or at least to comment on this point in the paper. The reason why this is important, for example, is that (6.14) does not identify the marginal operator driving the deformation.

4 - I would like to ask the authors to confirm if in (6.14) and in the following equations the tensor product is precisely the operation that they mean to write (and not ). The reason I ask is that TsT deformations are current-current deformations of the form J1J2, where the currents may or may not have chirality properties, and the wedge product is important. The reason why the authors have and not should be related to the fact that they don't discuss "pure" TsT transformations, since they mix them with target-space coordinate transformations and B-field gauge transformations. I would like to ask them to comment on this point.

Recommendation

Ask for major revision

---

## Round 1 · Referee Report · Anonymous (Referee 2) · 2024-8-5

Strengths

1 - The authors construct 17(15)-dimensional deformations of known AdS3 backgrounds in type IIB and Heterotic supergravity. They generalize the results of [40,8,18] to include more parameters.

2 - The authors present a unified framework to study the two cases in gauged supergravity.

3 - They discuss the uplift of some of the solutions to 10D and discuss the spectrum of fluctuations around the background.

4 - They then discuss the holographic intrerpretation of the flat directions as JˉJ deformation of the original 2D SCFT.

Weaknesses

1 - In general the text (notably in the introduction) is not always clear or accurate, it could benefit from some revising.

2 - The paper heavily relies on exceptional field theory and gauged supergravity. However it seems that many of the results can be obtained directly from 10 dimensions using known results in the literature. This is already presented in section 5, but could be presented more prominently earlier on in the paper.

3 - Sometimes it is not clear which parameters are new and which are found in previous work. Also the range of parameters is not stated clearly when it is relevant. I think this should be organized better and maybe stated clearly when (2.51) is first written down.

4 - It is slightly confusing that the authors use half-maximal supergravity instead of maximal given their own earlier works. This is not explained in the paper.

5- the goal of section 6 is not completely clear.

Report

This paper discussed deformations of the well-known AdS3×S3×T4 and AdS3×S3×S3×S1 solutions of type IIB/Heterotic supergravity.

The paper builds on earlier works by the authors, notably in [40,8,18] and it deserverves publication. However, before recommending publication in Scipost I would like to ask the authors a few questions and request clarifications.

Requested changes

1 - In eq. (2.40) a parameter α is introduced in the embedding tensor. As far as I can tell α=0 corresponds to type IIB on S3×T4 and α0 corresponds to type IIB on S3×S3×S1. Is this the correct understanding. If so this can be anticipated already in sec. 2.3 to clarify the discussion.

2 - Also, is α a real parameter? or is it a discrete parameter in the sense that for any α0 one can perform a field redefinition that sets α=1. If this is the case then this must be highlighted for clarity. On the other hand if α is a true parameter, then what does it correspond to in 10D?

3 - Equation (2.51) constitutes one of the main new results in the paper, but it is unclear to me how it is obtained. From the text above it, I understand that this solution is obtained by first uplifting to 10 dimensions where squashings and fibrations can be obviously generalized. Then this general deformation is reduced back to 3D leading to (2.51). Is this a correct understanding? if so what is the benefit of performing this analysis in 3D rather than just directly in 10D? Indeed, a more careful reading of section 5 seems to confirm that all deformations given in (2.51) correspond to rather straight-forward deformations of the original AdS background that are more clearly understood in 10D (I am referring to the discussion on page 29 of the manuscript). If this is true then the paper may benefit from this explanation earlier in the manuscript as for many readers this will improve readability. Currently the manuscripts demands quite extensive knowledge on gauged supergravity and exceptional field theory to follow the text.

4 - In the paragraph below (2.51) it is pointed out that some of the parameters in the solution corresponds to TsT transfromations of the original undeformed vacuum. The authors then comment that it is remarkable that TsT parameters are contained in gauged supergravity. However, I do not think this is surprising for the S3×T4 reduction of type IIB as the TsT transformation can be formulated as a transformation in type IIB (or 11D) on T^3 (see appendix A in ref [19]). It is perhaps a bit more surprising for the S3×S3×S1 case.

5 - In section 6 it is argued that the flat directions in the supergravity solution correspond to JˉJ deformation of the 2D SCFT. Is the claim that for all 17 parameters this is the case or only a subset of them? If all 17 parameters can be thought of as JˉJ deformtaions, can you then predict how large the moduli space of deformations should be using QFT perspective? I ask since in the conclusion it is stated that a search for flat directions in 3D sugra has not been carried out to a conclusive extent, and it would be nice to know how far of the 17-dimensional space is.

Recommendation

Ask for minor revision

---

## Round 1 · Referee Report · Nikolay Bobev (Referee 3) · 2024-8-19

Strengths

1. The authors employ new techniques to construct a large multi-parameter family of novel AdS_3 supergravity solutions with NS-NS fluxes that can be viewed as consistent backgrounds of type II or heterotic string theory.

2. The authors present explicitly some examples of these supergravity solutions and calculate the spectrum of KK excitations in supergravity by employing ExFT techniques and discuss the corresponding 2d CFT interpretation.

Weaknesses

1. The identification of the continuous supergravity parameters with exactly marginal couplings in the holographically dual 2d CFT is discussed only briefly.

2. The discussion of the Zamolodchikov metric on the conformal manifold is also very brief and incomplete.

Report

The authors construct large families of new AdS_3 supergravity solutions by employing techniques from 3d gauged supergravity and Exceptional Field Theory. The paper should be of interest to researchers working on string theory, supergravity, and holography. I think this work merits publication.

Requested changes

I recommend that the authors address the following points before publication

1. The difference between marginal and exactly marginal should be explained and emphasized more strongly in the Introduction, especially given that the discussion in the paper is mostly valid to leading order in the large N limit of the dual 2d CFT and thus exact marginality is not always guaranteed.

2. The meaning of parameter \alpha introduced in Section 2 from the 10d perspective should be explained more clearly. Is it the same as the parameter \alpha used in Section 4.3? If yes, this too should be emphasized.

3. The notion/definition of a TsT transformation should be explained better somewhere in the paper, perhaps at the beginning of Section 5 or in the Introduction.

4. The authors should provide more details on how the metric in (5.26) is computed. The authors should also consider calculating and presenting the Zamolodchikov metric for the examples studied in Section 5.2, 5.3 and 5.4.

5. It is unclear to me why below (6.27) the authors say that one needs to calculate 3pt functions of KK modes in order to check exact marginality. Presumably this should be done by doing a worldsheet calculation? I think that this point needs further clarification.

Recommendation

Ask for minor revision

---

## Round 2 · Referee Report · Anonymous (Referee 4) · 2024-9-21

Report

The authors have addressed all the comments raised in my previous report and I recommend the manuscript for publication in Scipost.

Recommendation

Publish (meets expectations and criteria for this Journal)

---

## Round 2 · Referee Report · Nikolay Bobev (Referee 3) · 2024-9-21

Report

The authors have address the comments in my previous report and therefore I recommend the paper for publication.

Recommendation

Publish (meets expectations and criteria for this Journal)

---

## Round 2 · Referee Report · Riccardo Borsato (Referee 1) · 2024-10-2

Report

I thank the authors for addressing my points and for improving the paper. I have no further request and I recommend the paper for publication.

I take the opportunity to reply to the comment of the authors on the wedge vs tensor product of the current bilinears. What I called "pure" TsT would correspond to a deformation with $\rho=Id$ and $s=0$. In that case, the infinitesimal deformation would give only the antisymmetric combination of the currents (with the wedge product). I think this is also what the authors mean when they say that the wedge vs tensor product is a "gauge-dependent statement".

Recommendation

Publish (meets expectations and criteria for this Journal)

---

## Round 2 · Author Response

Dear editor, 

Thank you for providing us with feedback from the referees. We proceed to addressing their points in order.

Regarding the comments in Report 1,

  • In our conventions, the second factor in (6.5) should indeed appear with a minus sign. We thank the referee for spotting this typo. It has been fixed in the current submission together with a sign in $j_1^L + i j_2^L$ in eq. (6.8).

  • We have expanded the discussion in section 6.1 to clarify our method, and expanded on the results. These currents that participate in the deformations are defined in terms of the Killing vectors in eq. (6.14) via (6.13). We have checked explicitly that the currents thus defined are (anti-)holomorphic upon imposing the equations of motion, and details of this computation are included in the current manuscript in app. C. The notation to refer to the currents has also been updated for the sake of clarity.

  • As mentioned in point 2 above, we have now proven that all currents involved in the deformations are (anti-)holomorphic. Exact marginality can thus be claimed following the results of [7, 68, 70].

  • The deformations in (6.17) (previously (6.14)) and below need to be described in terms of tensor products, and not wedge products in general. For the TsT deformation in (6.17), which is parameterised by $\beta$ as discussed in section 5.1 of ref. [69], the deformation is indeed proportional \mbox{$\delta\beta_{1}(j_{\varphi_{1}}\otimes \bar{j}{y^{7}}-j}\otimes \bar{j}{\varphi})$}. Nevertheless, it should be noted that associating TsT deformations to antisymmetric products is a gauge-dependent statement. An instance of this fact is the deformation in (6.28) (previous (6.23)) with $\delta\beta_1=\delta\tilde{\beta}_1=\delta\tilde{\beta}_2=0$, which is symmetric in the currents but still a pure TsT transformation in the frame described below eq. (5.40).

Concerning the weaknesses that the second report notes,

  • Some extra details have been introduced that hopefully make the discussion clearer.

  • The fact that the solutions can be obtained by other means was only apparent after the $3d$ vacua were noted to be inside SO(7,7), as discussed in Section 5. The gauged supergravity techniques are more general and could lead to other solutions, as our set has not been proven to be exhaustive, and are a necessary ingredient for the spectral analysis which is used to assess perturbative stability.

  • Some further comments have been added regarding this point below (2.51), although given the different parameterisation with respect to the other relevant references in the literature, we defer the detailed comparison to section 5. Regarding the range of the parameters, it remains in general an open question.

  • Half-maximal supergravity both in $3d$ and $10d$ is used for the uplifts into heterotic supergravity. For type IIB, the solutions we present can be found in the half-maximal truncation, but their spectral analysis requires the complete theory. We have added some extra remarks on this in the introduction and section 3.

  • A brief motivation has been included at the beginning of the section.

As for the requested changes in that report,

  • The understanding of the parameter $\alpha$ is correct, and we have added clarifications around eq. (2.40), (4.2), (4.12) and (4.30).

  • The parameter is real and takes values in the unit interval, as noted in footnote 6. Its value affects the geometry and the masses, as can be seen in e.g. eq. (4.33), and therefore different values correspond to physically different solutions. In $10d$, $\alpha$ corresponds to the ratio of the sphere radii. This has been added around eq. (2.40), (4.12) and (4.30).

  • In this case, this understanding is not correct. As remarked before, the solutions were obtained directly in $3d$ by a careful analysis of the scalar potential, and only later uplifted via ExFT methods; and this $3d$ origin allows an otherwise unattainable study of the solutions. We have highlighted this before (2.51).

  • The results in app. A of ref. [19] would inform consistent truncations only if the SL$(3,\mathbb{R})$ transformation acted exclusively on the $T^3$ angles. In this case, we do agree it would be unsurprising. However, even in the $S^3\times T^4$ case, the TsT transformations discussed in the text do involve angles on the $S^3$, where the link to the consistent truncation is much less obvious due to the absence of a global symmetry. We have clarified this on page 46 in the discussion, with the sentence ``Among the directions in the conformal manifold, the possibility of describing TsT deformations acting on spheres in a consistent truncation is a three-dimensional peculiarity, as in higher-dimensions the moduli triggering those transformations sit within higher Kaluza-Klein levels''.

  • To the best of the authors' knowledge, no QFT techniques are available for non-supersymmetric conformal manifolds, except for the case of current-current deformations.

Regarding the requested changes in the third report,

  • We have added new comments about the supergravity approximation and the distinction between classically marginal and exactly marginal in the second paragraph of the introduction.

  • We have added clarifications around eq. (2.40), (4.2), (4.12) and (4.30) about the r\^ole of the parameter $\alpha$.

  • In section 5, the notion of TsT transformation is now explained before eq. (5.3).

  • Also in section 5, the general strategy to obtain the Zamolodchikov metric is introduced (after eq. (5.26)) as well as the line elements corresponding to the different examples in sections 5.2--5.4.

  • The relation between three-point functions and marginal deformations is advanced in the introduction and better explained at the beginning of sec. 6.2. Following ref. [71], the corrections to the beta-functions in CFT perturbation theory are captured by the three-point functions of the supergravity fields, and one can extend this computation to higher orders in perturbation theory. A worldsheet calculation would indeed be necessary to claim exact marginality, as we do for the current-current deformations. We acknowledge that the holographic description of the marginal deformations identified in the text is brief, and plan to expand it future work.

Best regards,

The authors

---

## Editorial Decision

published